# Retinoic acid signaling modulation guides in vitro specification of human heart field-specific progenitor pools

Dorota Zawada [1,2,3,16], Jessica Kornherr[1,2,3,16], Anna B. Meier[1,2,3,16], Gianluca Santamaria [1,2,3,4,16], Tatjana Dorn[1,2,3], Monika Nowak-Imialek[1,2,3], Daniel Ortmann[5,6], Fangfang Zhang[1,2,3], Mark Lachmann[1,2], Martina Dreßen [7], Mariaestela Ortiz[5], Victoria L. Mascetti[8], Stephen C. Harmer [9], Muriel Nobles[10], Andrew Tinker [10], Maria Teresa De Angelis[1,2,4], Roger A. Pedersen[11], Phillip Grote [12,13], Karl-Ludwig Laugwitz [1,2] ✉, Alessandra Moretti [1,2,3,14] ✉ & Alexander Goedel [1,15] ✉

Cardiogenesis relies on the precise spatiotemporal coordination of multiple progenitor populations. Understanding the specification and differentiation of these distinct progenitor pools during human embryonic development is crucial for advancing our knowledge of congenital cardiac malformations and designing new regenerative therapies. By combining genetic labelling, single-cell transcriptomics, and ex vivo human-mouse embryonic chimeras we uncovered that modulation of retinoic acid signaling instructs human pluripotent stem cells to form heart field-specific progenitors with distinct fate potentials. In addition to the classical first and second heart fields, we observed the appearance of juxta-cardiac field progenitors giving rise to both myocardial and epicardial cells. Applying these findings to stem-cell based disease modelling we identified specific transcriptional dysregulation in first and second heart field progenitors derived from stem cells of patients with hypoplastic left heart syndrome. This highlights the suitability of our in vitro differentiation platform for studying human cardiac development and disease.

The heart is the first organ that forms during embryogenesis and its functionality depends on the proper spatiotemporal assembly of multiple progenitor cell populations that ultimately give rise to the diverse cell types within the specialized cardiac structures.

Lineage tracing and clonal cell analyses in mice have identified two main sources of cardiovascular progenitors in the developing mesoderm, known as the first and second heart fields (FHF and SHF), which show differential contribution to specific heart compartments[1]. A growing body of evidence suggests that the specification of FHF and SHF progenitors already occurs during gastrulation depending on their position within the primitive streak, which impacts the signaling cues they receive[2,3]. Moreover, besides having discrete molecular signatures at primitive streak stages, progenitors that later contribute to

the ventricles or to the outflow tract (OFT) and atria leave the streak at different time points and form distinct regions of the cardiogenic mesoderm[4]. The heterogeneity of the heart fields is further highlighted by the recent discovery of a multipotent cardiac progenitor that generates cardiomyocytes (CMs) of the left ventricle (LV), atria and atrioventricular canal (AVC) as well as cells of the epicardium. This so-called juxta-cardiac field (JCF) resides in a distinct region within the FHF in close proximity to the extraembryonic tissue[5,6].

The patterning of the primitive streak within the embryo occurs in response to specific signaling cues including Wnt, BMP, and Activin signaling[7]. Stimulation and inhibition of these signaling pathways in a temporally controlled manner can be used to direct in vitro differentiation of human pluripotent stem cells (hPSCs) into a posterior or

anterior primitive streak-like fate, with cardiovascular progenitor cells arising from the latter[8]. Formation of the anterior-posterior boundaries of the cardiogenic mesoderm is, in part, controlled by retinoic acid (RA) signaling[9], which restricts the border of the anterior SHF (aSHF−ultimately giving rise the outflow tract (OFT) and the right ventricle (RV)[10] while maintaining its posterior component (posterior SHF (pSHF)−which generates the atria and the sinus venosus). Posteriorization of the SHF through RA signaling has been utilized during cardiac differentiation of hPSCs to promote the formation of atrial CMs and epicardial cells[11–14]. Notably, the addition of low-dose RA during early mesoderm formation results in the generation of hPSC-derived cardiovascular progenitors that can self-organize into three-dimensional (3D) structures comprising of CMs and endocardial cells, as reported recently[15]. On a transcriptional level, these progenitors resemble murine FHF cells, which are the first to migrate from the primitive streak during mid-streak stages, form the primitive heart tube, and later give rise to the LV[4,15]. Such effect of RA during early in vitro cardiogenesis could reflect its inhibitory role on aSHF development providing a permissive environment for FHF lineage commitment, which has yet not been investigated in detail.

Our study aimed at systemically defining how RA impacts human cardiovascular progenitor specification during early mesoderm formation. We approached this by combining genetic labeling, transcriptomics, ex vivo human-mouse embryonic chimeras, and in vitro disease modeling. Our findings indicate that a wide spectrum of mesoderm-derived cardiovascular progenitor cells including progenitors of the JCF can be differentiated from hPSCs by modulating RA signaling in early mesoderm-like cells. These progenitors contribute to the expected cardiac structures when injected into the cardiac crescent of ex vivo cultured mouse embryos, highlighting the similarity to their native counterparts. Moreover, they allowed us to uncover heart field-specific differentiation defects in an in vitro disease model of hypoplastic left heart syndrome (HLHS), a congenital heart disease that primarily affects the LV and left ventricular outflow tract (LVOT). Overall, this study presents a versatile toolbox for the generation of defined cardiovascular progenitor pools from hPSCs and shows its applicability for studying chamber-specific heart disease.

## Results

### RA signaling and cardiovascular progenitor specification

To investigate the influence of RA dosage and timing on the appearance and characteristics of early human cardiovascular progenitors in the cardiogenic mesoderm, we utilized a growth-factor based protocol for the directed differentiation of hPSCs towards CMs, which is distinct from other in vitro CM differentiation protocols utilizing RA to generate atrial CMs[11,13,14] (Fig. 1a). In the absence of RA, mesoderm induction (TBXT) was detected within the first 24 h of differentiation, followed by the expression of early cardiac progenitor markers (NKX2.5, ISL1) at day 3, shortly after Wnt signaling inhibition (Fig. 1b). Concomitant upregulation of BMP4, FGF10, CXCR4, and LGR5 along with TBX1 and WNT5A at day 5 suggested that these cells adopted a fate resembling cardiovascular progenitors of the aSHF (Fig. 1c, d; Supplementary Fig. 1a)[16–19]. Interestingly, addition of RA from day 1.5 to day 5.5 resulted in a dose-dependent downregulation of aSHF markers at the progenitor state (days 4−5) and an upregulation of genes related to a posterior fate, such as WNT2 and HOXB1, suggesting a posteriorization of the cardiac progenitor pool[17,20,21] (Fig. 1d). This is in line with observations during cardiac development in the mouse, where RA is critical in establishing the anterior-posterior axis within the heart fields[22]. The expression level of the pan cardiac progenitor marker NKX2.5 was comparable between the different treatments reflecting a similar efficiency in cardiac progenitor specification (Supplementary Fig. 1a). Notably, we also detected a strong upregulation of TBX5, which at the equivalent stage in vivo marks cardiac progenitors of the murine FHF[17], as well as THBS4, which is highly enriched in FHF progenitor cells

in the mouse embryo at embryonic day 7.75 (E7.75)[6] (Fig. 1d). During further differentiation, we observed earlier and higher expression of sarcomeric genes such as TNNT2 and MYL3 in presence of RA compared to differentiation without RA, which was accompanied by a faster downregulation of ISL1 (Fig. 1e, Supplementary Fig. 1b). In the first 24 h after RA application ISL1 expression remained similar in both conditions suggesting that RA does not directly downregulate ISL1 but rather induces changes in the transcriptional network leading to its rapid downregulation (Fig. 1e). In concordance with these transcriptional changes, beating foci emerged on average two days earlier after addition of RA (Fig. 1f). Notably, rapid downregulation of ISL1 expression and rapid generation of functional CMs are among the characteristic features of FHF cardiovascular progenitors, which are necessary to form the primitive heart tube in vivo[4].

At day 30, expression levels of TNNT2 as well as the percentage of cTnT+ cells were comparable between differentiations with and without RA, suggesting efficient CM differentiation in both conditions (Supplementary Fig. 1c, d). Without RA, cells expressed high levels of ventricular CM genes such as MYL2, IRX4, and IRX3, while atrial and nodal CM markers such as NR2F2, KCNA5, KCNJ3, and SHOX2 were absent (Fig. 2a; Supplementary Fig. 1d). In addition, these cells expressed transcripts typical of CMs found in the OFT, such as LTBP3 and RSPO3[18] (Fig. 2a), providing further evidence that this differentiation condition induces an aSHF-like fate. Upon RA addition, we detected a dose-dependent change in expression of FHL2 (Fig. 2a), which was reported to be enriched in left ventricular CMs of adult human hearts[23]. At low dosage of RA (0.5 μM), FHL2 expression increased with time of treatment, but it declined again at higher dose (1 μM) and prolonged treatment. We observed similar dynamics for the ventricular marker MYL2. Genes associated with CMs of the OFT showed an inverse relation to the FHL2 expression pattern with the lowest expression levels at 0.5 μM RA for 4 days (Fig. 2a). Atrial CM markers increased only marginally at low dosage of RA, but a more substantial expression was observed upon higher dose and longer treatment (Fig. 2a, Supplementary Fig. 1d). Taken together, these results suggest that the occurrence of FHF-like progenitors is tightly controlled by timing and dosage of RA addition.

To confirm our findings on the transcriptomic level we performed immunofluorescence analysis for the ventricular specific myosin light chain isoform−MLC2v, and atrial isoform−MLC2a (Fig. 2b) as well as flow cytometry-based quantification of cells expressing the atrial specific protein NR2F2 (Fig. 2c, Supplementary Fig. 1e). Without addition of RA, most of the cells were exclusively positive for MLC2v and we observed only around 10% of NR2F2+ cells, confirming that most cells acquired a ventricular-like fate (Fig. 2b, c). This did not change upon addition of RA up to 1 μM for 4 days (Fig. 2b, c). However, longer exposure to RA (1 μM for 10 days) resulted in an increase of MLC2a+ CMs (Fig. 2b) as well as NR2F2+ cells (Fig. 2c), suggesting partial atrial fate acquisition in line with previous reports[11].

To further explore the identity of the CMs generated with and without RA, we performed electrophysiological examination and focused on the highest RA dose that did not lead to a significant increase in atrial fate acquisition (1 μM for 4 days). In line with our findings on the transcriptomic and protein level, this analysis showed that the majority of CMs obtained in both conditions had indeed ventricular identity (Fig. 2d, e; Supplementary Data 1). Without RA, over a quarter of cells showed "intermediate" action potential characteristics, suggesting that they have not fully differentiated yet (Fig. 2d, e; Supplementary Data 1). However, among the cells classified as ventricular action potential properties such as the mean diastolic potential and amplitude were comparable between the two groups (Fig. 2f). Upon RA addition, we did observe a slight increase in cells with a nodal-like action potential (Fig. 2d) which is in line with the observed increase in SHOX2 expression (Supplementary Fig. 1d), probably reflecting a permissive environment for sinus nodal commitment[24].

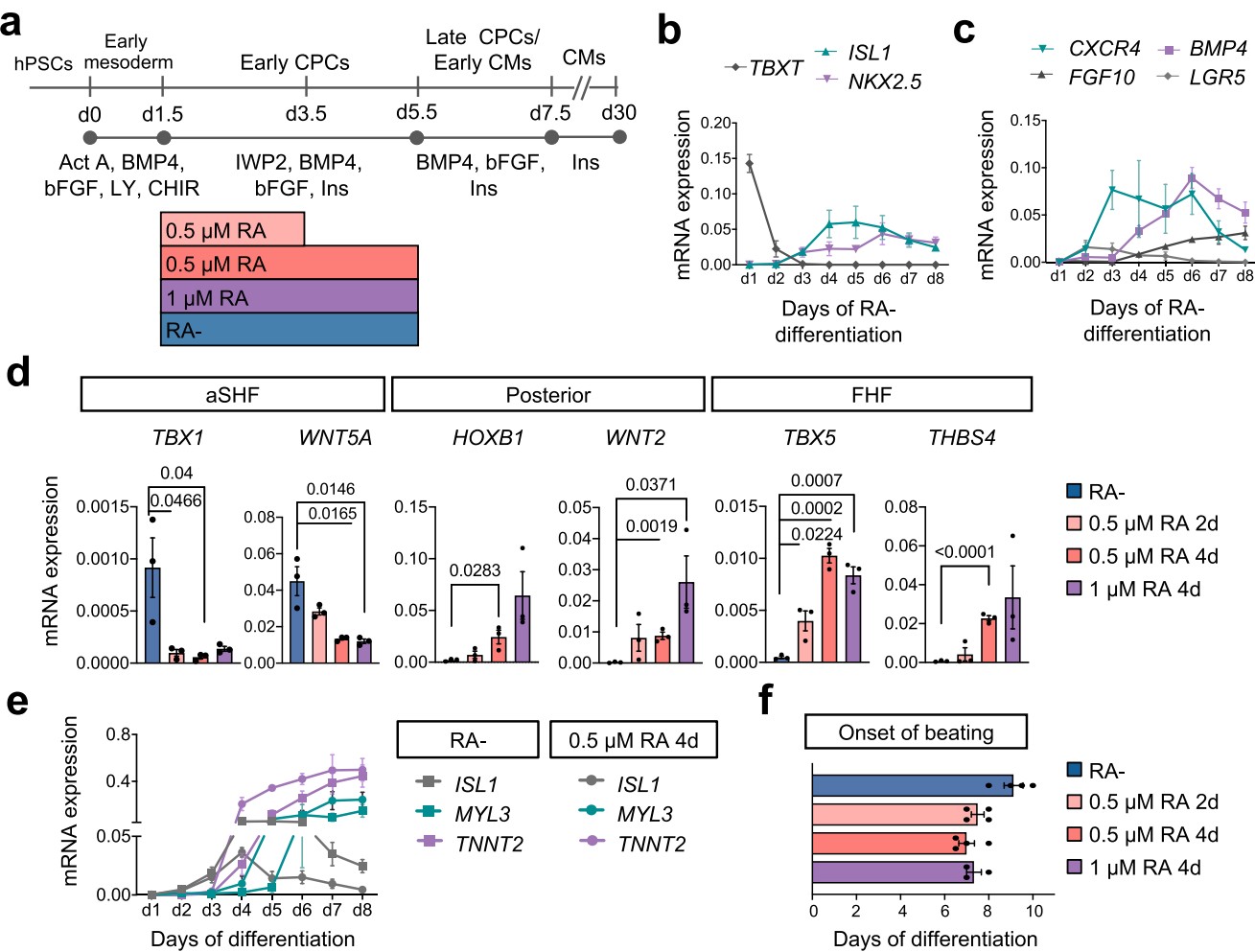

**Fig. 1 | RA signaling modulation impacts cardiovascular progenitor specification. a** Schematic representation of the protocol used to differentiate human pluripotent stem cells (hPSCs) into cardiomyocytes (CMs) through defined steps of early mesoderm and cardiovascular progenitor cells (CPCs) without retinoic acid (RA−) or with RA added at the indicated dosages and times. Act A Activin A, CHIR CHIR-99021, LY LY-29004, Ins insulin. **b, c** Time course of mRNA expression of **b** *TBXT, ISL1, NKX2.5*, and **c** *CXCR4, FGF10, BMP4, LGR5* during differentiation without RA. Data are mean ± SEM. For *FGF10 n* = 4 independent experiments; for all other genes *n* = 3. mRNA expression relative to *GAPDH*. **d** mRNA expression of key anterior second heart field (aSHF) markers, posteriorization markers, and first heart field (FHF) markers at day 5 the indicated differentiation conditions. Data are mean ± SEM; *n* = 3 independent experiments. Exact *p*-values of unpaired two-tailed *t*-test are shown. **e** Time course of mRNA expression of *ISL1, MYL3*, and *TNNT2* during RA− and RA + (0.5 µM 4d) differentiation. Data are mean ± SEM. *n* = 3 independent experiments. **f** Day of the onset of spontaneous beating at the indicated differentiation conditions. Data are mean ± SEM; for RA+(1 µM 4d) *n* = 3 independent experiments; for all other conditions *n* = 4.

Based on these findings, we hypothesized that a time- and dose-limited exposure to RA during early mesoderm formation is not efficient in inducing a pSHF-like fate resulting in atrial CMs, but rather promotes the specification of progenitors resembling a FHF-like fate, resulting in CMs with a (left) ventricular identity. Since the intermediate dose of RA (0.5 µm, 4 days) was sufficient to suppress aSHF genes and induce FHF commitment (Fig. 1d) we decided to use this dose in further experiments.

To track the appearance of distinct cardiovascular progenitor pools and enable enrichment of specific subpopulations for in-depth analysis, we generated a double reporter human embryonic stem cell (hESC) line expressing eGFP and nuclear mCherry (H2B-mCherry) under the control of the endogenous *NKX2.5* and *TBX5* locus, respectively (ESO3 TN cell line, Fig. 3a). We confirmed proper reporting of the fluorescent markers using live imaging (Fig. 3b), qPCR, and immunofluorescence staining (Supplementary Fig. 2a, b). During differentiation, the modified hESC line displayed only a slight reduction in expression levels of *NKX2.5* and *TBX5* compared to the parental line, most likely due to partial compensation of the loss of transcription from the targeted gene allele by the untargeted allele (Supplementary

Fig. 2c). In addition, the expression patterns of key cardiac differentiation genes and the percentage of cTnT⁺ cells were comparable between the reporter and parental line as well as an unrelated human induced pluripotent stem cell (hiPSC) line, suggesting that the reporter line is not affected by relevant haploinsufficiency of the targeted transcription factors (Supplementary Figs. 2c, 3a–c). Without addition of RA, eGFP⁺ cells (NKX2.5⁺) appeared from day 3 while mCherry⁺ (TBX5⁺) cells were virtually absent until day 8 (Fig. 3c; Supplementary Fig. 4a). eGFP⁺ cells maintained ISL1 expression while progressively upregulating the CM marker cTnT from day 5 (Fig. 3d). With addition of RA, eGFP⁺ cells were first detected at day 3 and both mCherry⁺ and eGFP⁺/mCherry⁺ populations emerged from day 4 (Fig. 3c; Supplementary Fig. 4b). In this condition, we observed a rapid downregulation of ISL1 and upregulation of cTnT in all populations (Fig. 3d; Supplementary Fig. 4c, d), again suggestive of a FHF-like fate.

**RA signaling patterns the cardiogenic mesoderm**
To gain further insights into the formation of cardiac progenitors in response to limited RA exposure, we performed single-cell RNA sequencing (scRNA-seq) on early mesodermal cells at day 1.5 (right

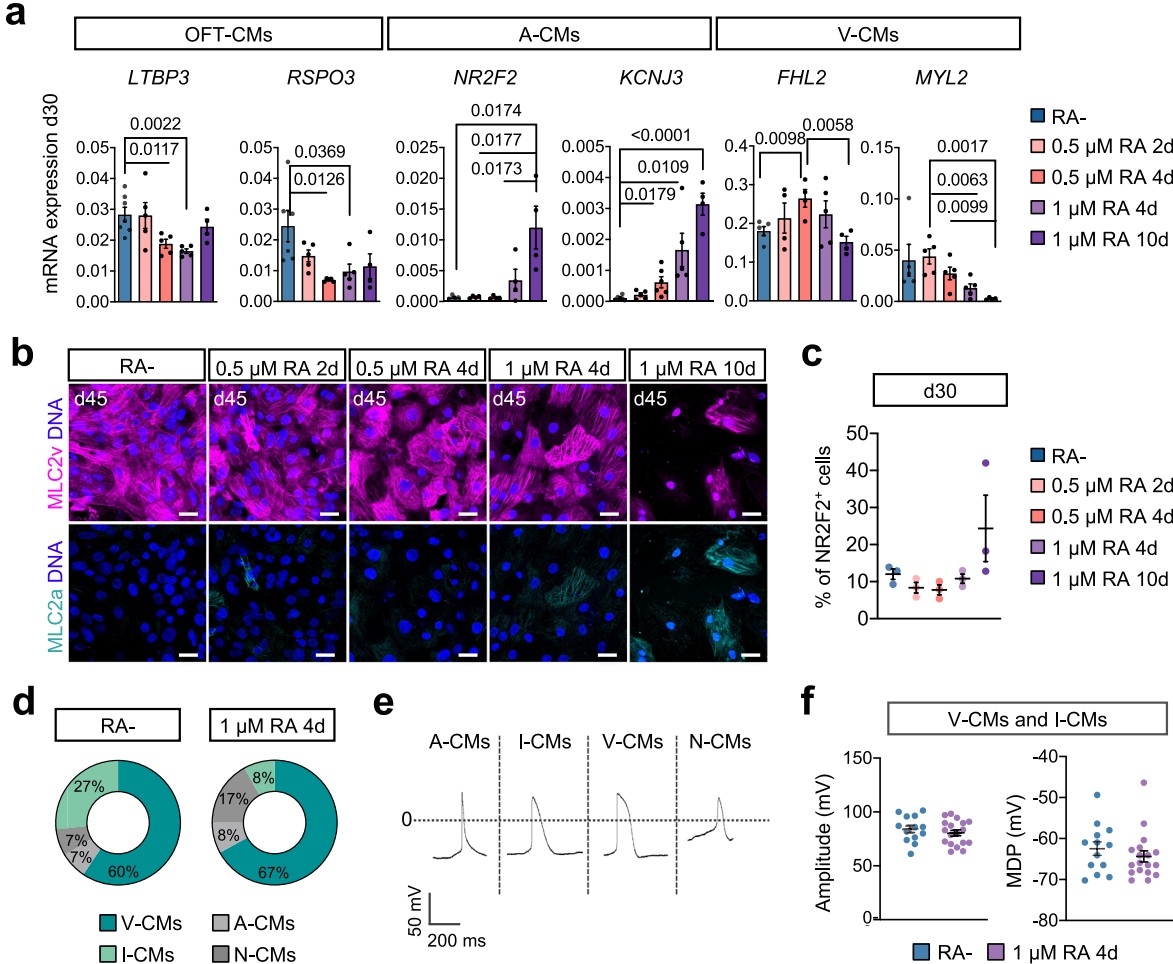

**Fig. 2 | RA signaling modulation guides differentiation into distinct cardiomyocyte subtypes. a** mRNA expression of markers of outflow tract CMs (OFT-CMs), atrial CMs (A-CMs) and ventricular CMs (V-CMs) relative to *TNNT2* and *GAPDH* at day 30 of the indicated differentiation conditions. Data are mean ± SEM. *n* = 4 independent experiments except for: RA− *LTBP3* *n* = 7; *MYL2* and *FHL2* *n* = 5; *RSPO3* and *KCNJ3* *n* = 6; RA+ except for RA+ (1 μM 10d) *LTBP3, RSPO3, MYL2, KCNJ3* *n* = 5; RA+ (1 μM 4d) *FHL2* *n* = 5; RA+ (0.5 μM 4d) *n* = 6. Exact *p*-values of unpaired two-tailed *t*-test are shown. **b** Representative immunofluorescence images after staining for MLC2v (magenta) and MLC2a (turquoise) at day 45 of indicated differentiations conditions. Nuclei were counterstained with Hoechst-33258 (blue). Scale bar = 25 μm. **c** Percentage of cells stained positive for NR2F2 at day 30 of indicated differentiation conditions as determined by flow cytometry. Data are mean ± SEM. *n* = 3 independent experiments. **d** Percentage of cardiomyocyte subtypes at day 30 of differentiation with 1 μM RA for 4d or without RA (RA−) as defined by the ratio of action potential duration at 50% and 90% repolarization determined by whole-cell voltage clamp recording. I-CMs intermediate ventricular cardiomyocytes, V-CMs ventricular cardiomyocytes, A-CMs atrial cardiomyocytes, N-CMs nodal cardiomyocytes. **e** Representative action potential traces of indicated cardiomyocyte subtypes at day 30. **f** Amplitude (left) and maximum diastolic potential (MDP, right) of action potentials in cardiomyocytes at day 30 of differentiation with 1 μM RA for 4d or without RA (RA−). Data are mean ± SEM. For 1 μM RA for 4d *n* = 18 cells; for RA− *n* = 13 cells examined over two independent experiments. Source data are provided as a Source Data file.

before the addition of RA) as well as on individual cardiovascular progenitor populations at day 4.5 (Fig. 3e). Using flow cytometry-based sorting, we isolated eGFP⁺ (NKX2.5) cells obtained in the absence of RA and the three populations with differential eGFP (NKX2.5) and mCherry (TBX5) expression emerging in the presence of RA (Fig. 3e; Supplementary Fig. 5a). Low dimensionality embedding (UMAP) of all cells merged resulted in the formation of five distinct groups: early mesodermal cells (day 1.5), endothelial/endocardial progenitor cells (E-PCs), endodermal cells, and cardiovascular progenitor cells derived in the absence (CPC-RA−) or presence of RA (CPC-RA+) (Fig. 3f; Supplementary Fig. 5b; Supplementary Data 2).

Cells at day 1.5 represented a homogenous cell population expressing mesodermal genes such as *TBXT* and *MESP1* (Supplementary Fig. 5c). The expression of genes such as *EOMES, GSC*, and *MIXL1* suggested that these cells match a mid/anterior primitive streak fate[8] (Supplementary Fig. 5d). Integration of this data with a recently published scRNA-seq dataset of a gastrulating mouse embryo confirmed

that these cells correspond to mid/late primitive steak cells, which give rise to LV and RV CMs in the mice[4] as opposed to the no-bud/early bud stage streak cells, which generate atrial CMs (Supplementary Fig. 5e). In line with this, the expression profile of RA receptors (RXRs) and co-receptors (RARs) was comparable between cells at day 1.5 and the mid/late primitive streak cells in the mouse (Supplementary Fig. 6a). Moreover, we observed high expression of the cytochrome member *CYP26A1*, which has been reported to mark a subpopulation of the cardiogenic mesoderm prone to giving rise to ventricular cardiomyocytes[11] (Supplementary Fig. 6b).

The cardiovascular progenitors derived in the absence of RA (CPC-RA−) separated into two main clusters based on their proliferative state (cluster 3, non-proliferative; cluster 13, proliferative; Fig. 3f; Supplementary Fig. 6c, d), both showing expression of aSHF genes such as *ISL1, FGF10*, and *BMP4* (Supplementary Fig. 6c). Two small additional clusters were shared with the RA+ condition: cluster 12 was characterized by high expression of *PDGFRA*, indicating

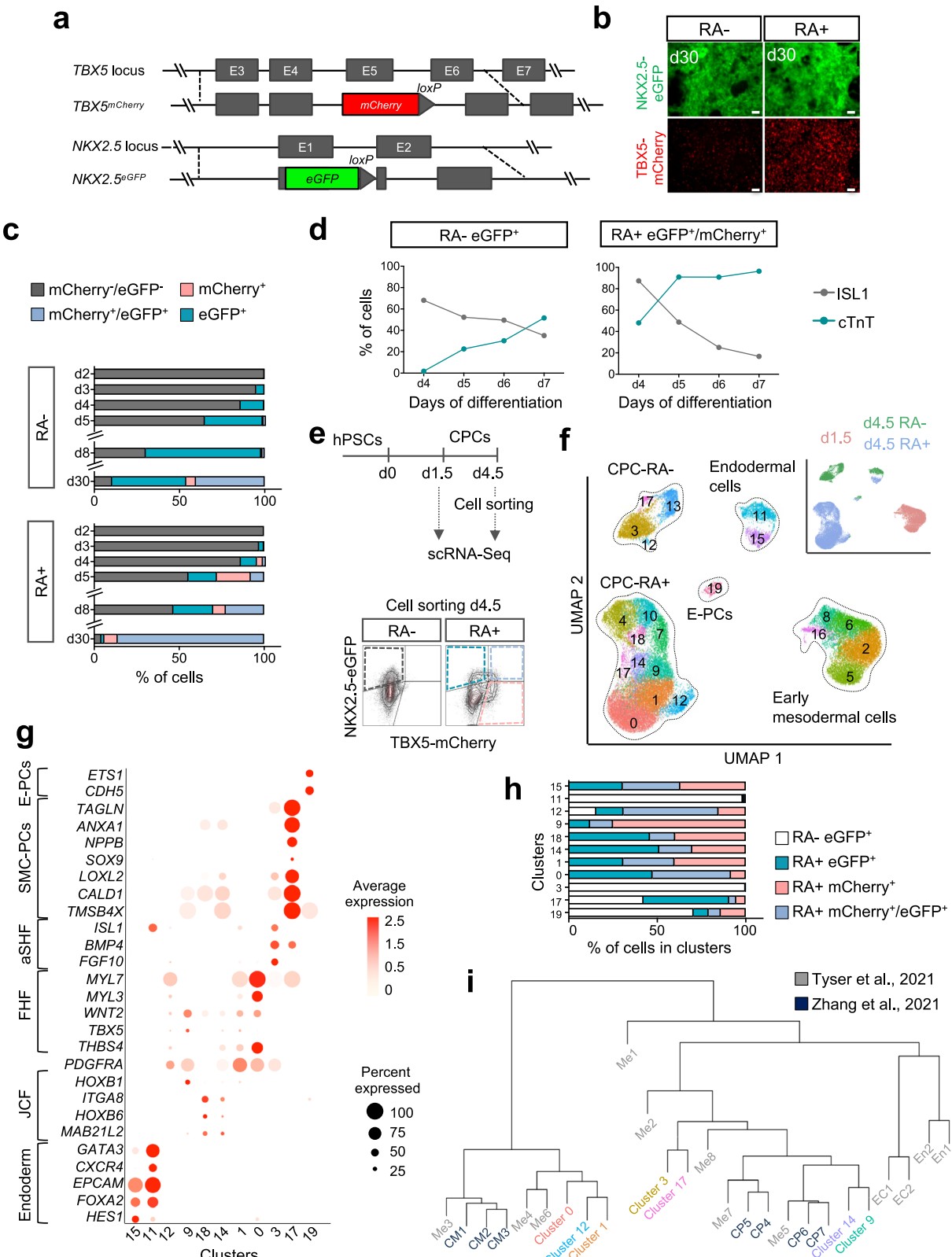

undifferentiated cardiovascular progenitor cells, and cluster 17 was enriched in genes related to smooth muscle cells (SMCs) such as *TAGLN*, *NPPB*, *LOXL2*, *TMSB4X*, and *CALD1* (Fig. 3f–h).

In the presence of RA, the cardiovascular progenitor cells (CPC-RA+) formed several clusters (Fig. 3f). Clusters 0 and 1 showed expression of *THBS4* as well as cardiac sarcomere protein genes such as *MYL3*, and *MYL7* most likely resembling progenitors

of the FHF that are in the process of converting to CMs (Fig. 3g). Clusters 4, 7, and 10 represented the proliferative equivalents of clusters 0, and 1 (Supplementary Fig. 6c, d). Notably, among the CPC-RA+ we observed two small additional clusters (cluster 14 and its proliferative counterpart cluster 18) that showed a gene expression profile (*MAB21L2, HOXB6, HOXB5, BNC2*) resembling that of the recently described JCF, which is a distinct subset of the

**Fig. 3 | RA signaling patterns the cardiogenic mesoderm. a** Schematic representation of the genetic modifications in the TBX5^mCherry/NKX2.5^eGFP hESC reporter cell line (ES03 TN). **b** Representative live images of cells expressing eGFP (NKX2.5; green) Cherry (TBX5; red) at day 30 of differentiation with (RA+) or without RA (RA−). Scale bar = 50 μm. **c** Percentage of cells expressing mCherry (TBX5) and eGFP (NKX2.5) at the indicated days of differentiation with (RA+) or without RA (RA−) as determined by live flow cytometry. Data are mean; for day 2 *n* = 2 independent samples; days 3 and 4 *n* = 5; day 5 *n* = 7; days 8 and 30 *n* = 3 examined over seven independent experiments. **d** Quantification of flow cytometry time course analysis of cells expressing ISL1 and cTnT within mCherry⁺/eGFP⁺ (TBX5⁺/NKX2.5⁺) population from RA+ differentiation; and eGFP⁺ (NKX2.5⁺) population from RA− differentiation. Data are mean; *n* = 2 independent experiments. **e** Top: Graphic representation of the experimental design applied for scRNA-Seq analysis at days 1.5 and 4.5. Bottom: Representative flow cytometry plots showing the gating strategy for sorting cells at day 4.5. Cells differentiated without RA (RA−) were isolated based on the expression of eGFP (NKX2.5); cells differentiated with RA (RA+) were isolated based on expression of eGFP (NKX2.5), mCherry (TBX5), or both. **f** UMAP clustering of single cells captured at days 1.5 and 4.5; main cell types are annotated. Inset: UMAP plot showing the contribution of the indicated samples. CPCs cardiovascular progenitor cells, E-PCs endothelial/endocardial progenitor cells. **g** Dot-plot showing expression level of selected differentially expressed genes for clusters identified as 15, 11−endoderm; 12−early CPCs; 1, 0−first heart field (FHF); 3−anterior second heart field (aSHF); 17−smooth muscle cell progenitor cells (SMC-PCs); 19−E-PCs; 9, 18, 14−juxta-cardiac field (JCF). **h** Contribution of cells from each population to the indicated clusters relative to total number of cells in the cluster normalized for total number of cells in each population. **i** Dendrogram showing hierarchical clustering of the averaged, corrected, normalized expression values per clusters after integration of indicated scRNA-seq clusters with scRNA-seq datasets of mouse early heart development[5,6]. Source data are provided as a Source Data file.

FHF in mice[6] (Fig. 3g). Cluster 9 was closely related to cluster 14 (Fig. 3i) and was characterized by expression of *TBX5*, as well as posterior transcripts such as *WNT2* and *HOXB1* while also expressing low levels of the JCF marker *MAB21L2* (Fig. 3g).

The cluster identities were further evaluated by integrating the cells from the non-proliferative CPC clusters (clusters 0, 1, 3, 9, 12, 14, and 17) with the scRNA-seq datasets of Tyser et al.[6] as well as Zhang et al.[5] covering stages of mouse embryonic heart development that include JCF emergence[6]. This analysis confirmed the FHF- and aSHF-like characteristic of the above-mentioned clusters and showed that clusters 14 and 9 are closely related with the mouse JCF cluster me5 described by Tyser et al (Fig. 3i). Notably, these cells were also very similar to the cardiac progenitor clusters CP6 and CP7 described by Zhang et al., which are related to the late extraembryonic mesoderm in mice and have a comparable transcriptional signature with the JCF[5]. This suggests that cluster 9 as well as cluster 14 both represent JCF-like cells (Fig. 3i). Cluster 9 was mainly derived from TBX5⁺/NKX2.5⁻ CPCs (~75%; Fig. 3h), which is in line with findings in the mouse where the JCF arises from a TBX5⁺/NKX2.5⁻ progenitor population[6]. Cluster 14 was mainly derived from TBX5⁺/NKX2.5⁺ cells and expressed low levels of *NKX2.5* (Supplementary Fig. 6e). This suggests that cluster 9 likely represents an early JCF population and cluster 14 a further committed JCF population that, due to the cardiogenic differentiation conditions, has started to express *NKX2.5* and acquire a cardiomyogenic fate. Progenitors residing in the JCF in the mouse have the potential to contribute to the cardiomyocytic as well as the epicardial lineage[6]. Our data suggests that a similar population could exist during human cardiogenesis.

### ITGA8 allows isolation of human JCF-like progenitor cells
To further explore the potential of this putative JCF cell population, we aimed at identifying a surface marker allowing the specific isolation of these cells. Comparative differential gene expression analysis between the mouse dataset of Tyser et al.[6] and ours revealed *TNC*, *AHNAK*, and *ITGA8* as promising candidates showing co-expression with other key JCF markers (*MAB21L2*, *HOXB5*, *HOXB6*, *HAND1*, *BNC2*; Supplementary Data 3)[6]. Among these, *ITGA8* showed the most restricted expression pattern (Fig. 4a, b; Supplementary Fig. 7a, b). Flow cytometry analysis confirmed the presence of a small population of ITGA8⁺ cells from day 4 to day 6 (Fig. 4c, d). This population initially expressed only TBX5 (mCherry) but quickly upregulated NKX2.5 (eGFP) (Fig. 4e, f), which is in line with observations in the murine JCF[6]. Cells sorted based on the presence of ITGA8 showed significantly higher expression of JCF-specific markers compared to unsorted cells (*MAB21L2*, *BNC2*, *HAND1*; Fig. 4g; Supplementary Fig. 7c). After cultivating ITGA8⁺ sorted cells in myocardial differentiation conditions, we obtained cTnT⁺ CMs (Fig. 4h, i). On the other hand, when exposed to epicardial differentiation conditions, the cells formed a tight epithelial layer stained positive for the epithelial markers ZO1 and CK18 as well as the epicardial markers TCF21 and WT1 (Fig. 4h, j).

To assess the fate potential of the ITGA8⁺ population at a clonal level, we took advantage of hiPSCs constitutively expressing *eGFP* and sorted ITGA8⁺ and ITGA8⁻ cells emerging at day 5 of differentiation in the presence of RA (0.5 μM, 4d). We plated these cells at very low density with equivalent, unsorted day 5 CPCs generated from unlabeled hiPSCs (~5 eGFP⁺ cells with 20,000 eGFP⁻ cells/well of a 12 well chamber slide) and differentiated them for 10 days in culture conditions that allow for simultaneous generation of epicardial cells and CMs (Fig. 4k). As compared to the eGFP⁺/ITGA8⁻ fraction, eGFP⁺/ITGA8⁺ cells showed a higher propensity to give rise to epicardial cells, while maintaining the ability for CM differentiation (Fig. 4l, m; Supplementary Data 4). While the vast majority of eGFP⁺/ITGA8⁺ clones differentiated into one lineage, around 10% of them gave rise to both epicardial cells and CMs (Fig. 4m, Supplementary Fig. 7d, Supplementary Data 4). This data provides further proof that a bipotent JCF progenitor pool giving rise to CMs and epicardial cells also exists during human cardiogenesis and can be enriched using ITGA8 as a membrane marker.

### CPC-RA+ and RA− give rise to distinct cardiac cell populations
To characterize the descendants of the cardiovascular progenitors emerging in our system, cells differentiated with and without RA were sorted at day 4.5 based on eGFP (NKX2.5) and mCherry (TBX5) expression and further cultured as 3D aggregates to allow more physiological conditions for differentiation. Aggregates were then collected at day 30 for scRNA-seq (Fig. 5a). Cells from day 30 formed in UMAP three distinct groups: endoderm-derived cells, non-myocytic cardiac cells, and CMs (Fig. 5b, c; Supplementary Fig. 8a–c; Supplementary Data 5).

The CMs separated into four main groups: ventricular-like CMs (clusters 0, 1, 2, 3, 4) OFT-like CMs (cluster 6), AVC-like CMs (cluster 13), and proliferating CMs (clusters 8, 10, 12) (Fig. 5d; Supplementary Fig. 8b, c).

The transcriptional profiles of ventricular-like CMs derived from the CPC-RA+ (clusters 0, 2, 3 and 4) and CPC-RA− (cluster 1) were overall quite similar, in line with data from human in vivo embryonic hearts showing that transcriptional differences between ventricular CMs located in the LV vs. RV are subtle[25,26] (Fig. 5d, e; Supplementary Data 6). Among the genes that show a preferential expression in either the LV or the RV in vivo[25], we observed higher levels for *TBX5* and *GJA1* in the LV-like clusters (0, 2, 3, 4) as compared to the RV-like cluster (1), while the expression levels for *VCAN* and *GPRIN3* was higher in the RV-like cluster (1) as compared to the LV-like clusters (0, 2, 3, 4) (Supplementary Fig. 9a, b). Notably, the expression level of *HAND1* was highest in the OFT-CM cluster 6 (Supplementary Fig. 9a). This is in line with reports in the literature showing that *HAND1* is expressed not only in LV CMs, but also in OFT and AVC CMs[1,17]. OFT-like CMs (cluster 6) were almost exclusively found as derivatives of CPC-RA− (Fig. 5e) and expressed markers of human conoventricular CMs, such as *BMP2* and

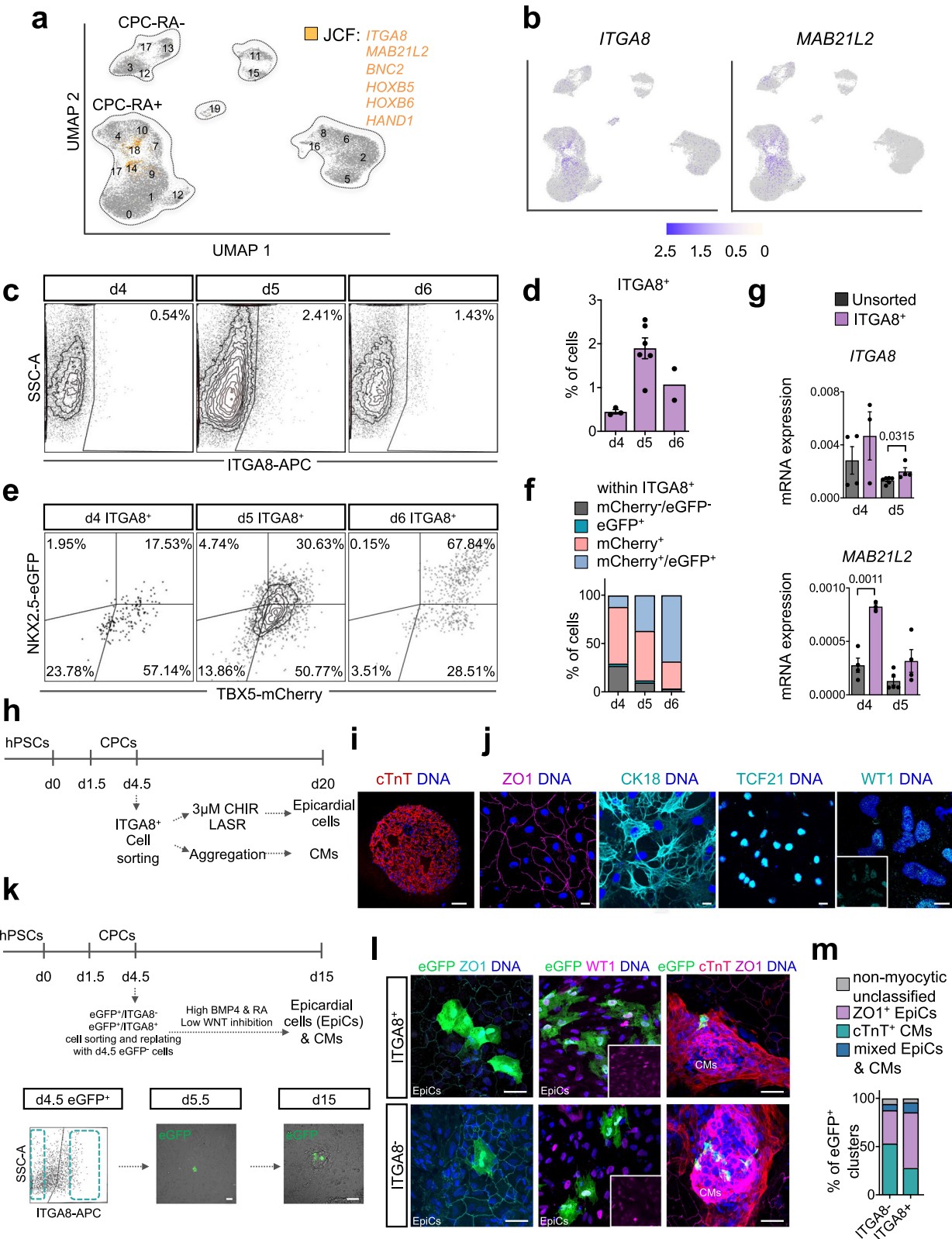

RSPO3[18], as well as SMC markers (e.g., *ACTA2*, *TAGLN*, *COL1A2*) and only low expression levels of *MYL2*, according with the transcriptional profile of OFT-CMs in mice[27] (Fig. 5d; Supplementary Fig. 9c). Moreover, they did not express *TBX5*, which is consistent with data from human fetal tissue showing absence of TBX5 in OFT structures[28] (Supplementary Fig. 8c). We also identified a small cluster 14 which was enriched in *CPNE5* and *IGFBP5* transcripts, and could correspond to a

transitional population of nodal CMs occurring during in vitro differentiation of hiPSC[29,30] (Fig. 5d, Supplementary Fig. 9c). This cluster was almost exclusively derived from the CPC-RA+ line with the increased number of nodal cells upon RA exposure observed in the electrophysiology assay (Fig. 2d).

The identity of the OFT-like and ventricular-like CMs was confirmed on a transcriptome-wide level using enrichment analysis for

**Fig. 4 | Human JCF-like progenitors expressing ITGA8 differentiate into epicardial and myocardial cells. a** Feature plot showing cells co-expressing key JCF markers in orange. **b** Feature plot showing the expression of *ITGA8* and *MAB21L2*. **c** Representative plots and **d** quantification of live flow cytometry analysis of cells expressing ITGA8 (APC) during differentiation with RA. Data are mean ± SEM; day 4 *n* = 3 independent samples; day 5 *n* = 6; day 6 data are mean n = 2 over 6 independent experiments. **e** Representative plots and **f** quantification of live flow cytometry analysis of cells expressing mCherry (TBX5) and eGFP (NKX2.5) within the ITGA8⁺ (APC). Data are mean; day 4 *n* = 3 independent samples; day 5 *n* = 6; day 6 *n* = 2 over 6 independent experiments. **g** mRNA expression of *ITGA8* and *MAB21L2* relative to *GAPDH* in cells flow cytometry-based sorted for ITGA8 (APC) and unsorted cells. Data are mean ± SEM; *n* = 4 independent experiments except for: *ITGA8* sorted day 4 *n* = 3; sorted day 5 *n* = 6; *MAB21L2* sorted day 4 *n* = 3, unsorted day 5 *n* = 5. Exact *p*-values of unpaired two-tailed *t*-test are shown. **h** Schematic representation of the experimental timeline for flow cytometry-based sorting of ITGA8⁺ cells followed by epicardial and myocardial differentiation. **i, j** Representative images of cells stained for **i** cTnT (red) and **j** ZO1 (magenta), CK18, TCF21, WT1 (turquoise), 15 days after replating/reaggregation of ITGA8⁺ sorted progenitors at day 4.5 of differentiation examined over 3 independent experiments. For **g**, scale bar = 100 μm. For **h**, scale bar = 20 μm. Inset shows cells stained for WT1 (turquoise). Nuclei were counter-stained with Hoechst-33258 (blue). **k** Schematic representation of the experimental timeline for flow cytometry-based sorting followed by simultaneous epicardial and myocardial differentiation. Scale bar = 50 μm. **l** Representative images of eGFP⁺ cells (green) stained for ZO1 (turquoise/magenta), WT1 (magenta) and cTnT (red) 10 days after replating of eGFP⁺/ITGA8⁺ or eGFP⁺/ITGA8⁻ sorted progenitors. Scale bar = 50 μm. Inset shows cells stained for WT1 (magenta). EpiCs epicardial cells, CMs cardiomyocytes. **m** Percentage of eGFP⁺ cell clusters stained for epicardial marker ZO1, myocardial marker cTnT, both or none of those markers 10 days after replating eGFP⁺/ITGA8⁺ or eGFP⁺/ITGA8⁻ cells. For eGFP⁺/ITGA8⁺ cells *n* = 87 cell clusters; for eGFP⁺/ITGA8⁻ cells *n* = 130 over 3 independent experiments. EpiCs epicardial cells, CMs cardiomyocytes. Source data are provided as a Source Data file.

gene signature scores derived from human and mouse in vivo heart samples[25,31] (Fig. 5f).

Subclustering of the non-myocytic cells, which had in common a high expression level of *VIM* (Fig. 5c, h), revealed the presence of endothelial/endocardial-like cells (*APLN*, *CDH5*), smooth muscle-like cells (*TAGLN*, *ACTA2*, *CALD1*), fibroblast-like cells (*COL1A1*, *THY1*, *LUM*), myofibroblasts (*MYOZ2*, *MYL2*), and valvular-like cells[32–34] (*SOX9*, *POSTN*; Fig. 5g–i; Supplementary Data 7). In addition, we detected cells expressing genes associated with endothelial to mesenchymal transition (EndoMT), such as *ZEB2* and *LUM* (Fig. 5i). While endothelial/endocardial-like cells and smooth muscle-like cells were found as derivatives of all populations, valvular-like cells were almost exclusively derived from aSHF-like progenitors (CPC-RA−; Fig. 5j). This is in agreement with previous reports in the mouse showing that valve formation mainly relies on contribution from the aSHF[35,36].

Overall, these findings confirmed that the differences observed at the cardiovascular progenitor state translated into the formation of distinct progeny reflecting FHF-like (CPC-RA+) vs aSHF-like (CPC-RA−) fate potential. Notably, the clusters at day 30 of differentiation arising from the sorted populations (eGFP⁺, mCherry⁺, double mCherry⁺/eGFP⁺) of the CPC-RA+ (0, 2, 3, 4) where highly similar in their transcription profiles (Fig. 5e; Supplementary Data 5), suggesting that the different subpopulations of the CPC-RA+ sorted at day 4.5 (eGFP⁺, mCherry⁺, double mCherry⁺/eGFP⁺) do not represent distinct progenitor pools but are more likely reflecting different stages of progression on the same differentiation trajectory. In addition, we found that the differentiation efficiency into the myocytic lineages, in both RA− and RA+ conditions, was similar between sorted (RA+:-87−96% *TNNT2*⁺ CMs, -4−12% *FN1*⁺ non-myocytic cell types; RA−: 84.7%; and 15.3%, respectively) and unsorted cells (RA+:-91−93% of cTnT⁺ CMs, -5−10% of FN1⁺ non-myocytic cell types; RA−: -83−92%; and -13−17%, respectively), indicating that the negative (mCherry⁻/eGFP⁻) population present at day 4.5 does not seem to significantly impact the differentiation outcome (Supplementary Fig. 10a–c). Taken together, these findings suggest that the differentiation protocol described in this manuscript can be utilized independently of the fluorescent marking of the hESC-line, providing a substantial advantage over previously described methods to isolate heart-field specific populations[37].

### Human CPCs contribute to distinct structures in mouse hearts

To further investigate the lineage commitment and functionality of in vitro derived cardiovascular progenitors, we injected CPC-RA+ or CPC-RA− into the heart region of developing mouse embryos at the cardiac crescent stage, where the FHF and SHF are located in close proximity[38] before initiation of visible contraction. These embryos were then subjected to whole-embryo ex vivo culture for 24 or 48 h (Fig. 6a). The embryos' heart developed normally, including beginning of contraction, formation of the linear heart tube, its looping and onset

of chamber formation (Fig. 6b–d). Since separation of the heart compartments in these early stages of heart development is still incomplete, we defined rather broadly delineated cardiac areas (OFT/RV, RV/LV, LV, IFT) to evaluate the regional contribution of the injected CPCs, as shown in Supplementary Movie 1.

The injected human cardiovascular progenitors, identified by human nuclear antigen (HNA) immunofluorescence staining, from both conditions (CPC-RA+ and CPC-RA−) successfully integrated into the host myocardium (Fig. 6e–g; Supplementary Fig. 11a; Supplementary Data 8). Immunofluorescence analysis in whole-mounted cleared embryos and in sectioned embryos showed that CPC-RA+ contributed mainly to the LV, with minor additions to the inflow-tract (IFT; Fig. 6f, g). No contribution to the OFT/RV region was observed, aside from one single cell in one out of eight embryos (Fig. 6f, g). Conversely, CPC-RA− contributed mainly to the OFT/RV, and we did not find any of these cells in the LV (Fig. 6f, g). Immunofluorescence staining for cTnT indicated that cardiovascular progenitor cells from both conditions differentiated into CMs, with sarcomere organization of early myofibrils similar to the host CMs (Fig. 6h, Supplementary Fig. 11b). Notably, the percentage of human non-myocytic cells (HNA⁺/cTnT⁻) within the cardiac region was significantly higher after injection of CPC-RA− as compared to CPC-RA+ (Fig. 6i; Supplementary Data 8), most likely reflecting the broader lineage potential of the CPC-RA− population.

This data not only confirms our in vitro findings on the identity of CPC-RA+ and CPC-RA− at day 4.5 but also highlights that the cells at this stage are already committed to contributing to their respective cardiac structures.

### CPC-RA+ and RA− show disparate phenotypes in HLHS disease model

To evaluate the utility of our fate-committed cardiac progenitors in studying human disease, we employed them as an in vitro model of HLHS, which is characterized by underdevelopment of the LV and the LVOT and is one of the most severe congenital heart defects[39]. We previously reported that HLHS patient-specific hiPSCs show defects in the differentiation of early cardiac progenitors resulting in abnormal cardiomyocyte subtype lineage specification and maturation[40]. However, the differentiation protocol used in that study had not allowed the investigation of heart field-specific differences.

Differentiating two of the previously reported HLHS hiPSC lines (see Methods for details on line ID and genotype) into FHF-like (CPC-RA+) and aSHF-like progenitors (CPC-RA−), we observed, in both conditions, a downregulation of the key CPC marker *NKX2.5* (Fig. 7a), which is in agreement with our previous findings[40]. This was paralleled by a substantial reduction of *TNNT2* levels in CPC-RA+, suggestive of impaired myocytic commitment, and a significant dysregulation of the aSHF-marker *FGF10* in the CPC-RA−, together indicating that altered CPC-fate acquisition in HLHS seems to affect both, the FHF-like (CPC-

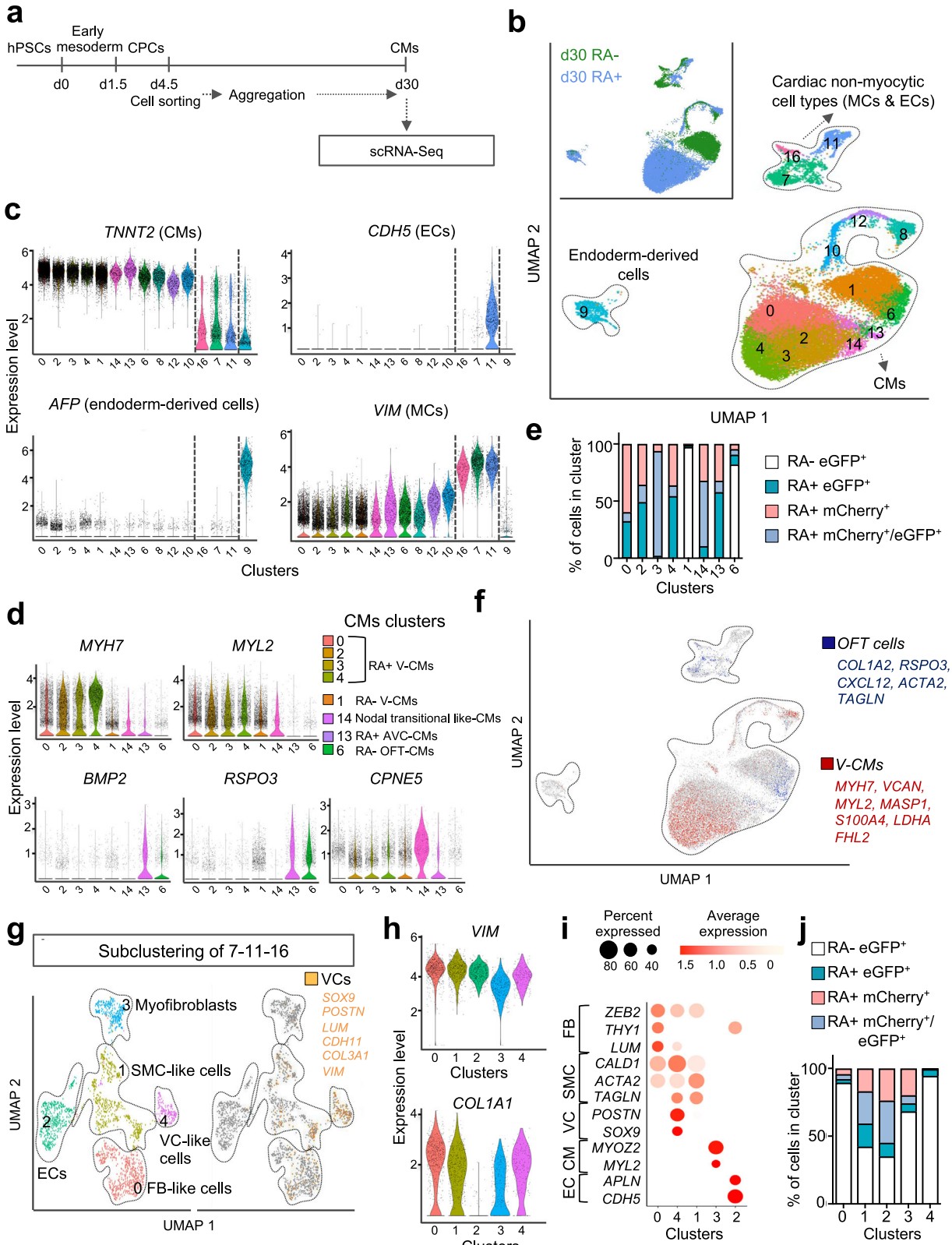

RA+) as well as the aSHF-like (CPC-RA−) population (Fig. 7a). During further differentiation, HLHS patient-specific hiPSCs gave rise to CMs in both conditions as shown by immunofluorescence analysis of cTNT and qPCR for *TNNT2*, although at lower efficiency (in particular in the CPC-RA− conditions) as compared to the control (Fig. 7b, Supplementary Fig. 11c). In addition, these CMs showed much less, if any, expression of MLC2v, suggesting impaired attainment of ventricular

identity (Fig. 7c). qPCR analysis confirmed a reduction of ventricular cardiomyocyte transcripts (*MYH7*, *MYL2*; Fig. 7d) and revealed increased *MYL7/MYL2* and *TNNI3/TNNI1* isoform ratio as indicator of cardiac immaturity (Fig. 7d), corroborating our previous findings on maturation defect in HLHS patient-specific CMs[40]. Notably, HLHS CMs arising from the CPC-RA+ also displayed lower levels of the LV-specific marker *FHL2* (Fig. 7d). At the same time, they exhibited higher

**Fig. 5 | CPC-RA- and CPC-RA+ give rise to distinct cardiac cell types. a** Graphic representation of the experimental design for scRNA-Seq analysis at day 30. At day 4.5, cells differentiated without RA (RA−) were sorted based on expression of eGFP (NKX2.5); cells differentiated with RA were sorted based on expression of eGFP (NKX2.5), mCherry (TBX5), or both. Sorted cells were reaggregated in 3D and analyzed at day 30. **b** UMAP clustering of single cells captured at day 30; main cell types are annotated. Inset: UMAP plot showing the contribution of the indicated samples. CMs cardiomyocytes, MCs mesenchymal cells, ECs endothelial cells. **c** Violin plots showing the expression level of *TNNT2*, *CDH5*, *AFP*, and *VIM* in all clusters from **b**. **d** Violin plots showing the expression level of *MYH7*, *MYL2*, *BMP2*, *RSPO3*, and *CPNE5* in CM clusters from **b**. **e** Contribution of day 30 cells derived from each sorted CPC population to the indicated CM clusters relative to all cells in the cluster: 0, 1, 2, 3, 4−ventricular cardiomyocytes (V-CMs); 6−outflow tract cardiomyocytes (OFT-CMs); 13−atrioventricular canal cardiomyocytes (AVC-CMs),

14−Nodal transitional-like CMs. Data are normalized to the total number of cells in each population. **f** Feature plot showing in red cells co-expressing top differentially expressed genes defining V-CMs clusters of Cui et al.[25] and Asp et al.[26] and confirmed with specific expression in V-CMs clusters of Asp et al. (online search tool); in blue cells co-expressing genes defining OFT clusters of Li et al.[31] and Asp et al.[26] and confirmed with specific expression in OFT cell clusters of Asp et al.[26] (online search tool). **g** Left: UMAP subclustering of cardiac non-myocytic clusters (7, 11, 16) at day 30; right: feature plot showing in yellow cells co-expressing key markers of valvular cells (VCs). **h** Violin plots showing the expression level of *VIM* and *COL1A1* in clusters from **g**. **i** Dot-plot showing the expression level of selected differentially expressed genes for the subclusters shown in **g**. **j** Contribution of day 30 cells derived from each sorted cell population to subclusters shown in **g** relative to all cells in the cluster.

expression of genes related to OFT-CMs (*RSPO3*) rendering their transcriptional profile similar to the patient-specific CMs arising from the CPC-RA−, indicative of a failed commitment to a LV CM fate (Supplementary Fig. 11c). We also observed dysregulation of OFT-CM transcripts (*RSPO3*) in the CMs derived from the CPC-RA−, which could likely be a consequence of the altered *FGF10* level in the progenitors, since this gene is crucially required in the aSHF for proper OFT formation in mice[41–43]. The fact that both progenitor pools (CPC-RA+ and CPC-RA−) and derived CMs appear affected is consistent with observations in vivo showing that CMs from the LV as well as the RV of HLHS patient share similar transcriptional defects[40,44].

To further evaluate the fate commitment of HLHS-derived CPCs, we injected the cells into the heart region of developing mouse embryos at the cardiac crescent stage and subjected those to whole-embryo ex vivo culture for 24 or 48 h. Immunofluorescence analysis indicated that HLHS-derived CPCs still contributed preferentially to their respective heart compartments (CPC-RA+ to the LV, CPC-RA− to the RV/OFT). However, their segregation to specific cardiac areas was less defined as compared to the control, with cells from both conditions being also present in other heart compartments (Fig. 7e–g; Supplementary Fig. 11d). In addition, we observed a significant reduction in the percentage of human HNA+/cTnT+ cells after injection of the CPC-RA+ as compared to the control (43% vs. 78%, *p* = 0.0087), suggesting impaired CM differentiation in line with the findings from the 2D culture (Figs. 6i, 7h; Supplementary Fig. 11e; Supplementary Data 8). Interestingly, this was not the case in the embryos injected with the HLHS-derived CPC-RA−, where we detected a similar proportion of human HNA+/cTnT+ cells as compared to the control (29% vs. 34%) (Figs. 6i, 7h; Supplementary Fig. 11e; Supplementary Data 8). As far as the limited number of embryos in each group allows, this suggests that the differentiation defect seen in the 2D system for this population can be, at least partially, compensated in the ex vivo setting in line with observations in patients where RV cardiomyocytes, although altered on the transcriptional level, form a heart chamber[40,44].

Taken together, these findings suggest that HLHS patient-specific hiPSCs fail to acquire a FHF-like fate at the progenitor level, resulting in perturbed CMs without proper LV CM identity. Notably, they also show a defect in aSHF progenitor differentiation resulting in CMs with significant dysregulation of ventricular and OFT CM markers, providing a potential link to the LVOT defects observed in HLHS patients. This highlights how heart-field specific progenitors can increase the resolution of in vitro disease models and provide additional insights in human congenital heart disease.

## Discussion
Due to the limited accessibility of early human embryonic tissue samples, our knowledge on the emergence of human cardiovascular progenitors and their contributions to the developing heart relies largely on findings from model organisms and in vitro model systems.

In the current study, we describe a versatile in vitro differentiation platform that allows the generation and isolation of various cardiac progenitor pools from hPSCs by modulating RA-mediated signaling. RA signaling is one of the key signaling pathways in the developing embryo and is critically required for the organization of the trunk and the organogenesis of various tissues of all three germ layers, including the hindbrain, the eye, the spinal cord, the pancreas, and the heart[45,46]. It is first detected at late primitive streak stages in the paraxial mesoderm[47] and plays an important role in defining the posterior boundary of heart fields[46,48] as well as the proper specification of the pSHF that gives rise to the IFT[10].

The latter has been exploited previously showing that addition of RA during in vitro differentiation of hPSCs towards CMs generates atrial CMs that are suitable for use in drug screening[11,13,14]. In our study we did not observe an upregulation of atrial CM markers after exposure to low dosages of RA. Instead, the majority of cardiovascular progenitors found in our differentiation conditions had acquired a FHF-like fate rapidly giving rise to ventricular-like CMs. Upon exposure of higher dosage of RA for longer periods of time, we did detect a slight increase of atrial cardiomyocytes, however not comparable to protocols targeted at generating atrial CMs[49]. Differently to our approach, the aforementioned studies have used an alternative cardiac differentiation protocol[13] most likely resulting in distinct mesodermal progenitor subsets. There is an increasing body of evidence, that the commitment of mesodermal progenitors to form specific cardiac structures is already laid out during primitive streak formation[4,50]. Mesodermal progenitors that leave the streak early (at mid/late stage) −to which the mesodermal progenitors generated with our protocol show resemblance− preferentially give rise to ventricular cardiomyocytes, while later cells (no-bud/early bud stage) generate atrial cardiomyocytes[4]. In addition, the dosage of RA applied in our protocol is lower, and it has been reported previously that low doses of RA during hPSC differentiation improve differentiation efficiency but do not affect cardiac subtype specification[8,12].

Taken together this shows that timing and dosage of RA as well as the type of mesodermal progenitors exposed results in distinct differentiation outcomes highlighting the pleiotropic roles of RA during development[51].

In addition to the "classical" first and second heart field progenitors, we identified a subset of cells that resembled progenitors of the recently discovered JCF, which contribute to the epicardium as well as myocardium in mice[5,6]. We uncovered that the surface protein ITGA8, which is expressed during development on epicardial cells in the mouse and the chicken[52,53], can be used to enrich these cells using live cell sorting. Sorted JCF-like progenitors readily differentiated into epicardial as well as cardiomyocytic progeny, proposing the presence of a similar progenitor population during human cardiogenesis. Remarkably, this population was only present upon exposure to RA, suggesting that RA-mediated signaling might be critically involved in the generation of the JCF. Whether the JCF-like cells arising in our

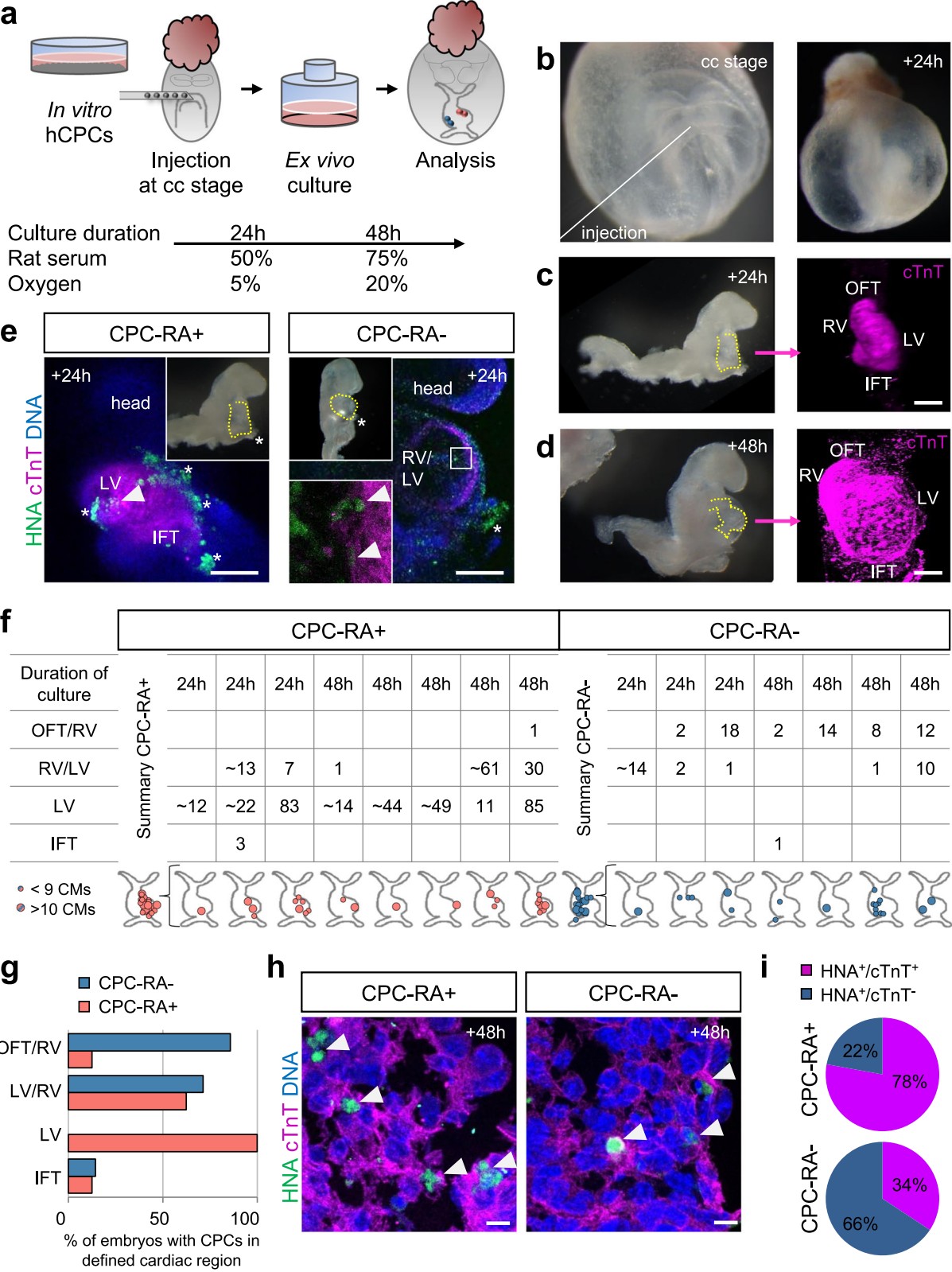

differentiation conditions represent the entire JCF population or only a subset remains elusive. Further studies on the origin of this population, its relation to the extraembryonic tissue, its exact contributions to the developing heart as well as its heterogeneity are needed to shed further light on this aspect.

The toolbox presented in this study can be used to differentiate hPSCs into various mesodermal cardiac progenitor pools that contribute to heart formation, including aSHF, FHF, as well as the JCF, offering novel approaches for deciphering human cardiac development and for studying congenital heart disease. Combined with the current advances in self-organizing 3D cardiac tissue cultures[15,54–56], it could provide the opportunity to generate even more complex heart structures and gain unprecedented insights into human cardiac development and disease.

**Fig. 6 | Human CPC-RA+ and CPC-RA− integrate within distinct regions of the developing mouse embryo heart. a** Schematic representation of the protocol used to inject and culture murine embryos ex vivo. **b**–**d** Representative brightfield images of: **b** cell injection into a cardiac crescent (cc) stage embryo (solid line indicates site of injection) and after **c** 24 h and **d** 48 h of ex vivo culture (embryos without yolk sac and amnion; dotted lines indicate the heart region). The right panels of **c** and **d** show a frontal 3D view of the heart stained for cTnT (magenta) showing elongation of the heart tube at 24 h and looping at 48 h. Scale bars = 100 μm. **e** Representative immunofluorescence images of the heart region (marked by cTnT in magenta) of embryos injected with human CPC-RA+ and CPC-RA− (marked by HNA in green) after 24 h of culture ex vivo. Scale bars = 100 μm. Top insets show an overview of the whole embryo. Bottom inset depicts a high magnification of the boxed region. Arrowheads indicate human cells. Stars indicate unspecific stain of the yolk sac. **f** Table and schematics summarizing the regional distribution and number of human cells derived from CPC-RA+ and CPC-RA− found integrated in the murine heart after 24 or 48 h of ex vivo embryo culture. **g** Percentage of embryos with cells derived from CPC-RA+ or CPC-RA− in specific cardiac regions of the mouse heart after 24 or 48 h of culture ex vivo. IFT inflow tract, LV left ventricle, OFT outflow tract, RV right ventricle. **h** High magnification images showing the integration of human CPC-derived cardiomyocytes (HNA⁺/cTnT⁺) in the murine myocardium (marked by cTnT in magenta) following injection of CPC-RA+ or CPC-RA−. Arrowheads indicate human cells marked by HNA (green). Nuclei were counterstained with Hoechst-33258 (blue). Scale bars = 10 μm. **i** Percentage of human CMs (HNA⁺/cTnT⁺) and non-CMs (HNA⁺/cTnT⁻) found within the mouse heart following injection of CPC-RA+ or CPC-RA−. Data represents average percentages for each condition. n = 4 independent embryos for each condition.

## Methods

This study was approved by the Ethics Commission of the Technical University of Munich (TUM) Faculty of Medicine (# 447/17S).

### hESC/hiPSC cell culture

Human iPSCs were generated using the CytoTune-iPS 2.9 Sendai Reprogramming Kit (Invitrogen; A16157) as previously described[40,57]. The following hiPSC lines were used in differentiation experiments: hPSCreg MRIi003-A (hiPSC/CTRL), hPSCreg MRIi003-A-8 (hiPSC eGFP⁺), MRIi018-A (HLHS-1), DHMi003-A (HLHS-2). HLHS-1 carries de novo mutations in *DENND5B*, *SYBU* and *BAI2*, and HLHS-2 carries de novo mutations in *MACF1*, *NFDUFB10*, *MYRF*, *AIM1L*. Details on de novo calling, genetic background and clinical phenotype have been previously reported[40]. Authorization to use the hESC line ES03 (ES03/CTRL) (hPSCreg ESI-BIe003) generated by ES Cell International Pte Ltd in Singapore was granted by the Central Ethics Committee for Stem Cell Research of the Robert Koch Institute to AM (AZ 3.04.02/0131). Karyotype analysis was performed at the Institute of Human Genetics of the Technical University of Munich using standard methodology.

All hESC and hPSC lines were maintained in E8 medium containing 0.5% Penicillin/Streptomycin (15140-122, ThermoFisher Scientific) on Geltrex (A1413302, ThermoFisher Scientific) coated dishes under standard conditions (37 °C, 5% CO₂). Cells were non-enzymatically passaged every 4 days using 0.5 mM EDTA (AM9912, ThermoFisher Scientific) in PBS(−/−). To promote survival, Thiazovin (SML1045, Sigma-Aldrich) was added at a concentration of 2 μM for 24 h after passaging.

### Generation of the reporter cell line by gene targeting

The method of gene targeting applied here was developed by the Stanley and Elefanty lab in Melbourne Australia[58,59]. In brief, ES03 cells were grown on MEF feeders in KSR + bFGF (233-FB, R&D) and passaged enzymatically using trypsin-like enzyme (TrypLE, 12605010, ThermoFIsher Scientific) for at least three passages before electroporation. Ten million cells were suspended in a total of 800 μl of ice-cold PBS containing 20 μg of linearised targeting vector in a 4 mm gap cuvette. Electroporation was performed at 250 V and 500 μF. Cells were pipetted in pre-warmed medium and centrifuged at 259 × g for 3 min. The cell pellet was resuspended in warm KSR containing 12 ng/ml bFGF and spread over nine 6-cm dishes containing antibiotic resistant MEF feeders. 5 days after plating, medium was changed to KSR + FGF containing G418 (A1720, Sigma-Aldrich). About 7 days after the start of selection, colonies were picked and split in two wells, a DNA well and a maintenance well. Cells in the DNA well were lysed for at least 3 h at 55 °C (Lysis buffer: 100 mM TrisHCl, pH 8.0, 200 mM NaCl, 5 mM EDTA, pH 8.0, 0.2% (w/v) SDS powder (L4509, Sigma-Aldrich), 200 μg/ml proteinase K (P2308, Sigma-Aldrich) and DNA was isolated using isopropanol precipitation. The DNA pellet was washed once in 70% Ethanol and after air-drying resuspended in 50–100 μl TE buffer. Long-range PCR over the 5′ homology arm was used to screen for correct integration.

The resistance cassette can potentially interfere with expression of the reporter gene[60,61]. Therefore, removing the resistance cassette ensures that reporting of locus activity of the targeted gene is faithful and specific. The design we used for the targeting vectors includes rox sites flanking the puromycin resistance cassette. These rox sites can be recombined with Dre, the D6 site-specific DNA recombinase[62] similar to the Cre/loxP system. To avoid issues with genomic integration of the expression plasmid, we chose to transduce the DRE protein directly using a fragment of the HIV TAT protein, which has been previously used as a cell-penetrating peptide[63,64]. The TAT-DRE fusion protein was provided by Dr. Marko Hyvönen from the Department of Biochemistry (University of Cambridge). Cells grown on feeders were incubated with 2 μM TAT-DRE/CRE in serum free medium for 6 h. After this treatment, the cells were placed back in KSR medium supplemented with bFGF. Removal of the resistance cassette was evaluated using primers flanking the resistance cassette. At the end of the process, cells were clonally derived by sorting single cells into 96 well plates in order to ensure a homogeneous population of targeted cells (and excision of the resistance cassette). Correct clones were identified by long range PCR on genomic DNA for targeting and the absence of the resistance cassette.

### Cardiac differentiation

Cardiac differentiation protocol was adapted from Mendjan et al.[8], Hofbauer et al.[15]. Briefly, cells were plated on Geltrex-coated 24-well plates at a density of 200,000 cells/well in E8 medium containing 2 μM Thiazovin (day −1). The next day, medium was replaced with CDM-BSA medium—consisting of 1:1 DMEM/F-12 with Glutamax (31331028, ThermoFisher Scientific) and IMDM (21980032, ThermoFisher Scientific) containing 0.1 g/ml BSA (A9647, Sigma-Aldrich), 15 μg/ml transferrin (10652202001, Roche), 1% chemically defined lipid concentrate (11905031, ThermoFisher Scientific), -0.46 mM (0.004%) of thioglycerol (T1753, Sigma-Aldrich)—supplemented with 10 ng/ml BMP4 (314-BP, R&D), 1.5 μM CHIR99021 (4423, R&D), 50 ng/ml Activin A (SRP3003, Sigma-Aldrich), 30 ng/ml bFGF (233-FB, R&D) and 5 μM LY 294002 hydrochloride (1130, R&D). After 40 h, the medium was replaced with CDM-Meso medium consisting of CDM-BSA supplemented with 10 ng/ml BMP4, 8 ng/ml bFGF, 10 μg/ml insulin (11376497001, Roche/Sigma-Aldrich), 5 μM IWP2 (72122, Stem Cells). In addition, the medium was supplemented or not with RA (R2625, Sigma-Aldrich) in varying concentrations between 0.5 and 1 μM as indicated. CDM-Meso was changed the next 4 days in 24 h intervals. Depending on the treatment, RA was added to media for 2 or 4 days. On day 5 and 6, the medium was replaced with CDM-BSA supplemented with 10 ng/mL BMP4, 8 ng/ml bFGF, and 10 μg/ml insulin. From day 7 on, the medium was replaced every second day with CDM-Maintenance medium consisting of CDM-BSA supplemented with 10 μg/ml insulin.

### FACS and flow cytometry analysis of live cells

For flow cytometry based sorting and analysis of live cells, cells between day 0 and day 8 of differentiation were washed two times with

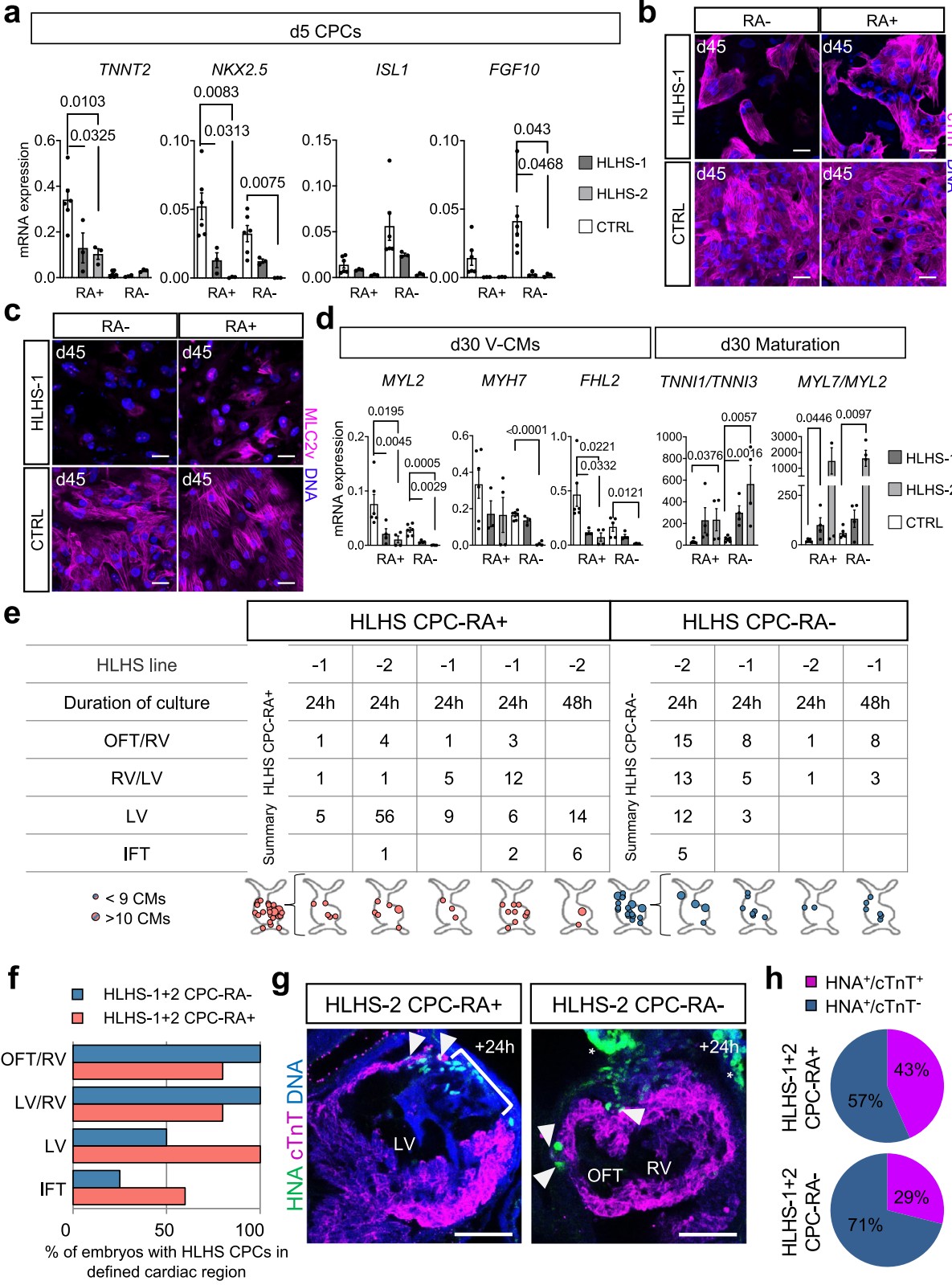

PBS(−/−), dissociated with Accutase (A11105013, ThermoFisher Scientific 5 min, 37 °C), centrifuged at 300 × *g* for 5 min, resuspended in 2% FCS in PBS, filtered through a 40 µm filter and subjected to sorting procedure on a FACS Aria III Cell Sorter (BD) or subjected to live flow cytometry analysis procedure on a Cytoflex S (Beckman-Coulter). DAPI (D3571, ThermoFisher Scientific) staining was used to discriminate dead and live cells (final concentration 0.01 ng/µl). Cells were sorted

based on mCherry and eGFP expression and collected into 50–100% FCS in PBS, unless otherwise indicated. Analysis was performed using Kaluza software (v2.1) Beckman-Coulter.

### 3D culture of sorted cells
After sorting, mCherry⁺, eGFP⁺, double eGFP⁺/mCherry⁺, ITGA8⁺, eGFP⁺/ITGA8⁺, eGFP⁺/ITGA8⁻ cells were centrifuged, resuspended in

**Fig. 7 | Human CPC-RA+ and CPC-RA− generated from HLHS patients-derived hiPSCs show heart-field specific defects in vitro and ex vivo. a** mRNA expression of cardiovascular progenitor (CPC) markers relative to *GAPDH* at day 5 of differentiation of HLHS patients-derived hiPSCs (HLHS-1, HLHS-2) and healthy control hiPSCs (CTRL). Data are mean ± SEM. CTRL values depict data from two independent cell lines. *n* = 3 independent experiments. Exact *p*-values of unpaired two-tailed *t*-test are shown. **b**, **c** Representative immunofluorescence images of cells stained for **b** cTnT and **c** MLC2v (both in magenta) at day 45 of RA− or RA+ differentiation examined over 2 (HLHS-1) and 3 (CTRL) independent experiments. Nuclei were counterstained with Hoechst-33258 (blue). **d** mRNA expression of markers of ventricular cardiomyocytes (V-CMs), and ratio between mRNA expression levels of *TNNI1* and *TNNI3* as well as *MYL7* and *MYL2* relative to *GAPDH* and *TNNT2* at day 30. Data are mean ± SEM. CTRL values depict data from two independent cell lines. *n* = 3 independent experiments except for *MYL7/MYL2* ratio for which CTRL-2 *n* = 2. For HLHS *n* = 4 independent experiments except for *MYH7* for which HLHS-1 *n* = 3. Exact p-values of unpaired two-tailed *t*-test are shown.

**e** Regional distribution of human cells derived from HLHS-derived CPC-RA+ and HLHS-derived CPC-RA− integrated in the murine heart after ex vivo embryo culture. IFT inflow tract, LV left ventricle, OFT outflow tract, RV right ventricle. **f** Percentage of embryos with cells in specific cardiac regions of the mouse heart after 24 or 48 h of culture ex vivo. **g** Representative immunofluorescence images of the heart region (marked by cTnT, magenta) of embryos injected with human HLHS-derived CPC-RA+ and CPC-RA− (marked by HNA, green) after 24 h of ex vivo culture. Two independent experiments were performed showing similar results within each group. Arrowheads indicate human cells. The bracket displays a gap within the host LV myocardium filled with non-myocytic HLHS-derived CPC-RA+. Stars indicate unspecific stains of the yolk sac. Nuclei were counterstained with Hoechst-33258 (blue). Scale bars = 50 μm. **h** Percentage of human CMs (HNA$^+$/cTnT$^+$) and non-CMs (HNA$^+$/cTnT$^-$) within the mouse heart following injection of HLHS-derived CPC-RA+ or CPC-RA−. Data represents average percentages for each condition. CPC-RA+ *n* = 5; CPC-RA− *n* = 4 embryos. Source data are provided as a Source Data file.

CDM-BSA, counted, centrifuged again, and resuspended in CDM-Meso containing 0.5% Penicillin/Streptomycin, 10 μM of Rock Inhibitor Y-27632 (688000, Calbiochem) and supplemented or not with RA in varying concentrations as indicated at a concentration of 100,000 cells per 200 μl per well of a U-shaped 96-well plate previously coated with 5% Poly(2-hydroxyethyl methacrylate) (P3932, Sigma-Aldrich). Plates were centrifuged for 2 min at 300 × *g* and transferred to an incubator. From the next day on, the cardiac differentiation protocol was applied as described above. At day 7 aggregates were transferred into the wells of a 48-well plate previously coated with 5% Poly(2-hydroxyethyl methacrylate) and put on a shaker. From this point on CDM-Maintenance was supplemented with 50 ng/μl VEGF (293-VE, R&D).

### Epicardial differentiation

For epicardial differentiation, day 4.5 FACS sorted ITGA8$^+$ cells were re-plated on 0.1% gelatin-coated wells of 12-well chamber slides (81201, Ibidi) in density of 40,000 cells per 1 cm$^2$ and subjected to modified protocol Bao et al.[65]. Briefly, cells were re-plated in LaSR medium consisting of Advanced DMEM/F12 containing Glutamax, Ascorbic Acid (0.1 mg/ml; A5960, Sigma) and 0.5% Penicillin/Streptomycin (LaSR) with addition of 1% FCS and 10 μM of Rock Inhibitor Y-27632. On day 6 and day 7 the medium was replaced with LaSR supplemented with 3 μM CHIR99021. From day 8 onwards LaSR medium was replaced daily till day 12.

### Epicardial–myocardial differentiation

For simultaneous epicardial and myocardial differentiation, day 4.5 eGFP$^-$ cells were replated on Geltrex-coated wells of 12-well chamber slides (81201, Ibidi) and 96-well plates (89646, Ibidi) in density of 40 000 cells per 1 cm$^2$. Immediately, FACS-sorted eGFP$^+$/ITGA8$^+$ and eGFP$^+$/ITGA8$^-$ cells were added in the number of 5–10 cells per well resulting in 0–5 cells per well on the next day. Cells then were subjected to modified cardiac differentiation protocol described above. Modulations of BMP, RA and WNT signaling cues were described previously to favor either myocardial or epicardial lineages[29] and thus we decided to use low levels of WNT signaling inhibition with concomitant addition of BMP4 and RA. Briefly, cells were re-plated in CDM-BSA media supplemented with 20 ng/ml BMP4, 8 ng/ml bFGF, 10 μg/ml insulin, 2.5 μM IWP2, 1 μM RA with addition of 0.5% Penicillin/Streptomycin, 1% FCS, 10 μM of Rock Inhibitor Y-27632. On day 6 and day 7 the medium was replaced with CDM-BSA supplemented with 20 ng/mL BMP4, 8 ng/ml bFGF, 10 μg/ml insulin, and 1 μM RA with addition of 0.5% Penicillin/Streptomycin. From day 8 onwards CDM-Maintenance with addition of 0.5% Penicillin/Streptomycin media was replaced daily till day 15.

### Single-cell dissociation

Cells were dissociated using Accutase up to day 8. From day 8 on cells or aggregates were subjected to papain-based dissociation. Briefly, cells/aggregates were washed two times with 2 mM EDTA. Dissociation was carried out using a papain solution prepared as described by Fischer et al.[66]. Cells/aggregates were incubated with papain solution for 20–40 min at 37 °C (the optimal dissociation time was previously determined for each cell line/time point to obtain a single cell suspension without compromising cell quality and survival). Aggregates were incubated on a shaker at 37 °C. After that time, a solution containing 1 mg/ml trypsin inhibitor (T9253, Sigma-Aldrich) was added. Cells/aggregates were dissociated by pipetting, transferred to a tube containing PBS(−/−), centrifuged at 300 × *g* for 5 min and resuspended in appropriate solutions depending on the downstream analysis.

### Cardiomyocytes replating for immunofluorescence

At day 30 of cardiac differentiation, media was changed to EB2 media consisting of DMEM/F12 (21331-020, ThermoFisher Scientific), 2% FBS, 1% L-glutamine (25030-081, ThermoFisher Scientific), 1% non-essential aminoacids (11140-050, ThermoFisher Scientific), 0.5% Penicillin/Streptomycin and 0.1 mM beta-mercaptoethanol and changed every second day. At day 40 of cardiac differentiation, cells were subjected to papain-based dissociation as described above. Cells were replated on fibronectin (F1141, Sigma-Aldrich) coated wells of 12-well chamber slides in a density of 140,000–200,000 cells per 1 cm$^2$ in EB20 media consisting of DMEM/F12, 20% FBS, 1% L-glutamine, 1% non-essential aminoacids, 0.5% Penicillin/Streptomycin and 0.1 mM beta-mercaptoethanol. The next day media was changed for EB2 and changed daily till day 45.

### RNA isolation, reverse transcription PCR (RT-PCR), and quantitative real-time PCR (qPCR)

For RNA isolation cells/aggregates were dissociated as described above. For time course analysis of day 0 to day 8 RNA collection was performed 2 h after media change. In case of aggregates, 3–5 aggregates were collected from each differentiation for RNA collection. Cell pellets were lysed and RNA was isolated using the Absolutely RNA Microprep Kit (400805, Agilent Technologies), Absolutely RNA Nanoprep Kit (400753, Agilent Technologies) or Rneasy Microkit (74004, Qiagen), depending on the cell number. For RNA isolation of ITGA8$^+$ sorted cells, after sorting and centrifugation in PBS, cell pellets of 20,000 cells were lysed. 0.3–0.5 μg of RNA was used to synthesize cDNA with the High Capacity cDNA Reverse Transcription kit (4368813, Applied Biosystems). 40 ng was used to synthesize cDNA from ITGA8$^+$ cells with the SuperScript IV Vilo Master Mix (11756050, ThermoFisher Scientific). Gene expression was quantified by qPCR using 1 μl cDNA, the Power SYBR Green PCR Master Mix (4367659, Applied Biosystems), the primers listed in Supplementary Data 9 and a 7500 Real-Time PCR System (Applied Biosystems). Gene expression levels were quantified relative to *GAPDH* expression using the ΔCt method, unless otherwise indicated.

## Single-cell RNA sequencing (scRNA-seq)

For scRNA-seq at day 1.5, cells were washed two times with PBS, dissociated with Accutase (3–5 min, 37 °C), centrifuged at $300 \times g$ for 5 min, resuspended in 0.04% BSA in PBS, filtered through a 40 µm filter and counted. For scRNA-seq at day 4.5 of differentiation, cells were subjected to sorting as described above. Cells were sorted based on mCherry and eGFP expression and collected into 10% FCS in PBS. Then cells were centrifuged, washed in 0.04% BSA in PBS, centrifuged again, filtered through a 40 µm filter and resuspended in 0.04% BSA in PBS for counting. For scRNA-seq at day 30, aggregates were dissociated with papain as described above, filtered through a 40 µm filter, centrifuged at $200 \times g$ for 3 min and resuspended in 0.04% BSA in PBS for counting. After counting, 10,000 cells for each sample were processed using the Chromium Single Cell 3′ Library & Gel Bead Kit v3.1 (1000075, 10x Genomics), Chromium Single Cell B Chip Kit (1000073, 10x Genomics), and Chromium i7 Multiplex Kit (220103, 10x Genomics) to generate Gel Bead-In-EMulsions (GEMs) and single-cell sequencing libraries. Libraries were pooled and sequenced using the NextSeq 500/500 (Ilumina; High Output v2 kit 75 cycles v2.5 flow cell) with 28 cycles in read1 for the 10x barcodes and UMIs in and 8 cycles i7 index read and 58 cycles for cDNA in read2 with a read depth of at least 20,000 pair reads per cell. All day 4.5 or day 1.5 and day 30 samples subjected to scRNA-Seq procedure came from one differentiation ($n = 1$) and were processed at the same day together. Day 4.5 or day 1.5 and day 30 libraries were pooled and sequenced together.

The Cell Ranger pipeline (v6.1.1) was used to perform sample demultiplexing, barcode processing and generate the single-cell gene counting matrix. Briefly, samples were demultiplexed to produce a pair of FASTQ files for each sample. Reads containing sequence information were aligned using the reference provided with Cell Ranger (v6.1.1) based on the GRCh37 reference genome and ENSEMBL gene annotation. PCR duplicates were removed by matching the same UMI, 10x barcode and gene were collapsed to a single UMI count in the gene-barcode UMI count matrix. All the samples were aggregated using Cell Ranger with no normalization and treated as a single dataset. The R statistical programming language (v3.5.1) was used for further analysis. Count data matrix was read into R and used to construct Seurat object (v4.0.1)[67]. The Seurat package was used to produce a diagnostic quality control plots and select thresholds for further filtering. Filtering method was used to detect outliers and high numbers of mitochondrial transcripts. These pre-processed data were then analyzed to identify variable genes, which were used to perform principal component analysis (PCA). Statistically significant PCs were selected by PC elbow plots and used for UMAP analysis. Clustering parameter resolution was set to 1 for the function FindClusters() in Seurat. For sub-clustering analysis we used clustree package (v0.4.3). The clustering analysis of the day 30 dataset resulted in three clusters that were characterized solely by very low read counts (5, 15, 17) and were removed from further analyses.

All DEGs were obtained using Wilcoxon rank sum test using as threshold $p$-value $\leq 0.05$. We used adjusted p-value based on Bonferroni correction using all features in the dataset. For the cell type-specific analysis, single cells of each cell type were identified using FindConservedMarkers function as described within Seurat pipeline. For all the gene signatures analyzed, we used a function implemented in *yaGST* R package (v2017.08.25) (https://rdrr.io/github/miccec/yaGST/)[68].

To compare our dataset with published scRNA-sequencing datasets we identified common genes with datasets from humans[25] based on Homo sapiens gene symbols. For datasets from mouse[4–6] IDs were converted to Homo sapiens gene symbols according to the ortholog list from Ensemble (only one-to-one orthologues were considered[69]). The datasets were combined using the CCA workflow implemented in the Seurat package (v4.1.1)[67] based on 10 dimensions and 2000 anchor features. After integration the Euclidean distances between the different clusters were calculated based on the averaged, corrected, normalized counts followed by hierarchical clustering using Ward's hierarchical agglomerative clustering method[70]. The results were plotted as a dendrogram.

For evaluating lowly expressed transcripts as in the case of RA receptors (RXRs) and co-receptors (RARs) missing values were imputed using adaptively thresholded Low-Rank Approximation (ALRA) with standard settings[71]. In case imputed values are displayed the figure panels are labeled accordingly.

## Flow cytometry

**Intracellularly/intranuclearly stained cells.** For intracellular/intra-nuclear staining for flow cytometry cells were dissociated, counted, and distributed in equal numbers per sample in 15 ml tubes (typically $2 \times 10^6$ cells per sample per tube). Cells were fixed with 4% PFA for 7 min at room temperature (RT; $500\,\mu l/10^6$ cells), then centrifuged 3 min at RT at $400 \times g$. Next cells were washed three times with PBS (shaked for 5 min and centrifuged 5 min at $400 \times g$ between each wash). Then cells were stored in 2% FCS in PBS+/+ at 4 °C or incubated right away with blocking/permeabilization buffer containing 10% FCS, 0.1% Triton-X-100 (Sigma, T8707), 0.1% saponin in PBS(+/+) (for intranuclear staining) or 10% FCS, 0.1% saponin in PBS+/+ (for intracellular/membrane staining) for 1 h at RT on a shaker ($1\,ml/10^6$ cells). Then cells were centrifuged and incubated with the primary antibody (antibodies and dilutions used are provided in Supplementary Data 10) diluted in 1% FCS, 0.1% saponin with or without 0.1% Triton-X-100 in PBS+/+($500\,\mu l/10^6$ cells), overnight at 4 °C on a shaker. Then cells were washed three times with 0.1% saponin in PBS+/+ with or without 0.1% Triton-X-100 for a total of 45 min on a shaker (cells were centrifuged between washes). Then the secondary antibody (Supplementary Data 11) was added diluted 1:500 in 1% FCS, 0.1% saponin in PBS+/+ with or without 0.1% Triton-X-100, ($500\,\mu l/10^6$cells) and cells were incubated for 1 h at RT, protected from light, on a shaker. After that, cells were washed three times with 0.1% saponin in PBS+/+ with or without 0.1% Triton-X-100 for a total of 45 min on a shaker. Next, cells were resuspended in 2% FCS in PBS+/+ ($100\,\mu l/10^6$ cells), passed through a 40 µm strainer and subjected to analysis on a CytoFlex S or Gallios (Beckman Coulter). Analysis was performed using Kaluza software (v2.1) (Beckman Coulter). No-primary antibody, no-secondary antibody, IgG antibody controls were performed.

**Surface protein-stained cells.** For flow cytometry analysis of surface protein-stained cells, cells were dissociated, counted, and distributed in equal numbers per sample in 15 ml tubes (typically $5 \times 10^6$ cells per sample per tube). Samples were incubated with ITGA8-Alexa-Fluor-647-conjugated (ITGA8-APC) or IgG1-APC-conjugated (IgG-APC) antibodies (Supplementary Data 10) diluted in the FACS buffer containing 2% FCS in PBS (10 µl antibody/100 µl buffer/$10^6$ cells or 1:50, respectively (Supplementary Data 10) for 30 min on ice. Then cells were washed three times with FACS buffer and resuspended in FACS buffer for sorting. Before sorting DAPI was added to samples at a final concentration of 0.01 ng/µl to discriminate dead and live cells. Cells were sorted into tubes containing 1 ml of 50% FCS in PBS, washed with PBS, centrifuged, and resuspended in lysis buffer for RNA extraction or cell culture media for further differentiation.

## Immunofluorescence staining of cells

Cells were fixed with 4% PFA for 15 min at RT, washed 3 times with PBS, blocked for 1 h in 0.1% Triton-X-100 PBS containing 3% BSA and stained using primary antibodies (antibodies and dilutions used are provided in Supplementary Data 10) diluted in 0.1% Triton-X-100 PBS containing 0.5% BSA overnight at 4 °C. Samples were then washed three times for 5 min with 0.1% Triton-X-100 PBS and incubated with secondary antibodies (Supplementary Data 11) diluted 1:500 in 0.1% Triton-X-100 PBS

containing 0.5% BSA for 1 h at RT. Specimens then were washed three times with 0.1% Triton-X-100 PBS. Nuclei were detected with 5 μg/ml Hoechst 33258 (94403, Sigma-Aldrich) (5 min). Slides were washed once again with PBS, covered with mounting medium and a cover slip and stored at 4 °C.

## Aggregates embedding, cryosectioning, and immuno-fluorescence staining

In preparation for cryosectioning and immunofluorescence staining aggregates were subjected to modified protocol of Lancaster & Knoblich[72]. Briefly, aggregates were transferred to wells of a 24-well plate containing 1 ml PBS+/+ using a cut 1000 μl pipette tip. PBS+/+ was replaced with 4% PFA, and aggregates were incubated for 30 min at room temperature. After that time. aggregates were washed three times for 5 min with PBS+/+. Then, 30% sucrose (Sigma-Aldrich, S9378) in PBS+/+ was added for overnight in 4 °C. For aggregates embedding, the 30% in PBS+/+ sucrose on aggregates was replaced with warmed up to 37 °C 1 ml 10% sucrose and 7.5% gelatin in PBS+/+ solution and allowed to equilibrate for 15 min at 37 °C on aggregates. Meanwhile, the bottom of a cryo-mold was covered with 400 μl sucrose/gelatin solution and placed at 4 °C to solidify. Next aggregates were transferred to molds using a cut 1000 μl tip, and placed for 3–5 min at 4 °C, then the cryomold was filled with 500 μl sucrose/gelatin solution and placed at 4 °C for 20 min to solidify. For aggregates freezing, such prepared molds were placed in cold 2-methylbutane (M32631, Sigma-Aldrich) for 1–2 min and stored at −80 Å°C. Molds were cryosected using Cryostat (Microm HM 560, ThermoFisher Scientific) into 12–16 μm slices on poly-L-lysine coated slides (J2800AMNZ, ThermoFisher Scientific), dried for 30 min at room temperature and stored at −80 °C. For immunofluorescence analysis, slides were dried for 30 min at room temperature after taking out of −80 °C freezer, fixed with 4 °C PFA for 10 min at room temperature, washed three times for 5 min with PBS+/+. Then permeabilized with 0.25% Triton X-100 for 15 min at room temperature, washed again three times for 5 min with PBS, blocked with 3% BSA in 0.05% Tween-20 (P2287, Sigma-Aldrich) in PBS+/+(PBST) for 1 h. Then primary antibodies (Supplementary Data 10) diluted in 0.5% BSA in PBST were added for overnight at 4 °C. Next day, slides were washed five times for 10 min with PBST. Then secondary antibodies (Supplementary Data 11) diluted in 0.5% BSA in PBST were added and slides were incubated for 2 h at room temperature. After that time, slides were washed five times for 10 min with PBST. Next, Hoechst 33258 at 5 μg/mL diluted in PBS was added for 15 min at room temperature. Slides were washed once with PBS, covered with mounting medium and cover slip and stored at 4 °C.

## Mouse-human embryo chimeras

**Timed mating and embryo dissection.** All animal experiments were performed in accordance with German animal protection laws and EU ethical guidelines (Directive 2010/63/EU). For this study, we used a mouse line with C57/B6 background. C57BL/6 male (2–6 months) and female (2–3 months) mice were housed at 20–24 °C, 45–60% humidity, and a dark/light cycle of 12/12 h. A day after overnight mating, mice were separated and checked for the presence of a vaginal plug (ED0.5). On the desired day of embryonic development, the status of gestation was evaluated and pregnant females were sacrificed. Pregnant females sacrificed for embryo collection within the first two-thirds of gestation were reported to the inspection authority as organ collection and did not require any additional ethical approvement. Embryos were dissected out of the uterus and placed in a dish with pre-warmed dissection medium (5% FCS, 1% Pen/Strep, 20 mM HEPES in DMEM (ThermoFisher Scientific, 31966). Embryos were carefully removed from the decidua and Reichert's membrane. Attention was paid to not remove the ectoplacental cone or destroy the yolk sack. Damaged

embryos were not used further. Embryos used for further experiments were not disaggregated for sex.

**Injection and whole embryo culture ex utero.** Unsorted CPCs at day 4.5 of differentiation were dissociated using Accutase as described above, resuspended in the CDM-BSA media and injected into the heart region of dissected mouse embryos at the cardiac crescent stage. At this stage, structures within the mouse embryo are small and FHF and SHF regions appear in the close proximity enabling targeting the whole heart-forming region with a single injection. Approximately 20–50 cells per embryo were introduced via a glass capillary of 20 μm inner diameter. Operated and stage-matched unoperated embryos were placed into media containing 50% rat serum (S2150, Biowest) and 50% DMEM for 24 h. Medium was changed after 24 h to media containing 75% rat serum and 25% DMEM (adapted from Aguilera-Castrejon et al.[73], Pliszek et al.[74]; similarly to other reports[75]). Embryos were cultured using a rotating incubator system (BTC Engineering) at 5% $CO_2$ and 5–20% $O_2$ depending on their developmental stage (adapted from Aguilera-Castrejon et al.[73], Pliszek et al.[74]; similarly to other reports[75]). At the end of the culture, embryos were removed from the incubator, dissected from the yolk sac and amnion, and evaluated in terms of heart beat and overall morphological development.

**Whole-mount immunofluorescence staining and clearing of embryos.** After evaluation, properly developed embryos were transferred to tubes and fixed in 4% PFA at RT for 1–2 h depending on size. Next, embryos were washed in PBS, and incubated in blocking/permeabilization solution containing 10% FBS, 0.1% Triton X-100 in PBS on a shaker at RT for 4 h. Embryos were then incubated with primary antibodies (Supplementary Data 10) diluted in 1% FBS, 0.1% Triton X-100 in PBS on a shaker at 4 °C overnight; after which they were 3× washed in 0.1% Triton X-100 in PBS for 3 h in total; and incubated with secondary antibodies (Supplementary Data 11) and Hoechst dye diluted in 1% FBS, 0.1% Triton X-100 in PBS on a shaker at RT for 2 h. After another round of washing the embryos were cleared using a protocol adapted from Masselink et al[76]. Briefly, embryos were sequentially transferred into dehydration solutions containing 30%, 50%, 70% and 2 × 100% 1-Propanol in PBS (pH adjusted to 9.0–9.5 using trimethylamine) and incubated on a shaker at 4 °C. Incubation times were adjusted considering the embryos' size (at least 4 h per dehydration step). After complete dehydration, the embryos were stored in ethyl cinnamate.

**Immunofluorescence staining of cryosections.** Embryos were subjected to a sucrose gradient (5–20%) followed by embedding in a 1:1 mixture of Tissue-Tek O.C.T. (Sakura, 4583) and 20% sucrose (Sigma-Aldrich, S9378), and frozen in a bath of 2-methylbutane chilled with liquid nitrogen. Samples were stored at −80 °C or cryo-sectioned into 8 μm slices transferred to polysine-coated slides. Sections were fixed in 4% PFA at RT for 10 min. After washing with PBS, samples were incubated in solution containing 10% FBS, 0.1% Triton X-100 in PBS at RT for 1.5 h. Next, they were incubated with primary antibodies (Supplementary Data 10) diluted in 1% FBS, 0.1% Triton X-100 in PBS at 4 °C overnight. The next day, samples were washed 3x with 0.1% Triton X-100 in PBS for 15 min total and incubated with secondary antibodies (Supplementary Data 11) and 5 μg/ml Hoechst 33258 diluted in 1% FBS, 0.1% Triton X-100 in PBS at RT for 1 h. After another round of washing, sections were covered with mounting medium and a cover slip and stored at 4 °C.

## Microscopy and image analysis

All images of stained differentiated cells were acquired using using confocal laser scanning microscopy (Leica Microsystems, SP8).

Images were assigned with pseudocolours and processed with Leica Application Suite X software (v3.5.7.23225) from Leica Microsystems. Pictures were processed only in terms of brightness and contrast. Changes were applied equally across the entire image, and to all samples imaged. Cleared embryos were transferred into an 8 Well Glass Bottom µ-Slide (80827, ibidi) and whole embryos imaged using confocal laser scanning microscopy (Leica Microsystems, SP8). Using the 3D reconstruction and section view tools from Leica Application Suite X software (v3.5.7.23225) heart compartments were determined to be OFT/RV, RV/LV, LV, and IFT region and integration of human cells into the host myocardium was verified, respectively. Sections of embryos were imaged using a THUNDER system (Leica Mircrosystems, 11525679). Images were captured using the Leica K5 camera (Leica Mircrosystems, 11547112), a 40× objective (Leica Mircrosystems, 11506203) and the filter cube set (Leica Mircrosystems, 11525480). To quantify the contribution of human cells to the heart of injected embryos, HNA⁺ cells were counted and the region of integration of HNA⁺/cTnT⁺ cells was noted. OFT and IFT could be clearly distinguished by their morphology and its right/cranial or caudal position, respectively. OFT and RV at the analyzed stage of heart development are not as clearly separated therefore combined in the category OFT/RV. Human cells which were not clearly found in left or right ventricle were assigned to the category RV/LV. Images were analyzed with Image J (Version v1.53a).

**Patch clamp electrophysiological recordings and data analysis**

Cardiomyocyte monolayers were washed twice with PBS and dissociated to single cells by adding 0.25% Trypsin-EDTA (Thermo-Fisher Scientific; 25200072) and incubating at 37 °C for 2–3 min. If clumps of cells were still present, the cells were incubated for a further 2–3 min. The cells were then pipetted gently six times to ensure dissociation and FBS was added to neutralize the activity of trypsin. Dissociated cells were centrifuged at $130 \times g$ for 3 min and resuspended in RPMI-1640 media (ThermoFisher Scientific; 11875093) supplemented with Penicillin/Streptomycin (Thermo-Fisher Scientific; 15140122), ITS-G (ThermoFisher Scientific; 41400045), chemically defined lipid concentrate (ThermoFisher Scientific, 11905031) and 1-Thioglycerol (M6145) (RI medium) containing 20% FBS. Cells were then seeded onto glass coverslips pre-coated with laminin (Sigma-Aldrich; L2020). Twenty-four hours later, cells were inspected for attachment and the medium was changed to RI medium. Cells were patch clamped 3–7 days after seeding (medium was changed every 3 days). Whole-cell current-clamp recording was carried out at room temperature ($22 \pm 2$ °C) using an Axopatch 200B amplifier (Molecular Devices, USA). Pipette (Intracellular) solution contained (mM): 110 K-D-gluconate, 20 KCl, 10 NaCl, 2 EGTA, 10 HEPES, 1 MgCl₂, 0.3 GTP, and 2 MgATP (pH 7.4 with KOH). Extracellular solution (Tyrode's) contained (mM): 135 NaCl, 5.4 KCl, 5 HEPES, 1 MgCl₂, 0.33 NaH₂PO₄, 2 CaCl₂ and 10 Glucose (pH 7.4 with NaOH). Pipette resistance, when filled with intracellular solution, was ~2–3.5 MΩ. Action potentials were Liquid Junction Potential (LJP) corrected (LJPc). The LJPc (13.4 mV) was calculated using the *Clampex* Junction Potential Calculator (Molecular Devices, USA). AP (Action Potential) properties were analyzed using *Clampfit* software (v10.4 and v10.7) (Molecular Devices, USA). Cardiomyocytes based on their action potentials were classified into 4 groups: Ventricular-like cardiomyocytes (V-CMs), Atrial-like cardiomyocytes (A-CMs); Nodal-like cardiomyocytes (N-CMs) and Intermediate (ventricular)-like cardiomyocytes (I-CMs) based on APD90/50 ratio. I-CMs = APD90/APD50 ratio between 1.4 and 1.8. V-CMs = APD90/APD50 ratio between 1.0 and 1.4. A-CMs = APD90/APD50 ratio > 1.8; N-CMs: were classified based on the typical shape of action potentials and signs of spontaneous depolarization.

**Statistics**

Statistical analysis was performed with GraphPad Prism version 5 and 8 (La Jolla California, USA). Bar graphs indicate the mean ± SEM with all data points displayed separately, unless otherwise indicated. Data from two experimental groups were compared either by unpaired Student's *t*-test or by Mann-Whitney-Wilcoxon test depending on the assumed distribution. For more than two experimental groups one-way analysis of variance (ANOVA) was used first, followed by Student's *t*-test or by Mann-Whitney-Wilcoxon for groupwise comparisons in case of a statistically significant result from the overall analysis. A *p*-value < 0.05 was considered statistically significant, unless otherwise indicated.

**Reporting summary**

Further information on research design is available in the Nature Portfolio Reporting Summary linked to this article.

## Data availability

The scRNA-seq data generated in this study have been deposited in the GEO database under accession code GSE197660. The raw scRNA-seq data by Zhang et al.[5] is available in the GEO database under accession code GSE176306. The processed data was downloaded from the University of California, Santa Cruz (UCSC) cell browser https://cells.ucsc.edu/?ds=chi-10x-mouse-cardiomyocytes. The raw scRNA-seq data by Tyser et al.[6] is available in the European Nucleotide Archive database under study PRJEB14363, and ArrayExpress, under accession E-MTAB-7403. The processed data was downloaded from the Cancer Research UK Cambridge Institute https://content.cruk.cam.ac.uk/jmlab/mouseEmbryonicHeartAtlas/. The raw and processed scRNA-seq data by Ivanovitch et al.[4] is available in the GEO database under accession code GSE153789. The raw and processed scRNA-seq data by Cui et al.[25] is available in the GEO database under accession code GSE106118. All the data generated in this study are either deposited in the above-mentioned repository or are provided in the Supplementary Information and Source Data file. Source data are provided with this paper.

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

## Acknowledgements

We would like to acknowledge Birgit Campbell, Christina Scherb, and Marco Crovella for their technical assistance, Gabrielle Lederer (Cytogenetic Department, TUM) for karyotyping, Dr. Rupert Öllinger (TUM, Germany) for sequencing, Dr. David Elliott for sharing the ESO3 and ESO3-NKX2.5^eGFP cell lines, Drs. Ed Stanley and Andrew Elefanty (MCRI, Australia) for advice in construct design and gene targeting, and Dr. Sasha Mendjan for advice and discussion. This work was supported by the European Research Council (ERC) (grant 788381 to A.Mo. and grant 261053 to K-.L.L.), the Else-Kroener-Fresenius Stiftung (EKFS, to A.G.), the German Research Foundation (grant GO3220/1-1 to A.G.; Transregio Research Unit 152 to A.Mo. and K-.L.L.; Transregio Research Unit 267 to A.Mo., K-.L.L., and P.G.), the German Centre for Cardiovascular Research (DZHK) (grant FKZ 81Z0600601 to A.Mo. and K-.L.L.; grant 81X3600607 to J.K.), the Fondazione Umberto Veronesi (to G.S.).

## Author contributions

D.Z., A.G., A.B.M., and A.Mo. conceived the study, interpreted the data, and wrote the manuscript. D.Z. designed and performed most of the experiments using hPSCs, including maintenance and differentiation of hPSCs, cell sorting, molecular assays, immunostainings, single cell libraries preparation, and analyzed data. G.S. and A.G. performed bioinformatic analyses. J.K. performed mouse embryo injections, ex vivo culture, and embryo analysis. D.O. and M.O. generated the ESO3-TN line and provided conceptual advice. M.N.-I. supported maintenance and differentiation of hPSCs and preparation of cells for downstream assays. M.L. performed some molecular assays and immunostainings. V.L.M. supported further development of differentiation protocol. T.D. supported analysis and data interpretation, and provided conceptual advice. S.C.H., M.N., A.T. performed patch clamp experiments. F.Z. generated hiPSC eGFP⁺ line. M.D. generated HLHS-2 line. M.T.D.A. supported some data interpretation. R.A.P. provided conceptual advice. P.G. provided conceptual advice and financial support. A.Mo., A.G., and K-.L.L. conceived and supervised the study and provided financial support. All authors read and approved the final manuscript.

## Funding

## Competing interests

D.O. is currently an employee at Bit Bio Ltd. (United Kingdom) and holds stock options. R.A.P. is an advisor at Meatable NV (Netherlands) and holds stock options, is co-founder of DefiniGen Ltd. (United Kingdom) and holds shares, and is an advisor at Bit Bio Ltd. (United Kingdom) and holds stock options. A.G. receives consultancy fees from Smart-Cella (Sweden). The remaining authors declare that they have no competing interests.

## Additional information

[1]First Department of Medicine, Cardiology, Klinikum rechts der Isar, Technical University of Munich, School of Medicine and Health, Munich, Germany. [2]German Center for Cardiovascular Research (DZHK), Munich Heart Alliance, Munich, Germany. [3]Regenerative Medicine in Cardiovascular Diseases, First Department of Medicine, Klinikum rechts der Isar, Technical University of Munich, School of Medicine and Health, Munich, Germany. [4]Department of Experimental and Clinical Medicine, University "Magna Graecia", Catanzaro, Italy. [5]Department of Surgery, University of Cambridge, Cambridge, UK. [6]Wellcome-MRC Cambridge Stem Cell Institute, Jeffrey Cheah Biomedical Centre, University of Cambridge, Cambridge, UK. [7]German Heart Center Munich, Department of Cardiovascular Surgery, Institute Insure - Technical University of Munich, School of Medicine and Health, Munich, Germany. [8]Bristol Heart Institute, Bristol Medical School, Translational Health Sciences, Bristol, UK. [9]School of Physiology, Pharmacology and Neuroscience, Faculty of Life Sciences, University of Bristol, Bristol, UK. [10]Clinical Pharmacology & Precision Medicine, William Harvey Research Institute, Barts and the London School of Medicine and Dentistry, Queen Mary University of London, London, UK. [11]Department of Obstetrics and Gynecology, Stanford School of Medicine, Stanford University, Stanford, USA. [12]Georg-Speyer-Haus, Institute for Tumor Biology and Experimental Therapy, Frankfurt am Main, Germany. [13]Institute of Cardiovascular Regeneration, Centre for Molecular Medicine, Goethe University, Frankfurt am Main, Germany. [14]Department of Surgery, Yale University School of Medicine, New Haven, USA. [15]Department of Cell and Molecular Biology, Karolinska Institute, Stockholm, Sweden. [16]These authors contributed equally: Dorota Zawada, Jessica Kornherr, Anna B. Meier, Gianluca Santamaria. ✉e-mail: laugwitz@mytum.de; amoretti@mytum.de; alexander.goedel@tum.de

