## [Peer Review File · Nature Communications]

Retinoic acid signaling modulation guides in vitro specification of human heart field-specific progenitor poolsEditorial Note: Parts of this Peer Review File have been redacted as indicated to remove third-party material where no permission to publish could be obtained.

REVIEWER COMMENTS

Reviewer #1 (Remarks to the Author):

The goal of Zawada et al. is to determine how retinoic acid impacts cardiac specification in human PSCs by utilizing genetic labeling, transcriptomics, and human-mouse embryonic chimeras. Here, the authors show that modulating retinoic acid (RA) at different levels and applying at different times of human PSC differentiation instructs human PSCs to form heart field specific progenitor pools with distinct fate potentials including first heart field (FHF) cells, second heart field (SHF) cells and a recently discovered juxta-cardiac field (JCF) cells. They further were able to purify JCF progenitor cells and culture them into myocardial and epicardial cells, thus confirming the potentials of these cardiac progenitors. Applying this system to a HLHS human PSC system, they also show how these heart fields may be disrupted using their RA differentiation protocol. Overall, the studies are timely and interesting, especially in light of the recent discovery of the JCF progenitor cells. However, some of the data is not always consistent with the authors' own conclusions and at times more in line with previous studies showing that RA promotes atrial cardiomyocyte differentiation. If the authors could clarify these results and possibly explain their findings better, it would help strengthen conclusions of the results and help explain differences between their RA findings and previous findings on the role RA and atrial cardiomyocyte differentiation. Overall, the findings are interesting and potentially a valuable contribution to the cardiac and stem cell fields but addressing some key issues as outlined below would help improve the manuscript for publication.

1. Figure 1g shows that atrial CM markers significantly increase, while MYL2 ventricular marker decreases with higher RA exposure. This finding does not seem to be consistent with the author's conclusions that application of RA makes FHF progenitors, and more specifically ventricular CMs. It is also inconsistent with Figure 1h, which shows only a 1% increase in atrial CMs, when the RNA expression of KCNA5 and KCNJ3 increases significantly in Figure 1g. Reconciling these discrepancies would be helpful to clarify the role of RA in specifying developmental heart fields and cardiac cell types.

2. It might be helpful for the authors to quantitate the number of cardiomyocyte cell types (atrial, ventricular, nodal) from the differentiation after different levels of exposure to RA. As is, much of the findings is based on somewhat conflicting transcriptional data in figure 1. Also, figure 1h is a percentage but it is unclear what the actual raw numbers of different cardiomyocytes that were used to calculate this percentage. In other words, how many cardiomyocytes were used to draw the conclusions that there is no appreciable increase in atrial cardiomyocytes after RA treatment, especially given the discrepancy in both the authors' transcriptional data and what has been previously reported after RA treatment.

3. Please compare data presented in this manuscript with previous findings that show RA treatment increases atrial cardiomyocyte levels after cardiomyocyte differentiation. What are the differences in the two studies that lead to the somewhat conflicting findings with the role of RA in atrial vs. ventricular cardiomyocyte differentiation.

4. The generated TBX5 and NKX2.5 generated lines are nice additions to the paper and the cardiac field. They appear to be knocked into respective loci, potentially disrupting the gene and raising concerns about potential haploinsufficiency issues with data generated from the lines. Can the authors please clarify whether this is indeed the case or not? If so, can they please confirm that these lines do not have issues with differentiation compared to a wild type line, particularly as it pertains to the manuscript with regards to generating specific heart fields and their downstream derivatives? At the very least, this is a limitation of the studies if the lines are haploinsufficient for these factors, especially as previous human studies have shown that haploinsufficiencies of these two factors can cause congenital heart disease.

5. It is difficult to appreciate what the clusters represent from the single cell data shown in figures 2, 3, etc. Please provide identification of all cell type clusters and the markers that were used to identify them. For instance, cluster 13 is a proliferative cpc ra- population but it is difficult to appreciate this based on dot plots, heatmaps, etc. shown. A more comprehensive and systematic representation of these clusters would help the readers and authors interpret and appreciate the data.

6. It would be helpful to look at the expression of RA receptors in the progenitor cells to understand the potential discrepancy between these findings and other results showing that RA generates atrial cardiomyocytes.

7. How many scRNA-seq replicates were performed? Also, please show evidence that there were no batch effects, particularly as cells appear to be segregating by samples on UMAP plots in some cases (e.g. Fig. 2b, 2b, etc.). Additionally, it seems that the cell number for each condition are different in the scRNA-seq analysis (Figure 3b), and this may affect the URD analysis. The time/stage should be provided for the URD. In the RA- URD, all of the cells come from a common *Mesp1* mesoderm progenitor, but in the RA+ URD, the non-cardiomyocytes and cardiomyocytes appear to come from two *Mesp1* mesoderm sources; however, RA isn't added until after the mesoderm stage, which would mean that there should not be two different mesoderm sources for the RA+ as shown in the URD. This difference may reflect the limitation of URD analysis as how tips are assigned may cause different URD tree outcomes. The authors should address and discuss this point and potential limitation, particularly as trajectories are only predictions that need to be functionally validated.

8. The results regarding the in vitro model for HLHS are interesting and should be expanded into a main figure. Additional experiments including functional studies on these cells with and without RA treatment would improve these results. Also providing information as the etiology/genetic mutation for the HLHS may shed light on the results observed from these studies.

Minor:

1. What are the transcriptional differences between the RA+ vs. RA- ventricular cardiomyocyte generated?

2. It would be nice to see a positive control for the ITGA8-APC antibody, since the population is very small and does not separate well from the rest of the cells (Figure 2f).

3. In figure 5, it seems that the author may have not labeled the LV and RV well; for example in Figure 5c (RA- heart), it seems that the positive cells are in LV. The authors should describe and show how they distinguish the LV and RV. The injection experiments are very difficult experiments and more replicates would help solidify findings.

Reviewer #2 (Remarks to the Author):

The manuscript from Zawada et al. describes a novel in vitro differentiation protocol that allows the generation and isolation of different cardiac progenitor cells from human pluripotent stem cells (hPSC) by modulating Retinoic Acid (RA)-mediated signaling. Several previous studies have shown the key role of RA signaling during heart formation from late primitive streak stage to defining the posterior boundary of heart fields. The authors investigated the influence of RA dosage and timing on the appearance of early human cardiovascular progenitors by using a growth-factor based protocol for the directed differentiation of hPSC towards cardiomyocytes. scRNAseq analysis of sorted cells demonstrated that cardiac progenitors in differentiation conditions acquire different cell fates depending on absence or presence of RA during early differentiation stages. In addition to the classical first and second heart field progenitors the authors propose that a subset of cells resembled progenitors of the recently discovered juxtacardiac field (JCF) by Tyser et al. (2021) and also Zhang et

al (2021), which contribute to epicardium as well as myocardium in mice.

-Although not easy to read this study is important as it is a protocol that can be used to differentiate hPSC into various mesodermal cardiac progenitor populations that contribute normally to heart development. The main scientific concern is that some proposed identities are not supported by the markers expressed in the identified clusters.

The authors used BMP4, FGF10, CXCR4 and LGR5 as markers of aSHF which are not the best markers for this population. What about TBX1?

After differentiation of hPSC into CMs with RA added at day 4 the authors observed strong upregulation of TBX5, which was used as FHF marker. However, studies have shown that TBX5 is also expressed in pSHF and its activation depends on RA activity (see also below).

THBS4 is used as FHF marker. However, Peisker et al (2022) have shown that fibroblast subpopulation (PDGFb+) expressed THBS4 after cardiac injury which suggest that this marker is not appropriate to identify FHF cell only.

The authors should be more convincing when they define cardiac progenitor identity.

- To trace the appearance of distinct cardiac progenitor cells the authors generated a double hESC line expressing eGFP and mCherry under the control of the endogenous NKX2.5 and TBX5 locus, respectively. Very useful tool. Using this double transgenic hESC the authors observed that after addition of RA both eGFP+/mCherry+ populations emerged from day 4 combined with a rapid downregulation of ISL1 and upregulation of cTnT in all populations. The authors should precise whether it is association to premature differentiation of cardiac progenitors or direct inhibition of ISL1 by RA signaling?

- The authors identified cluster 9 (Fig.2) as pSHF, which mainly derived from TBX5-pos/NKX2.5-neg progenitors confirming my point above that TBX5 is also a marker of the pSHF. The authors should discuss this point.

- Interestingly, among CPC-RA+ the authors observed two small clusters (Cluster 14 and 18) that expressed HAND1 and MAB21L2. They affirm that these progenitors resemble to the recently juxtacardiac field (JCF) that maps to the confluence of splanchnic and extraembryonic mesoderm (Tyser et al. 2021). This recent study identified that JCF cells express Hand1. However, the Me5 cluster identifies as JCF does not expressed NKX2.5 (Me5 cluster in figure 2; Tyser et al. 2021). In the current study CPC-RA+ that resembles to JCF derived from the NKX2.5-positive cells. This discrepancy should be addressed by the authors.

- Cardiac progenitors are multipotent and can differentiate into not only CMs but also into smooth muscle and endothelial cells during heart formation. The current study suggests that JCF progenitors are bipotent since they give rise to CMs and epicardial cells. Cell bipotency of JCF needs to be addressed. The authors identified ITG18 as marker that allows isolation of the JCF cells. This marker can be used to perform clonal analysis.

- To investigate the lineage commitment and functionality of in vitro derived cardiovascular progenitors the authors injected CPC-RA+ or CPC-RA- into cardiac crescent region of mouse embryos. Although I do agree with conclusion that the identity of CPC-RA+ and CPC-RA- at day 4.5 are already committed to contribute to their respective cardiac structure the authors should performed further experiment. We know that cardiac crescent contributes only to LV. So, to convince that CPC-RA+ cannot contribute to OFT/RV injection should be also performed in linear heart tube of mouse embryos.

Minor points:

Line 258: The CMs separated into five main groups. However, there are six groups in the figure. Can the authors describe also cluster 14?

Reviewer #3 (Remarks to the Author):

In their manuscript "Retinoic acid signaling modulation guides in vitro specification of human heart field-specific progenitor pools" Zawada and colleagues ask the important question of whether RA

signaling, when modulated in a dose- and time-specific manner, can control the formation of FHF versus aSHF progenitor populations from pluripotent stem cells. They show that an early pulse of RA, at the stage of emerging mesoderm cells, generates distinct CPCs from non-RA exposed cultures. These CPCs subsequently give rise to differentiated cells, cardiomyocytes and non-cardiomyocytes, that are different depending on whether they come from an -RA or +RA CPC. Lastly they show that CPC-RA and CPC+RA behave differently when transplanted into the early crescent stage mouse embryo, and that the new protocol to generate FHF CPCs can be used to establish an HLHS disease model.

Identifying new approaches to generate the heterogeneity of early CPC populations from PSCs is a highly relevant question with high impact for the field. Inducing RA signaling at a very early stage (as opposed to slightly later stages to form atrial cells) in order to generate a FHF CPC population has not been systematically tested to date, and represents an intriguing hypothesis. The paper is well written and the data are presented and described clearly. The experiments performed are well rationalized and appropriate for the question. The main concern is that the original hypothesis – that RA early during differentiation generated aSHF cells – is not fully supported by the data, or would need to be supported by additional analysis to be convincing.

Main comments:

1. Differentiation protocol and analysis of the role of RA during early cardiac development:

The manuscript carefully sets the stage for the RA analysis characterizing the early CPCs in different protocols (different RA dosage and exposure time). Some comments for this introductory analyses are:

-Given the well-established notion that a -RA differentiation protocol generates ventricular CMs, including left ventricular CMs, it is surprising that no FHF markers are detected in the -RA cells at the progenitor stage.

-Bulk PCR analysis can be biased by varying efficiencies or varying speed of differentiation in different protocols. A pan CPC marker such as NKX2-5 should be included in the bulk PCR analysis at the CPC stage to provide a sense for the amount of CPCs present in each condition. Additionally, TBX1 as an important aSHF marker and should be included in the analysis.

-Given the differential contributions of FHF and aSHF to the ventricular chambers, markers for left and right ventricular cardiomyocytes would be informative at day 30 (HAND1, CCK, PLN, CITED1). Related, the manuscript states that FHL2 is increased in a dose-dependent manner – the data shown do not support that (Fig. 1g).

-The most reliable metric to characterize CPC populations is to interrogate their progeny, which this study does well via multiple assays (EP analysis, bulk PCR). Given the importance of this question for the remainder of the study, differentiated cardiomyocytes at day 30 should be stained

(immunohistochemistry) for MLC2v and MLC2a or NR2F2, markers that reliably identify differentiated ventricular and atrial cardiomyocytes respectively during PSC differentiation. From most of the data presented, it looks like all of the cardiomyocytes in the +RA conditions express TBX5 at day 30 (transcript, protein and reporter), which may suggest that at least some cells have atrial identity.

-It appears that both markers (NKX2-5, TBX5) have been targeted with the same strategy. It is not clear why GFP is cytoplasmic and mCherry is nuclear in the reporter live analysis (Fig. 1l)?

2. Single cell RNAseq analysis on +/- RA treated cells:

Taking advantage of the unbiased single cell analysis approach to query identity and differences in different CPC populations during early development is a well-rationalized and promising approach. Some questions or concerns were noted as follows:

-The clustering analysis demonstrates distinct separation between the time points as well as the RA treatment conditions, except for the endoderm cells (Fig. 1b). This is rather surprising, given that cardiac progenitor cells from both the -RA and +RA protocol would be expected to be more similar to each other than to the mesoderm cells. Along similar lines, based on select marker expression and subsequent cluster annotation, there seem to be FHF and SHF populations in both conditions that

should locate more closely to each other. There are clusters (17 and 12) that exist in both populations yet they are located far apart. Collectively this indicates the presence of batch effects that obstruct the true understanding (similarities and differences) of some of these cell populations. The manuscript should better explain how the data from the different samples were integrated, and how batch effects were corrected in the analysis. If no batch correction was performed, this should be corrected in the analysis.

-The gene expression analysis in Fig. 1c is helpful to understand the different clusters. All of the identified clusters should be included in this analysis, not just what seems a selective choice for each sample.

-The D4.5 samples for this analysis were sorted. Feature plots of NKX2-5 and TBX5 of the UMPA in 1b should be included, either in main or supplementary figures to illustrate the success of this approach. It is not entirely clear why sorting was necessary, as the analysis of the resulting cell types of the -RA and +RA protocols, and thus the rationale for looking at the CPCs in each protocol were performed on non-sorted differentiations (all of Figure 1).

-The manuscript states several times that distinct clusters have a 'proliferative counterpart', but it is not explained how this conclusion has been taken, and only cell cycle status of the clusters is shown to support this statement (Fig. S4a).

3. Identification of the JCF population:

The study next delves deeper into a population that may represent the newly discovered JCF, and finds that ITG8 may identify such a population. IGF8+ cells at day 4 have the potential to form both cardiomyocytes and epicardium, which may constitute a new approach to study the JCF.

-WT1 should be included in the epicardial analysis, as it is a more epicardium-specific marker compared to the markers that are currently included (ZO1, CK18 and TCF21).

-Ideally one should compare another CPC population from the same differentiation for its potential to form cardiomyocytes and epicardium, to show that the ability to do so is to some extent specific to the JCF, and not broadly due to RA-exposure of mesoderm (as epicardium protocols are all based on addition of RA as well). The available reporter cell line should make this a feasible task.

4. Characterization of differentiated cells in -RA and +RA protocols:

The sorting followed by differentiation represents a rigorous and informative approach to test the differentiation potential of the CPCs in the -RA and +RA protocols, and this approach aligns well with the single cell transcriptomic data earlier in the paper. Most of my comments address data analysis strategies and whether single cell RNAseq analysis is sufficient to draw the conclusions drawn here.

-The rationale for including the D1 and D4.5 data in this analysis is unclear, as this comparison does not add any information on the day 30 cells themselves. Rather, day 30 cells should be analyzed separately, as this will likely provide a better understanding of the heterogeneity of these cells.

Without this, the current data display makes it challenging to truly understand the cellular heterogeneity of the cells derived from the distinct CPCs/protocols. The same batch correction comment noted in point 1 should be applied here as well.

-Related to the point above, the data shown for candidates such as MYL2 and TBX5 (Fig S6) suggest that these markers are indeed expressed, but they are not expressed in all cells of a given cluster but rather are expressed heterogeneously throughout all clusters, suggesting again that the current clustering analysis contains a substantial amount of heterogeneity still. Including that of important chamber-specific markers. In addition to interrogating known and expected markers in the analysis, the top differentially expressed candidates should be listed, for example as a supplementary table, to better understand what most prominently distinguishes the different CM clusters.

-While scRNAseq is a great first pass at the question of what CPC progeny is formed, the important conclusions in the paper should be strengthened by IF analysis of the many available distinct chamber-specific cardiomyocyte markers (MLC2v, MLC2a, Nr2f2) in both -RA and +RA derived populations.

-The paper in general ignores the fact that atrial cardiomyocytes can also be derived from FHF cells, albeit at a much lower propensity. Atrial markers, as well as left right ventricular markers should be included in the current analysis.

5. Fate decision tree analysis:

Using URD trajectory analysis is an informative approach to understand fate decisions, particularly as the data for both CPC and differentiated cells are available.

-To split the analysis between the -RA and +RA samples seems an unnecessarily biased way to do this however. Rather, the samples should be properly merged, batch corrected, subjected to URD analysis and then differences between -RA and +RA trajectories identified.

-Similarly to previous analysis, the study could benefit from identifying the candidates that most strongly mark the branching points in an unbiased manner, in addition to illustrating expression and segregation of known/expected markers.

6. Contribution of CPCs during mouse heart development:

This part of the study represents an ambitious approach to test CPC potential in vivo, during early heart development. Some comments apply as follows:

-The manuscript states that all CPCs have successfully integrated into the host myocardium. The data presented (Fig. 5c) do not support this conclusion, rather the CPC-derived cells appear to reside on the outside of the mouse heart, and the image quality does not allow to evaluate if the HNA+ cells do indeed express troponin. The manuscript should also state if the CPCs from both conditions were sorted prior to injection, to help understand how these results compare to the previous analyses.

-The distinction between a left versus a right ventricular location at this early stage seems challenging, as the chambers have not fully formed yet. It should be addressed what criteria were used to assign the cells a specific location.

-The analysis seems underpowered if the data is correctly interpreted: 5 embryos in -RA and 6 embryos in +RA?

7. HLHS disease modeling:

While an interesting question with respect to disease modeling in different differentiation conditions, or different CPC populations, the conclusions drawn are inappropriate given that the only data included are PCR data on bulk cultures from one control and one HLHS line each, without any additional characterization of the cells at any of the different stages.

Minor comments:

1. The citations for RA signaling to generate atrial cardiomyocytes are incomplete. Please add Devalla et al., 2015 and the original work by Zhang et al., 2011

2. Figure 2B: Cluster 17 in the -RA CPC sample in the UMAP is labelled in such a way that it might be interpreted as two clusters, 7 and 1

Point-by-point reply to the reviewer's comments for the manuscript
"Retinoic acid signaling modulation guides in vitro specification of human heart field-specific progenitor pools"

First, we want to thank all reviewers for their thoughtful comments and suggestions which helped us to significantly improve the manuscript. In the revised version of the manuscript, we have addressed all issues raised as outlined in the point-by-point reply below.

REVIEWER COMMENTS

Reviewer #1 (Remarks to the Author):

The goal of Zawada et al. is to determine how retinoic acid impacts cardiac specification in human PSCs by utilizing genetic labeling, transcriptomics, and human-mouse embryonic chimeras. Here, the authors show that modulating retinoic acid (RA) at different levels and applying at different times of human PSC differentiation instructs human PSCs to form heart field specific progenitor pools with distinct fate potentials including first heart field (FHF) cells, second heart field (SHF) cells and a recently discovered juxta-cardiac field (JCF) cells. They further were able to purify JCF progenitor cells and culture them into myocardial and epicardial cells, thus confirming the potentials of these cardiac progenitors. Applying this system to a HLHS human PSC system, they also show how these heart fields may be disrupted using their RA differentiation protocol. Overall, the studies are timely and interesting, especially in light of the recent discovery of the JCF progenitor cells. However, some of the data is not always consistent with the authors' own conclusions and at times more in line with previous studies showing that RA promotes atrial cardiomyocyte differentiation. If the authors could clarify these results and possibly explain their findings better, it would help strengthen conclusions of the results and help explain differences between their RA findings and previous findings on the role RA and atrial cardiomyocyte differentiation. Overall, the findings are interesting and potentially a valuable contribution to the cardiac and stem cell fields but addressing some key issues as outlined below would help improve the manuscript for publication.

We thank the reviewer for highlighting the relevance of our data and for the suggested additional analyses which helped us to improve our manuscript significantly.

1. Figure 1g shows that atrial CM markers significantly increase, while MYL2 ventricular marker decreases with higher RA exposure. This finding does not seem to be consistent with the author's conclusions that application of RA makes FHF progenitors, and more specifically ventricular CMs. It is also inconsistent with Figure 1h, which shows only a 1% increase in atrial CMs, when the RNA expression of KCNA5 and KCNJ3 increases significantly in Figure 1g. Reconciling these discrepancies would be helpful to clarify the role of RA in specifying developmental heart fields and cardiac cell types.

The reviewer is correct in stating that we observed some increase in atrial marker expression upon higher dosage of RA, albeit on a low expression level. We further confirmed this now with qPCR for the atrial marker *NR2F2* (revised Fig. 1g). However, this slight increase in expression level did not result in a significant increase in atrial-like CMs shown now by immunostaining for MLC2v and MLC2a as well as by flow cytometry for the atrial marker *NR2F2* (revised Fig. 1h, i), confirming our findings from the electrophysiology assay. To explore this further and resolve the discrepancy mentioned by the reviewer, we included another RA application scheme (1 μ M RA over 10 days). Applying this scheme, we observed a more pronounced increase in atrial markers on the expression level as well as an increase of *NR2F2*⁺ cells in flow cytometry (revised Fig. 1h, i). Notably, even in this condition, we only observed up to 40% *NR2F2*⁺ cells. We believe these results can be explained by

considering the type of mesoderm we are generating with our protocol, which, in line with observations in the literature (Lee et al., 2017), allows only a limited amount of atrial-like CMs to arise (more on this further below in our answer to question 3).

We have included the new results on page 5, starting at line 174, of the revised manuscript and stated now more clearly that there is a defined time and dosage window for RA that leads to an increase in FHF progenitors and subsequent (LV) ventricular CMs:

“To confirm our findings on the transcriptomic level we performed immunofluorescence analysis for the ventricular specific myosin light chain isoform – MLC2v, and atrial isoform – MLC2a (Fig. 1h) as well as flow cytometry-based quantification of cells expressing the atrial specific protein NR2F2 (Fig. 1i). Without addition of RA, most of the cells were exclusively positive for MLC2v and we observed only around 10% of NR2F2⁺ cells, confirming that most cells acquired a ventricular-like fate (Figure 1h, 1i). This did not change upon addition of RA up to 1 μM for 4 days (Figure 1h, 1i). However, longer exposure to 1 μM RA for 10 days resulted in an increase of MLC2a⁺ CMs (Fig. 1h) as well as NR2F2⁺ cells (Fig. 1i), suggesting partial atrial fate acquisition in line with previous reports²⁴.”

2. It might be helpful for the authors to quantitate the number of cardiomyocyte cell types (atrial, ventricular, nodal) from the differentiation after different levels of exposure to RA. As is, much of the findings is based on somewhat conflicting transcriptional data in figure 1. Also, figure 1h is a percentage but it is unclear what the actual raw numbers of different cardiomyocytes that were used to calculate this percentage. In other words, how many cardiomyocytes were used to draw the conclusions that there is no appreciable increase in atrial cardiomyocytes after RA treatment, especially given the discrepancy in both the authors’ transcriptional data and what has been previously reported after RA treatment.

Following the reviewer’s suggestion, we have performed immunostaining for MLC2v and MLC2a as well as flow cytometry for the atrial marker NR2F2 (revised Fig. 1h, i) to complement the data from the electrophysiologic studies. In addition, we have extended the qPCR panel and added *SHOX2* as well as *NR2F2* (revised Fig. 1g; revised Supplementary Fig. 1d). Moreover, we added the raw numbers for the quantification of the electrophysiology data to the supplementary table containing the details on this assay (Supplementary Data 1). Concerning the relation of our results to published literature we kindly refer the reviewer to our answers to question 1 and 3.

3. Please compare data presented in this manuscript with previous findings that show RA treatment increases atrial cardiomyocyte levels after cardiomyocyte differentiation. What are the differences in the two studies that lead to the somewhat conflicting findings with the role of RA in atrial vs. ventricular cardiomyocyte differentiation.

We agree with the reviewer that there is a substantial body of literature showing that RA can be used in promoting atrial differentiation and we thank the reviewer for pointing our attention in this direction, which helped us to clarify this further.

We are convinced that the observed differences are due to a combination of the differentiation protocol used (shorter and lower dosage of RA; earlier start of treatment) and the mesoderm population we are generating. There is an increasing body of evidence suggesting that the position and timing within the primitive streak limits the fate potential of cardiac progenitors (Lee et al., 2017; Ivanovitch et al. 2021; and others). To explore this further in relation to our data, we integrated streak cells from different stages of development from a single cell mouse dataset of early embryogenesis (Ivanovitch et al. 2021) with our mesoderm (day 1.5) population. This showed that our cells resemble mid- to late primitive streak cells, which eventually give rise to RV and LV

cardiomyocytes and are quite distinct from streak cells of the early bud stage, which are eventually giving rise to atrial cardiomyocytes (revised Supplementary Fig. 4e). In line with this, our mesoderm population expresses high levels of *CYP26A1* which was previously reported to mark a subset of mesoderm preferentially giving rise to ventricular CMs (Lee et al. 2017). Notably, this restricted fate potential was further confirmed in a more recent publication by the same group (Yang et al. 2022). By long exposure to RA, this mesoderm can still be directed into an atrial-like fate, but at low efficiency. As stated further above, this is in line with observations in the literature (Lee et al., 2017), suggesting that the type of mesoderm we are generating allows only a limited amount of atrial-like CMs to arise.

In the manuscript we have replaced the dendrogram showing the integration with the *in vivo* gastrula (former Supplementary Fig. 3b) with a dendrogram of the new integration (revised Supplementary Fig. 4e). We believe the latter is more informative and can better clarify the differences between our observations and previous findings showing that RA treatment increases atrial cardiomyocyte levels after cardiomyocyte differentiation. We have included this in the results section starting on page 6, line 238 and discuss these findings in the discussion part of the manuscript starting on page 13, line 488:

“Integration of this data with a recently published scRNA-seq dataset of a gastrulating mouse embryo showed that these cells correspond to mid/late primitive streak cells, which eventually give rise to the LV and RV CMs in the mouse, as opposed to the early bud stage streak cells, which eventually will give rise to atrial CMs (Supplementary Fig. 4e).”

“Upon exposure of higher dosage of RA for longer periods of time, we did observe some increase in atrial cardiomyocytes, but not comparable to protocols targeted at generating atrial CMs⁵⁰. Differently to our approach, the aforementioned studies have used an alternative cardiac differentiation protocol¹³ most likely resulting in distinct mesodermal progenitor subsets. There is an increasing body of evidence, that the commitment of mesodermal progenitors to form specific cardiac structures is already laid out during primitive streak formation^{4,51}. Mesodermal progenitors that leave the streak early (at mid-late stage), to which the mesodermal progenitors generated with our protocol show resemblance, preferentially give rise to ventricular cardiomyocytes, while later cells (early bud stage) generate atrial cardiomyocytes⁴.”

4. The generated *TBX5* and *NKX2.5* generated lines are nice additions to the paper and the cardiac field. They appear to be knocked into respective loci, potentially disrupting the gene and raising concerns about potential haploinsufficiency issues with data generated from the lines. Can the authors please clarify whether this is indeed the case or not? If so, can they please confirm that these lines do not have issues with differentiation compared to a wild type line, particularly as it pertains to the manuscript with regards to generating specific heart fields and their downstream derivatives? At the very least, this is a limitation of the studies if the lines are haploinsufficient for these factors, especially as previous human studies have shown that haploinsufficiencies of these two factors can cause congenital heart disease.

As the reviewer points out correctly, the reporters were knocked into the respective gene loci causing disruption of gene expression from one copy of the gene. However, as illustrated in the former Supplementary Fig. 2a (revised Supplementary Fig. 2c), this resulted only in a slight reduction in the overall gene expression of *TBX5* and *NKX2.5* as compared to an untargeted HES03 parental line, suggesting partial compensation by the non-targeted copy. Notably, the expression levels of these genes were still above the expression levels observed in an unrelated hiPSC-line suggesting that they were still within a “normal” range. We certainly agree with the reviewer that haploinsufficiency of cardiac TFs has been associated with congenital disease and is a relevant issue.

To account for this, we have included data from the untargeted parental HES03 line as well as an unrelated hiPSC line, which were comparable to the results obtained with our reporter line (former Supplementary Fig. 2a-c; revised Supplementary Fig. 2c-e).

5. It is difficult to appreciate what the clusters represent from the single cell data shown in figures 2, 3, etc. Please provide identification of all cell type clusters and the markers that were used to identify them. For instance, cluster 13 is a proliferative cpc ra- population but it is difficult to appreciate this based on dot plots, heatmaps, etc. shown. A more comprehensive and systematic representation of these clusters would help the readers and authors interpret and appreciate the data.

We apologize for the suboptimal presentation of the data. To clarify this, we have now provide a dotplot with all clusters in the Supplementary Figure 5c, which also highlights the “proliferative counterparts” nature of certain clusters. DEGs for all clusters are provided in Supplementary Data 2. To further verify the proposed identities, we have now integrated our data from day 4.5 with a single cell RNA sequencing dataset of embryonic heart development in the mouse (Tyser et al. 2021) and present the relationship of our clusters to their data in form of a dendrogram. This analysis revealed that our cluster 0 (CPC-RA+) is closely related to their cluster me3, which contains the most advanced cardiac progenitors with high levels of CM marker expression. Clusters 12 and 1 were related to Tyser's cluster me4 corresponding to an earlier stage of FHF-progenitors. Cluster 3 and cluster 17 (CPC-RA-) are related to cluster me7, which shows high expression of canonical SHF-genes, as well as cluster me8, a more pharyngeal SHF-like population. Cluster 14, which we annotated as JCF-like cells, showed close relations with their cluster me5 (annotated as JCF). Notably, our cluster 9, which we previously annotated as pSHF, was closely related to the me5 cluster as well. Through re-analyzing expression profiles, we noticed that cells of cluster 9 indeed express posterior genes (e.g. *HOXB1*), but also low levels of the JCF marker *MAB21L2*. Thus, we are now convinced that these cells rather represent a pre-JCF population instead of pSHF. We changed the annotation accordingly.

6. It would be helpful to look at the expression of RA receptors in the progenitor cells to understand the potential discrepancy between these findings and other results showing that RA generates atrial cardiomyocytes.

We thank the reviewer for this helpful suggestion. Since RA receptors (RXRs) and co-receptors (RARs) are lowly expressed transcripts and suffer from severe “dropouts” using droplet-based sequencing technologies like the one provided by 10x Genomics, we decided for imputing missing values using ALRA, which is an imputation method for single-cell RNA-seq data aiming at distinguishing biological zeros from technical zeros (Lindermann et al., 2022). After performing imputation we observed expression of *RXRA* and *RXRB* but not *RXRG* in CPC-RA+ as well as CPC-RA- without striking differences between the groups (Supplementary Figure 5a). Among the co-receptors, *RARG* expression levels were similar between the groups, while we observed higher expression of *RARB* and, to lesser extent, *RARA* among CPC-RA+ potentially reflecting increased RA signaling activity (Supplementary Figure 5a). Following up on the hypothesis, that differences in atrial potential is largely based on the type of mesoderm generated we have also assessed the expression levels of RA-receptors and co-receptors in our mesoderm in comparison with the single cell mouse dataset of early embryogenesis mentioned in our answer to question 3 (Ivanovitch et al. 2021). In line with previous reports investigating a similar mesoderm population (Lee et al. 2017), we observed expression of all three RA receptors *RXRA*, *RXRB* and *RXRG* in our mesoderm at day 1.5 as well as in both primitive streak populations in the mouse dataset (Supplementary figure 5a). Among the co-receptors *RARB* was neither expressed in the primitive streak populations in the mouse nor in our dataset, while *RARA* was expressed in our mesoderm population as well as in both primitive streak populations in the mouse. Notably, *RARG* which was highly expressed in our day 1.5 mesoderm is

only expressed in the mid/late streak mesoderm (eventually giving rise to ventricular CMs) as opposed to the early bud stage mesoderm (eventually giving rise to atrial CMs). This further highlights the similarities between our mesoderm population and the mid/late primitive streak cells in the mouse likely causative for the limited atrial fate potential observed in our protocol. We now provide these plots as part of the supplementary figure 5a and mention them in the results part of the manuscript on page 7 line 242:

“In line with this, the expression profile of RA receptors (RXRs) and co-receptors (RARs) was comparable between cells at day 1.5 and the mid/late primitive streak cells in the mouse (Supplementary Fig. 5a).”

7. How many scRNA-seq replicates were performed? Also, please show evidence that there were no batch effects, particularly as cells appear to be segregating by samples on UMAP plots in some cases (e.g. Fig. 2b, 2b, etc.). Additionally, it seems that the cell number for each condition are different in the scRNA-seq analysis (Figure 3b), and this may affect the URD analysis. The time/stage should be provided for the URD.

Since we did not perform replicates of the scRNA sequencing experiments we have limited ability to assess batch effect variation. However, all populations from each timepoint were sorted and sequenced simultaneously to limit the influence of technical variation. Since all 4 samples (NKX2.5+ from RA-, NKX2.5+, TBX5+, and NKX2.5+/TBX5+ from RA+) were processed separately, a technical effect would be seen by separation of these 4 samples. Instead, we observe a separation only between treatment (RA+ vs. RA-), which renders substantial technical effects unlikely. To assess this further, we have calculated the Euclidean distance between each cluster and plotted the data as a dendrogram. This reveals that the proximity of clusters is based on biological information (e.g. proliferation) rather than sample, arguing further against the presence of a significant batch effect. We provide this dendrogram now as part of the revised Supplementary Fig. 4 (panel b).

Concerning the limitations of the URD analysis, we kindly refer the reviewer to our answer to the following question 8.

8. In the RA- URD, all of the cells come from a common Mesp1 mesoderm progenitor, but in the RA+ URD, the non-cardiomyocytes and cardiomyocytes appear to come from two Mesp1 mesoderm sources; however, RA isn't added until after the mesoderm stage, which would mean that there should not be two different mesoderm sources for the RA+ as shown in the URD. This difference may reflect the limitation of URD analysis as how tips are assigned may cause different URD tree outcomes. The authors should address and discuss this point and potential limitation, particularly as trajectories are only predictions that need to be functionally validated.

We agree with the reviewer, that the URD analysis suffer from limitations (e.g. the “two sources of mesoderm population” mentioned by the reviewer). We have tried to improve the URD analysis by combining the samples and running it on random subsamples (since the full dataset is too large for the URD pipeline). However, this resulted in variable results, so that we did not feel confident to present a URD tree from the combined analysis. Due to the above mentioned limitations and the fact, that we do not generate novel insights through the URD analysis we have decided to remove the analysis from the manuscript, to gain space for highlighting the new biological data generated as part of the revision process.

9. The results regarding the in vitro model for HLHS are interesting and should be expanded into a main figure. Additional experiments including functional studies on these cells with and without RA

treatment would improve these results. Also providing information as the etiology/genetic mutation for the HLHS may shed light on the results observed from these studies.

We thank the reviewer for appreciating the relevance of our findings from the HLHS disease model and for encouraging us to expand our analysis, which we believe has clearly improved the value of our manuscript. In addition to the inclusion of functional studies as outlined below, we have also used a second control and a second unrelated HLHS hiPSC-line to further validate our findings. Moreover, we now provide information on the genetic background of both lines in the method section of the manuscript (page 14, line 526).

“HLHS-1 carries de novo mutations for DENND5B, SYBU and BAI2, and HLHS-2 carries de novo mutations for MACF1, NFDUFB10, MYRF, AIM1L. Details on de-novo calling, genetic background and clinical phenotype have been previously reported⁴¹.“

In addition, we now also provide stainings for MLC2v as well as cTnT. This data shows that HLHS patient-specific hiPSCs, despite the perturbed transcriptional profile of the CPCs, gave rise to differentiated CMs but at lower efficiency in particular in the CPC-RA- conditions as compared to the control (Fig. 6b, c). In addition, these CMs showed much less, if any, expression of MLC2v, suggesting impaired attainment of ventricular identity (Fig. 6c). To investigate that further, we now also provide functional data on the HLHS-derived CPCs by performing the *ex vivo* injection experiments with CPC-RA+ and CPC-RA- derived from the HLHS-lines. These experiments revealed that the HLHS-derived CPCs still preferentially located to their respective heart compartments (CPC-RA+ to the LV, CPC-RA- to the RV/OFT). However, we observed some cells of both conditions in other compartments of the heart suggesting that the progenitor identity is only partially conserved, in line with our findings of altered expression of heart-field specific markers. In addition, the percentage of cTnT⁺ cells among the HLHS-derived CPC-RA+ was significantly reduced as compared to wildtype CPC-RA+ (43% vs. 78%), suggesting a lower capacity of HLHS-derived CPC-RA+ to differentiate into LV vCMs. Notably, the HLHS-derived CPC-RA- showed a similar amount of cTnT⁺ cells in the injection experiments as compared to wildtype CPC-RA- (29% vs. 34%) suggesting that the differentiation defect observed in the 2D system for this population can be, at least partially, compensated in the *ex vivo* setting.

Taken together, this new set of data suggests that HLHS patient-specific hiPSCs fail to acquire a FHF-like fate at the progenitor level, resulting in perturbed CMs without proper LV vCM identity (also supported by a reduced capacity to form cTnT⁺ cells when injected *ex vivo*). Notably, they also show a defect in aSHF progenitor differentiation resulting in CMs with significant dysregulation of OFT-CM markers. However, this does not result in a reduced number of cTnT⁺ cells in the *ex vivo* injection experiments for this population.

We have included this new data together with the previous data into a new main figure 6 and describe it in the results part of the manuscript starting on page 11, line 419: *“Differentiating two of the previously reported HLHS hiPSC lines (see Methods for details on line ID and genotype) into FHF-like (CPC-RA+) and aSHF-like progenitors (CPC-RA-), we observed, in both conditions, a downregulation of the key CPC marker NKX2.5 (Fig. 6a), which is in agreement with our previous findings⁴¹. This was paralleled by a substantial reduction of TNNT2 levels in CPC-RA+, suggestive of impaired myocytic commitment, and a significant dysregulation of the aSHF-marker FGF10 in the CPC-RA-, together indicating that altered CPC-fate acquisition in HLHS seems to affect both, the FHF-like (CPC-RA+) as well as the aSHF-like (CPC-RA-) population (Fig. 6a). During further differentiation, HLHS patient-specific hiPSCs gave rise to CMs in both conditions as shown by immunofluorescence analysis of cTnT and qPCR for TNNT2, although at lower efficiency (in particular in the CPC-RA- conditions) as compared to the control (Fig. 6b, Supplementary Fig. 10c). In addition, these CMs showed much less, if any, expression of MLC2v, suggesting impaired attainment of ventricular*

identity (Fig. 6c). qPCR analysis confirmed a reduction of ventricular cardiomyocyte transcripts (MYH7, MYL2; Fig. 6d) and revealed increased MYL7/MYL2 and TNNI3/TNNI1 isoform ratio as indicator of cardiac immaturity (Fig. 6d), corroborating our previous findings on maturation defect in HLHS patient-specific CMs⁴¹. Notably, HLHS CMs arising from the CPC-RA+ also displayed lower levels of the LV-specific marker FHL2 (Fig. 6d). At the same time, they exhibited higher expression of genes related to OFT-CMs (RSPO3) rendering their transcriptional profile similar to the patient-specific CMs arising from the CPC-RA-, indicative of a failed commitment to a LV CM fate (Supplementary Fig. 10c). We also observed dysregulation of OFT-CM transcripts (RSPO3) in the CMs derived from the CPC-RA-, which could likely be a consequence of the altered FGF10 level in the progenitors, since this gene is crucially required in the aSHF for proper OFT formation in mice⁴²⁻⁴⁴. The fact that both progenitor pools (CPC-RA+ and CPC-RA-) and derived CMs appear affected is consistent with observations in vivo showing that CMs from the LV as well as the RV of HLHS patient share similar transcriptional defects^{41,45}.

To further evaluate the fate commitment of HLHS-derived CPCs, we injected the cells into the heart region of developing mouse embryos at the cardiac crescent stage and subjected those to whole-embryo ex vivo culture for 24 or 48 hours. Immunofluorescence analysis indicated that HLHS-derived CPCs still contributed preferentially to their respective heart compartments (CPC-RA+ to the LV, CPC-RA- to the RV/OFT). However, their segregation to specific cardiac areas was less defined as compared to the control, with cells from both conditions being also present in other heart compartments (Fig. 6e-g; Supplementary Fig. 10d). In addition, we observed a significant reduction in the percentage of human cTnT+ cells after injection of the CPC-RA+ as compared to the control (43% vs. 78%, $p = 0.0087$), suggesting impaired CM differentiation in line with the findings from the 2D culture (Fig. 6h; Supplementary Fig. 10e; Supplementary Data 8). Interestingly, this was not the case in the embryos injected with the HLHS-derived CPC-RA-, where we detected a similar proportion of human cTnT+ cells as compared to the control (29% vs. 34%) (Fig. 6h; Supplementary Fig. 10e; Supplementary Data 8). This suggests that the differentiation defect seen in the 2D system for this population can be, at least partially, compensated in the ex vivo setting in line with observations in patients where RV cardiomyocytes, although altered on the transcriptional level, form a heart chamber^{41,45}.

Minor:

1. What are the transcriptional differences between the RA+ vs. RA- ventricular cardiomyocyte generated?

Following the reviewer's suggestion, we have performed differential expression analysis between vCMs derived from the CPC-RA- (clusters 1) vs. vCMs derived from the CPC-RA+ (clusters 0, 2, 3 and 4). Overall, the gene expression profiles of vCMs from both conditions were quite similar. This is in line with observations in datasets from human embryonic hearts, which show that differences on the transcriptomic level between ventricular cardiomyocytes located in the LV vs. RV are relatively subtle (Cui et al., 2019; Asp et al. 2019; Supplementary Fig. 8b). Among the transcripts upregulated in vCMs derived from the CPC-RA-, we found genes which show preferential expression in RV myocytes in vivo such as *VCAN* or *GPRIN3* (Supplementary Fig. 8a-b). In vCMs derived from the CPC-RA+ we observed higher expression of genes with preferential expression in LV myocytes such as *GJA1* (Supplementary Fig. 8a-b). We now provide VlnPlots for those genes as part of Supplementary Figure 8. In addition, we provide the complete DEG list as part of Supplementary Data 6.

2. It would be nice to see a positive control for the ITGA8-APC antibody, since the population is very small and does not separate well from the rest of the cells (Figure 2f).

The relatively small separation between the ITGA8- and ITGA8+ population is probably due to the relatively low expression levels. To account for unspecific staining, we performed control

experiments using a matching fluorescently labeled IgG-subtype control (figure for the reviewer). In addition, the new assay on the fate decision of ITGA8+ vs. ITGA8- cells suggests that ITGA8 can indeed be used as marker to enrich for JCF-like cells. Concerning positive controls, we did not have access to a cell line expressing ITGA8, but we contacted the manufacturer of the antibody and received confirmation that it was tested for flow cytometry in a H4 human neuroglioma cell line, which is known to express ITGA8.

(c) Gating strategy for flow cytometry based cell sorting used for defining ITGA8+ cells. **(left)** negative control – cells stained with IgG conjugated with APC. **(right)** cells stained with ITGA8 conjugated with APC. Cells were processed at d4.5.

3. In figure 5, it seems that the author may have not labeled the LV and RV well; for example in Figure 5c (RA- heart), it seems that the positive cells are in LV. The authors should describe and show how they distinguish the LV and RV. The injection experiments are very difficult experiments and more replicates would help solidify findings.

We agree with the reviewer that assigning compartments in these early stages of heart development is challenging in particular in *ex vivo* cultured embryos. This is the reason, why we decided for relatively broad categories when allocating the position of cells. To enable the reader to better assess how we assigned locations, we now provide additional images including section views (revised Supplementary Fig. 10a) as well as a video of a 3D reconstruction with indicated structures (LV, RV, OFT, IFT) as supplementary material (Supplementary Movie 1). In addition, we provide a more detailed description on how we distinguished LV and RV as part of the methods section.

Regarding the embryo replicates, we have performed new injection experiments and added 4 additional embryos, split between the conditions. The results of these new experiments are in line with the previous findings, highlighting the reproducibility of the previous data. Additionally, we now provide also results from new injection experiments of CPCs from HLHS patient-specific hiPSC lines (see main point 9 above).

We have included the additional images as well as the data from the additional embryos into revised Figures 5 and 6 and provide the movie as part of the supplementary files.

Reviewer #2 (Remarks to the Author):

The manuscript from Zawada et al. describes a novel *in vitro* differentiation protocol that allows the generation and isolation of different cardiac progenitor cells from human pluripotent stem cells (hPSC) by modulating Retinoic Acid (RA)-mediated signaling. Several previous studies have shown the key role of RA signaling during heart formation from late primitive streak stage to defining the posterior boundary of heart fields. The authors investigated the influence of RA dosage and timing

on the appearance of early human cardiovascular progenitors by using a growth-factor based protocol for the directed differentiation of hPSC towards cardiomyocytes. scRNAseq analysis of sorted cells demonstrated that cardiac progenitors in differentiation conditions acquire different cell fates depending on absence or presence of RA during early differentiation stages. In addition to the classical first and second heart field progenitors the authors propose that a subset of cells resembled progenitors of the recently discovered juxtacardiac field (JCF) by Tyser et al. (2021) and also Zhang et al (2021), which contribute to epicardium as well as myocardium in mice. -Although not easy to read this study is important as it is a protocol that can be used to differentiate hPSC into various mesodermal cardiac progenitor populations that contribute normally to heart development. The main scientific concern is that some proposed identities are not supported by the markers expressed in the identified clusters.

1. The authors used BMP4, FGF10, CXCR4 and LGR5 as markers of aSHF which are not the best markers for this population. What about TBX1?

We thank the reviewer for acknowledging the relevance of our work and for pointing our attention to the need for improving the justification for cluster identification, which helped us to significantly enhance this aspect of our work.

Following the reviewer's suggestion, we added qPCR data for *TBX1* as well as *WNT5A* showing upregulation of these genes in the CPC-RA- population, in line with the proposed aSHF-like fate (revised Fig. 1d). To further verify the proposed identities, we integrated the cells from our day 4.5 CPC clusters (the non-proliferative clusters) with an *in vivo* single cell RNA sequencing dataset from mouse embryos (Tyser et al., 2021). This analysis revealed that our cluster 0 was closely related to cluster me3 of Tyser et al., which contains the most advanced cardiac progenitors with high levels of CM marker expression. Clusters 12 and 1 were related to Tyser's cluster me4 corresponding to an earlier stage of FHF-progenitors. Cluster 3 and cluster 17 (CPC-RA-) were related to cluster me7, which showed the highest expression of canonical SHF genes as well as cluster me8, a more pharyngeal SHF-like population. Cluster 14, which we annotated as JCF-like cells, showed close relations with Tyser's cluster me5, which they annotated as JCF. Notably, our cluster 9, which we previously annotated as pSHF, was closely related to the me5 cluster as well. Through re-analyzing expression profiles, we noticed that cells of cluster 9 indeed express posterior genes (e.g. *HOXB1*), which led to their original annotation, but also low levels of the JCF marker *MAB21L2*. Thus, we are now convinced that these cells rather represent a pre-JCF population instead of pSHF. We changed the annotation accordingly.

A dendrogram representing the integrated data is now presented in Figure 2 and described in the results part of the manuscript, page 7, line 267.

*"The cluster identities were further evaluated by integrating the cells from the non-proliferative CPC clusters (clusters 0, 1, 3, 9, 12, 14, and 17) with the scRNA-seq dataset of Tyser et al.⁶ covering stages of mouse embryonic heart development that include JCF emergence⁶. This analysis confirmed the FHF- and aSHF-like characteristic of the above-mentioned clusters and showed that clusters 14 and 9 are closely related with the mouse JCF cluster me5, suggesting that both represent JCF-like cells (Fig. 2i). Notably, cluster 9 was mainly derived from *TBX5+/NKX2.5-* CPCs (~75%; Fig., 2h), which is in line with findings in the mouse where the JCF arises from a *TBX5+/NKX2.5-* progenitor population⁶. Cluster 14 was mainly derived from *TBX5+/NKX2.5+* cells and expressed low levels of *NKX2.5* (Supplementary Fig. 5e). This suggests that cluster 9 likely represents an early JCF population and cluster 14 a further committed JCF population that, due to the cardiogenic differentiation conditions, has started to express *NKX2.5* and acquire a cardiomyogenic fate."*

2. After differentiation of hPSC into CMs with RA added at day 4 the authors observed strong upregulation of *TBX5*, which was used as FHF marker. However, studies have shown that *TBX5* is also expressed in pSHF and its activation depends on RA activity (see also below).

As the reviewer correctly points out, *TBX5* is expressed in pSHF as well as FHF. However, the clusters we annotated as FHF-like clusters lack canonical SHF markers such as *ISL1* and (aside from the JCF-like clusters) also posterior markers such as *HOXB1* or *HOXB6*. This notion is further supported by the new integrated analysis of our dataset with the in vivo mouse dataset of Tyser et al. described above, in which our FHF-like clusters show close relation to native FHF clusters and not to native SHF clusters. Moreover, we now confirmed the low abundance of atrial-like cells from CPC-RA+ (which would arise from the pSHF) by immunofluorescent staining and flow cytometry (revised Fig. 1h, i). Notably, prolonged exposure of high dose RA (1 μ M for 10 days), which we included now as a new protocol regimen, led to upregulation of atrial markers and an increased amount of NR2F2⁺ cells in flow cytometry. Notably, even in this condition, we only observed up to 40% NR2F2⁺ cells. (revised Fig. 1h, i). This is in line with observations in the literature (Lee et al., 2017), suggesting that the type of mesoderm we are generating with our protocol (now better characterized in Supplementary Fig. 4e) allows only a limited amount of atrial-like CMs to arise.

3. *THBS4* is used as FHF marker. However, Peisker et al (2022) have shown that fibroblast subpopulation (PDGFb+) expressed *THBS4* after cardiac injury which suggest that this marker is not appropriate to identify FHF cell only. The authors should be more convincing when they define cardiac progenitor identity.

As the reviewer points out correctly, *THBS4* also marks a fibroblast subpopulation in the adult heart and is not entirely specific for FHF cardiac progenitors. However, this is also true for other well-established CPC marker genes such as *ISL1*, which also marks neuronal cells, and *NKX2.5*, which is expressed in adult cardiac fibroblasts. Within the time window relevant for CPC specification during embryonic development *THBS4* appears to be quite specific for FHF-like cells, as shown by scRNA sequencing data from mouse embryos (Figure for the reviewer, Tyser et al., 2021). To further confirm that these cells are not the mentioned subset of fibroblasts, we performed co-expression analysis with *PDGFRB* (Figure for the reviewer). In addition, we now also provide staining for MLC2v showing that, in line with the findings from the qPCR, most of the cells are MLC2v⁺ (revised Fig. 1h), further arguing against a fibroblast-like fate.

[redacted]

(a) Feature plot showing expression levels of smooth muscle cell marker *PDGFRB* and *THBS4* at d1.5 and d4.5 in the UMAP plot shown in Fig. 2f. **(b) (left)** Feature plot showing the expression level of *THBS4* in cluster me4 and me3 corresponding to FHF in the mouse dataset of Tyser et al.⁶ **(right)** UMAP plot showing clustering results of the mouse dataset of Tyser et al.

4. To trace the appearance of distinct cardiac progenitor cells the authors generated a double hESC line expressing eGFP and mCherry under the control of the endogenous *NKX2.5* and *TBX5* locus, respectively. Very useful tool. Using this double transgenic hESC the authors observed that after addition of RA both eGFP+/mCherry+ populations emerged from day 4 combined with a rapid downregulation of *ISL1* and upregulation of cTnT in all populations. The authors should precise whether it is association to premature differentiation of cardiac progenitors or direct inhibition of *ISL1* by RA signaling?

To assess whether the rapid downregulation of *ISL1* in CPC-RA+ is a direct effect of RA itself or due to the rapid CM-fate acquisition of the FHF-like CPCs we looked at the expression dynamics of *ISL1* directly after application of RA in more detail. At day 2 (24h after RA application), we noticed a similar increase in *ISL1* expression in CPC-RA+ and CPC-RA- conditions (former Supplementary Fig. 2a; revised Fig. 1e; Supplementary Fig. 2c). At d3 of differentiation the expression level of *ISL1* still increases in both conditions, although not as strongly in the CPC-RA+ conditions. After that, *ISL1* expression is maintained in the CPC-RA- while getting rapidly downregulated in CPC-RA+. This rapid

downregulation is accompanied by the upregulation of myocardial genes suggesting myocardial fate acquisition. Thus, we are convinced that the observed changes are not caused by a direct effect of RA on *ISL1* expression, but rather reflect the faster CM-fate acquisition observed among progenitors of the FHF. Moreover, the fact that CMs at d30 of differentiation show distinct expression profiles reflecting OFT/RV vs. LV fate suggests that they arise from different progenitor populations arguing against a “premature differentiation” towards the same CM fate. We have included this consideration in the results part of the manuscript on page 4, line 148:

“In the first 24 h after RA application ISL1 expression remained similar in both conditions suggesting that RA does not directly downregulate ISL1 but rather induces changes in the transcriptional network leading to its rapid downregulation (Fig. 1e).”

5. The authors identified cluster 9 (Fig.2) as pSHF, which mainly derived from TBX5-pos/NKX2.5-neg progenitors confirming my point above that TBX5 is also a marker of the pSHF. The authors should discuss this point.

Following the reviewer’s advice on improving our cluster identification we have integrated our day 4.5 CPC clusters (the non-proliferative clusters) with an *in vivo* scRNA sequencing dataset from mouse embryos (Tyser et al., 2021). For a detailed description of the findings, we kindly refer the reviewer to our answer to question 1. Notably, cluster 9, which we previously annotated as pSHF, was closely related to Tyser’s cluster me5, which was annotated by the authors as JCF. In addition, we noticed a very close relationship to our cluster 14, which we annotated as JCF cluster before. Through re-analyzing the expression profile of cells in cluster 9, we noticed that they indeed express posterior genes (e.g. *HOXB1*) which led to their original annotation, but also low levels of the JCF marker *MAB21L2*. Thus, we are now convinced that these cells rather represent a pre-JCF population instead of pSHF. We changed the annotation accordingly. Concerning the expression dynamics of *TBX5* and *NKX2.5* in the JCF populations, we kindly refer to our answer to the following point 6.

6. Interestingly, among CPC-RA+ the authors observed two small clusters (Cluster 14 and 18) that expressed *HAND1* and *MAB21L2*. They affirm that these progenitors resemble to the recently juxtacardiac field (JCF) that maps to the confluence of splanchnic and extraembryonic mesoderm (Tyser et al. 2021). This recent study identified that JCF cells express *Hand1*. However, the Me5 cluster identifies as JCF does not expressed *NKX2.5* (Me5 cluster in figure 2; Tyser et al. 2021). In the current study CPC-RA+ that resembles to JCF derived from the *NKX2.5*-positive cells. This discrepancy should be addressed by the authors.

We agree with the reviewer that a substantial proportion of cells in cluster 14 stem from the *NKX2.5*+ population. However, the expression level of *NKX2.5* in these cells is lower as compared to the FHF-like clusters 0 and 1 (revised Supplementary Fig. 5d), which is in line with observations by Tyser et al., who reported lower but present expression of *NKX2.5* in the cluster me5 (their JCF-cluster). Based on our new findings, in particular the integration with the *in vivo* mouse dataset, we now assume that cluster 9 represents pre-JCF cells. Notably, the *NKX2.5* expression level in these cells is even lower and this cluster mainly stems from *TBX5*+/*NKX2.5*- cells. This suggests that cluster 14 could represent JCF cells that are further differentiated and have, due to the cardiogenic differentiation conditions, started to express *NKX2.5* as sign of their cardiomyogenic commitment. We discuss this now in the results part of the manuscript on page 7, line 267:

“The cluster identities were further evaluated by integrating the cells from the non-proliferative CPC clusters (clusters 0, 1, 3, 9, 12, 14, and 17) with the scRNA-seq dataset of Tyser et al.⁶ covering stages of mouse embryonic heart development that include JCF emergence⁶. This analysis confirmed the FHF- and aSHF-like characteristic of the above-mentioned clusters and showed that clusters 14 and 9

are closely related with the mouse JCF cluster me5, suggesting that both represent JCF-like cells (Fig. 2i). Notably, cluster 9 was mainly derived from TBX5+/NKX2.5- CPCs (~75%; Fig. 2h), which is in line with findings in the mouse where the JCF arises from a TBX5+/NKX2.5- progenitor population⁶. Cluster 14 was mainly derived from TBX5+/NKX2.5+ cells and expressed low levels of NKX2.5 (Supplementary Fig. 5e). This suggests that cluster 9 likely represents an early JCF population and cluster 14 a further committed JCF population that, due to the cardiogenic differentiation conditions, has started to express NKX2.5 and acquire a cardiomyogenic fate.”

7. Cardiac progenitors are multipotent and can differentiate into not only CMs but also into smooth muscle and endothelial cells during heart formation. The current study suggests that JCF progenitors are bipotent since they give rise to CMs and epicardial cells. Cell bipotency of JCF needs to be addressed. The authors identified ITGA8 as marker that allows isolation of the JCF cells. This marker can be used to perform clonal analysis.

We thank the reviewer for this suggestion, which helped us to better address the issue of potency within the JCF-like cell population. A “bona fide” clonal analysis was unfortunately not feasible, since the cells did not differentiate properly when seeded at too low densities. To circumvent this issue, we sorted hiPSCs that constitutively express GFP through an insertion into the AAV locus into an ITGA8⁺ and ITGA8⁻ population at d5 of differentiation and mixed them with non-labeled hiPSCs from the same differentiation timepoint approximating 5 eGFP⁺ cells with 20000 eGFP⁻ cells per well of a 12 well chamber slide. This seeding density allowed us to assess clones derived from a single GFP⁺ cell for fate decision. In newly established differentiation conditions enabling the occurrence of epicardial cells as well cardiomyocytes simultaneously, the ITGA8⁺ cells showed an increased potential to give rise to epicardial cells, while maintaining the potential for cardiomyocyte differentiation. While the vast majority of eGFP⁺/ITGA8⁺ clones differentiated into one lineage, around 10% of them gave rise to both epicardial cells and cardiomyocytes (Fig. 3m, Supplementary Data 4). In part, this could be due to technical aspects since the differentiation occurred in clusters where the surrounding cells could have influenced fate decision. The epicardial differentiation potential observed in the ITGA8⁻ fraction is most likely due to the imperfect overlap of ITGA8 and MAB21L2 expression (Fig. 3b) resulting in JCF-like progenitors also in the ITGA8⁻ fraction. Thus, ITGA8 can be used to enrich/deplete JCF-like cells but not for a black and white sorting.

These results have been included in Figure 3k-m and are now discussed in the results part of the manuscript. Starting on page 8, line 300:

“To assess the fate potential of the ITGA8 population at a clonal level, we took advantage of hiPSCs constitutively expressing eGFP and sorted ITGA8⁺ and ITGA8⁻ cells emerging at d5 of differentiation in presence of RA (1 μM, 4d). We plated these cells at very low density with equivalent, unsorted d5 CPCs generated from un-labeled hiPSCs (approx. 5 eGFP⁺ cells with 20000 eGFP⁻ cells/well of a 12 well chamber slide) and differentiated them for 10 days in culture conditions that allow for simultaneous generation of epicardial cells and CMs (Fig. 3k). As compared to the eGFP⁺/ITGA8⁻ fraction, eGFP⁺/ITGA8⁺ cells showed a higher propensity to give rise to epicardial cells, while maintaining the ability for CM differentiation (Fig. 3l, m; Supplementary Data 4). While the vast majority of eGFP⁺/ITGA8⁺ clones differentiated into one lineage, around 10% of them gave rise to both epicardial cells and CMs (Fig. 3m, Supplementary Fig. 6d, Supplementary Data 4). This data provides further proof that a bipotent JCF progenitor pool giving rise to CMs and epicardial cells also exists during human cardiogenesis and can be enriched using ITGA8 as a membrane marker.”

8. To investigate the lineage commitment and functionality of in vitro derived cardiovascular progenitors the authors injected CPC-RA⁺ or CPC-RA⁻ into cardiac crescent region of mouse embryos. Although I do agree with conclusion that the identity of CPC-RA⁺ and CPC-RA⁻ at day 4.5

are already committed to contribute to their respective cardiac structure the authors should performed further experiment. We know that cardiac crescent contributes only to LV. So, to convince that CPC-RA+ cannot contribute to OFT/RV injection should be also performed in linear heart tube of mouse embryos.

We apologize for the confusion about the exact injection location. To put it more precisely, we injected cells into the heart region of developing mouse embryos at the cardiac crescent stage. At this stage, the structures within the embryo are so small and, in absence of a genetic marking, it is impossible to distinguish the cardiac crescent from the directly posteriorly located SHF-region. Thus, our injections were not targeting a specific heart field location but rather the whole heart-forming region. We have now clarified this issue in the revised manuscript (page 10, lines 383) and have provided a detailed description of our injection procedure in the Method section starting on page 22, line 805.

“...injected into the heart region of dissected mouse embryos at the cardiac crescent stage. At this stage, structures within the mouse embryo are small and FHF and SHF regions appear in the close proximity enabling targeting the whole heart-forming region with a single injection.”

In general, we agree with the reviewer that the exact location of the cell injection in the early heart region could “per se” instruct the future allocation of CPCs to specific heart structures. However, CPC-RA+ and CPC-RA- cells were injected into the exact same region of the early embryo and still allocated to distinct, different compartments of the looping heart, suggesting that their positioning/structure contribution is indeed dictated by their fate commitment. To increase the solidness of our data, we have now performed additional injection experiments and increased the number of injected embryos in both conditions (revised Fig. 5f). The newly obtained results are in line with our previous findings. Moreover, we provide now also new results on injection of CPCs from HLHS patient-specific hiPSC lines indicating that “perturbed/diseased” CPCs have a “less defined” segregation to specific cardiac areas compared to the healthy CPCs, although still contributing preferentially to their respective heart compartments (CPC-RA+ to the LV, CPC-RA- to the RV/OFT), as illustrated in the revised Fig. 6e-g.

Minor points:

Line 258: The CMs separated into five main groups. However, there are six groups in the figure. Can the authors describe also cluster 14?

We apologize for the confusing presentation of data. Cluster 14 is a proliferating CM cluster and was excluded from the subsequent plots. To improve the data presentation, we now included a dot plot with all clusters which also highlights better the “proliferative counterparts” nature of certain clusters (revised Supplementary Fig. 7b).

Reviewer #3 (Remarks to the Author):

In their manuscript “Retinoic acid signaling modulation guides in vitro specification of human heart field-specific progenitor pools” Zawada and colleagues ask the important question of whether RA signaling, when modulated in a dose- and time-specific manner, can control the formation of FHF versus aSHF progenitor populations from pluripotent stem cells. They show that an early pulse of RA, at the stage of emerging mesoderm cells, generates distinct CPCs from non-RA exposed cultures. These CPCs subsequently give rise to differentiated cells,

cardiomyocytes and non-cardiomyocytes, that are different depending on whether they come from an -RA or +RA CPC. Lastly they show that CPC-RA and CPC+RA behave differently when transplanted into the early crescent stage mouse embryo, and that the new protocol to generate FHF CPCs can be used to establish an HLHS disease model.

Identifying new approaches to generate the heterogeneity of early CPC populations from PSCs is a highly relevant question with high impact for the field. Inducing RA signaling at a very early stage (as opposed to slightly later stages to form atrial cells) in order to generate a FHF CPC population has not been systematically tested to date, and represents an intriguing hypothesis. The paper is well written and the data are presented and described clearly. The experiments performed are well rationalized and appropriate for the question. The main concern is that the original hypothesis – that RA early during differentiation generated aSHF cells – is not fully supported by the data, or would need to be supported by additional analysis to be convincing.

We thank the reviewer for highlighting the relevance of our study, the acknowledgment of our experimental design and suggested additional analysis. Concerning the main message of our manuscript, we hypothesize that RA early during differentiation generates FHF-like CPCs by suppressing an aSHF-like fate. We assume this is what the reviewer meant, but we just wanted to clarify.

Main comments:

1. Differentiation protocol and analysis of the role of RA during early cardiac development: The manuscript carefully sets the stage for the RA analysis characterizing the early CPCs in different protocols (different RA dosage and exposure time). Some comments for this introductory analyses are:

1.1. Given the well-established notion that a -RA differentiation protocol generates ventricular CMs, including left ventricular CMs, it is surprising that no FHF markers are detected in the -RA cells at the progenitor stage.

We agree with the reviewer that there are several directed cardiac differentiation protocols that generate ventricular CMs from hPSCs with high efficiency and do not rely on the addition of RA. The same accounts for our RA- differentiation condition, which results in the efficient generation of ventricular cardiomyocytes. As a general note, we are convinced that distinguishing LV- and RV- cardiomyocytes based on transcriptional differences is challenging, since the marker expression changes over the time course of development and direct comparisons of ventricular myocytes from LV and RV in datasets from human embryonic hearts yield only limited numbers of differentially regulated genes with low log2fold changes, indicative of relatively similar expression profiles (Cui et al., 2019; Asp et al. 2019). In addition, these data show that markers such as *CITED1* or *CCK*, which can be used to distinguish LV and RV in the mouse, are not expressed in humans (figure for the reviewer). This was also the reasoning behind implementing the *ex vivo* mouse embryo injections to be able to distinguish an LV- vs. RV-fate not based solely on the transcriptional profile of *in vitro* differentiated CMs but rather through specific contribution and positional information in a developing heart. Nevertheless, we have followed the reviewer's suggestion and included additional markers with preferential expression in LV vs. RV cardiomyocytes (see our answer to question 1.3).

(a) Violin plots showing expression of CCK, CITED1, PLN in the indicated clusters at d30. UMAP plot shown in Fig. 4b. **(b)** Violin plots showing expression of CCK, CITED1, PLN in indicated heart compartments at different stages of fetal human heart development from Cui et al., (2019). LV = left ventricle; RV = right ventricle; LA = left atria; RA = right atria; 5w = 5th week of gestation. CM = cardiomyocytes; for this plot only data from early fetal hearts (5-7th weeks of gestation) were considered.

In addition, it is important to highlight that our differentiation protocol is distinct from other protocols that rely mainly on the manipulation of Wnt-signaling to generate FHF-like CPCs (e.g. Zhang et al., 2019). Concerning our specific growth-factor based differentiation (CPC-RA-), the lack of FHF markers is in line with other reports in the literature that use similar induction cues (e.g. Lee et al., 2017; Yang et al., 2022). Thus, we do not see any discrepancy between our findings and the reports in the literature in that respect.

To further verify the proposed identities of our CPC populations, we have now also integrated the cells from our day 4.5 CPC clusters (excluding the proliferative clusters) with an *in vivo* scRNA sequencing dataset from mouse embryos (Tyser et al., 2021). This analysis revealed that our cluster 0 (CPC-RA+) was closely related to Tyser's cluster me3, which contains the most advanced cardiac progenitors with high levels of CM marker expression. Clusters 12 and 1 were related to Tyser's cluster me4 corresponding to an earlier stage of FHF-progenitors. Cluster 3 and cluster 17 (CPC-RA-) were related to clusters me7, which showed the highest expression of canonical SHF genes, as well as cluster me8, a more pharyngeal SHF-like population. We have included this new analysis in the revised Fig. 2, panel i, and in the Results part of the manuscript (starting on page 7, line 267).

"The cluster identities were further evaluated by integrating the cells from the non-proliferative CPC clusters (clusters 0, 1, 3, 9, 12, 14, and 17) with the scRNA-seq dataset of Tyser et al.⁶ covering stages

of mouse embryonic heart development that include JCF emergence⁶. This analysis confirmed the FHF- and aSHF-like characteristic of the above-mentioned clusters and showed that clusters 14 and 9 are closely related with the mouse JCF cluster *me5*, suggesting that both represent JCF-like cells (Fig. 2i). Notably, cluster 9 was mainly derived from *TBX5*⁺/*NKX2.5*⁻ CPCs (~75%; Fig. 2h), which is in line with findings in the mouse where the JCF arises from a *TBX5*⁺/*NKX2.5*⁻ progenitor population⁶. Cluster 14 was mainly derived from *TBX5*⁺/*NKX2.5*⁺ cells and expressed low levels of *NKX2.5* (Supplementary Fig. 5e). This suggests that cluster 9 likely represents an early JCF population and cluster 14 a further committed JCF population that, due to the cardiogenic differentiation conditions, has started to express *NKX2.5* and acquire a cardiomyogenic fate.”

1.2 Bulk PCR analysis can be biased by varying efficiencies or varying speed of differentiation in different protocols. A pan CPC marker such as *NKX2.5* should be included in the bulk PCR analysis at the CPC stage to provide a sense for the amount of CPCs present in each condition. Additionally, *TBX1* as an important aSHF marker and should be included in the analysis.

Following the reviewer’s suggestion, we have now added *NKX2.5* to the qPCR panel presented in the revised supplementary figure 1 (panel a). This data shows that the expression level of *NKX2.5* is very similar at day 5 (CPC-stage) between the different RA regimes, arguing against a bias introduced by differences in differentiation speed or efficiency. We have also added qPCR results for *TBX1* as well as *WNT5A* to further confirm the aSHF-like fate (revised Fig. 1d).

Accordingly, the manuscript text has been edited as follows:

Starting on page 4, lines 140: “The expression level of the pan cardiac progenitor marker *NKX2.5* was comparable between the different treatments reflecting a similar efficiency in cardiac progenitor specification (Supplementary Fig. 1a).”

Page 4, line 132: “Concomitant upregulation of *BMP4*, *FGF10*, *CXCR4*, and *LGR5* along with *TBX1* and *WNT5A* at day 5 suggested that these cells adopted a fate resembling cardiovascular progenitors of the aSHF (Fig. 1c, d)”

1.3 Given the differential contributions of FHF and aSHF to the ventricular chambers, markers for left and right ventricular cardiomyocytes would be informative at day 30 (*HAND1*, *CCK*, *PLN*, *CITED1*).

Following the reviewer’s suggestion, we have investigated genes that show preferential expression in LV vs. RV cardiomyocytes during human heart development (Cui et al., 2019; Asp et al. 2019). As mentioned further above, *CITED1* and *CCK* were neither expressed in our dataset nor in the dataset from human embryonic hearts (figure for the reviewer). *PLN* showed similar expression levels between the LV-like clusters 0, 2, and 4, as well as the RV-like cluster 1. This matches with the expression level distribution between LV- and RV-CMs observed in the dataset from human embryonic hearts (figure for the reviewer). The expression levels of *TBX5* and *GJA1* were significantly higher in the LV-like clusters (0,2,4) as compared to the RV-like cluster (1), while the expression level for *VCAN* and *GPRIN3* was higher in the RV-like cluster (1) as compared to the LV-like clusters (0,2,4) (Supplementary Figure 8a). These findings were in line with the observations in the human embryonic heart (Supplementary Figure 8b). Notably, the expression level of *HAND1*, which shows preferential expression in LV vCMs in the dataset from human embryonic hearts (Supplementary Figure 8b) showed the highest expression in the OFT-CM cluster 6 (supplementary figure 8a). This is in line with reports in the literature indicating that *HAND1* is not only expressed in LV vCMs, but also in cardiomyocytes of the OFT and AV-canal (Meilhac and Buckingham, 2018; de Soysa et al., 2019). In addition, *HAND1* is dynamically regulated throughout differentiation (Supplementary Figure 8b) and is almost absent in adult cardiomyocytes (Litviňuková et al., 2020) rendering it a suboptimal candidate for distinguishing LV vs. RV fate in humans.

(a) Violin plots showing expression of CCK, CITED1, PLN in the indicated clusters at d30. UMAP plot shown in Fig. 4b. **(b)** Violin plots showing expression of CCK, CITED1, PLN in indicated heart compartments at different stages of fetal human heart development from Cui et al., (2019). LV = left ventricle; RV = right ventricle; LA = left atria; RA = right atria; 5w = 5th week of gestation. CM = cardiomyocytes; for this plot only data from early fetal hearts (5-7th weeks of gestation) were considered.

We have included data for *TBX5*, *GJA1*, *VCAN*, *GPRIN3* and *HAND1* into the new supplementary figure 8a and discuss this now in the results part of the manuscript:

Starting on page 9, lines 326: *“The transcriptional profiles of ventricular-like CMs derived from the CPC-RA+ (clusters 0, 2, 3 and 4) and CPC-RA- (cluster 1) were overall quite similar, in line with data from human in vivo embryonic hearts showing that transcriptional differences between ventricular CMs located in the LV vs. RV are subtle^{26,27} (Fig. 4d, e; Supplementary Data 6). Among the genes that show a preferential expression in either the LV or the RV in vivo²⁶, we observed higher levels for *TBX5* and *GJA1* in the LV-like clusters (0, 2, 3, 4) as compared to the RV-like cluster (1), while the expression levels for *VCAN* and *GPRIN3* was higher in the RV-like cluster (1) as compared to the LV-like clusters (0, 2, 3, 4) (Supplementary Fig. 8a, b). Notably, the expression level of *HAND1* was highest in the OFT-CM cluster 6 (Supplementary Fig. 8a). This is in line with reports in the literature showing that *HAND1* is expressed not only in LV CMs, but also in OFT and AVC CMs^{1,17}.”*

1.4 Related, the manuscript states that FHL2 is increased in a dose-dependent manner – the data shown do not support that (Fig. 1g).

The reviewer is correct in pointing out that the statement “*FHL2* is increased in a dose-dependent manner” is not correct, since *FHL2* expression upon high dosage of RA (1 μ M for 4 days) is slightly lower as compared to the lower dosage (0.5 μ M for 4 days). We have now revised our statement. Moreover, we have also investigate this aspect further by including another RA application scheme (1 μ M for 10 days). Applying this scheme, it became evident that with low dosage of RA longer treatment leads to an increase in *FHL2* expression, while higher and longer exposure to RA results indeed again in a decrease of *FHL2* expression. This decrease was accompanied by an increase in atrial markers expression (revised Fig. 1g) as well as an increase of NR2F2⁺ cells in flow cytometry (revised Fig. 1i).

We have included the new results in the results part of the manuscript and state now more clearly that there is a defined time and dosage window for RA that leads to an increase in FHF progenitors and subsequent (LV) ventricular CMs:

Starting on page 4, lines 165: *“At low dosage of RA (0.5 μ M), FHL2 expression increased with time of treatment, but it declined again at higher dose (1 μ M) and prolonged treatment. We observed similar dynamics for the ventricular marker MYL2. Genes associated with CMs of the OFT showed an inverse relation to the FHL2 expression pattern with the lowest expression levels at 0.5 μ M RA for 4 days (Fig. 1g). Atrial CM markers increased only marginally at low dosage of RA, but a more substantial expression was observed upon higher dose and longer treatment (Fig. 1g, Supplementary Fig. 1d). Taken together, these results suggest that the occurrence of FHF-like progenitors is tightly controlled by timing and dosage of RA addition.”*

1.5 The most reliable metric to characterize CPC populations is to interrogate their progeny, which this study does well via multiple assays (EP analysis, bulk PCR). Given the importance of this question for the remainder of the study, differentiated cardiomyocytes at day 30 should be stained (immunohistochemistry) for MLC2v and MLC2a or NR2F2, markers that reliably identify differentiated ventricular and atrial cardiomyocytes respectively during PSC differentiation.

We thank the reviewer for complementing on our assays. Following the reviewer’s suggestion, we have included immunofluorescent stainings for MLC2v and MLC2a (revised Fig. 1h) and flow cytometry quantification for NR2F2 (revised Fig. 1i). The results from these new analyses confirmed our previous findings from the bulk PCR as well as the EP analysis. We also extended these assays to the cells derived from the new RA application scheme (1 μ M RA for 10 days). Here we observed an increase in NR2F2⁺ cells in flow cytometry (revised Fig. 1i). Notably, even in this condition, we only observed up to 40% NR2F2⁺ cells. This is in line with observations in the literature (Lee et al., 2017), suggesting that the type of mesoderm we are generating allows only a limited amount of atrial-like CMs to arise.

1.6 From most of the data presented, it looks like all of the cardiomyocytes in the +RA conditions express TBX5 at day 30 (transcript, protein and reporter), which may suggest that at least some cells have atrial identity.

The reviewer correctly states that in the RA+ conditions the majority of cardiomyocytes express *TBX5* at day 30 and that *TBX5* is expressed by atrial myocytes. However, it is also expressed by LV myocytes in human embryonic hearts on the transcript level (Cui et al., 2019; see also answer to the question 1.3) as well as on the protein level (Hatcher et al., 2000). To further confirm that the TBX5+ CMs are indeed LV CMs and not atrial CMs, we have now added (following the reviewer’s suggestion further above) flow cytometry data for the atrial marker NR2F2 as well as staining for MLC2v and MLC2a confirming a ventricular identity in the vast majority of the cells arising from the CPC-RA+ at day 30 (revised Fig. 1h-i).

1.7 It appears that both markers (NKX2-5, TBX5) have been targeted with the same strategy. It is not clear why GFP is cytoplasmic and mCherry is nuclear in the reporter live analysis (Fig. 1l)?

We apologize for the incomplete representation of the targeting strategy. mCherry is fused to H2B causing the nuclear localization. We have clarified this in the manuscript as follows:

Starting on page 5, line 201: "To track the appearance of distinct cardiovascular progenitor pools and enable enrichment of specific subpopulations for in-depth analysis, we generated a double reporter human embryonic stem cell (hESC) line expressing eGFP and nuclear mCherry (H2B-mCherry) under the control of the endogenous NKX2.5 and TBX5 locus, respectively (ES03 TN cell line, Fig. 2a)."

2. Single cell RNAseq analysis on +/- RA treated cells:

Taking advantage of the unbiased single cell analysis approach to query identity and differences in different CPC populations during early development is a well-rationalized and promising approach. Some questions or concerns were noted as follows:

2.1 The clustering analysis demonstrates distinct separation between the time points as well as the RA treatment conditions, except for the endoderm cells (Fig. 1b). This is rather surprising, given that cardiac progenitor cells from both the -RA and +RA protocol would be expected to be more similar to each other than to the mesoderm cells. Along similar lines, based on select marker expression and subsequent cluster annotation, there seem to be FHF and SHF populations in both conditions that should locate more closely to each other. There are clusters (17 and 12) that exist in both populations yet they are located far apart. Collectively this indicates the presence of batch effects that obstruct the true understanding (similarities and differences) of some of these cell populations. The manuscript should better explain how the data from the different samples were integrated, and how batch effects were corrected in the analysis. If no batch correction was performed, this should be corrected in the analysis.

We thank the reviewer for bringing this point to our attention. Since we did not perform replicates of the scRNA sequencing experiments we have limited ability to assess batch effect variation. However, all populations from each timepoint were sorted and sequenced simultaneously to limit the influence of technical variation. Since all 4 samples (NKX2.5+ from RA-, NKX2.5+, TBX5+, and NKX2.5+/TBX5+ from RA+) were processed separately, a technical effect would be seen by separation of these 4 samples. Instead, we observe a separation between treatment (RA+ vs. RA-) while the three samples from the RA+ conditions do not separate, which renders substantial technical effects unlikely. The fact, that cluster 17 and 12 are spread apart is based on the fact, that the clusters were calculated on PCA and not on UMAP reduction. We do this to maintain additional information about similarities/dissimilarities of cells which are not represented in the UMAP plot.

To assess the presence of a potential batch effect further, we have calculated the Euclidean distance between each cluster and plotted the data as a dendrogram. This reveals that the proximity of clusters is based on biological information (e.g. proliferation, celltype) rather than sample, arguing further against the presence of a significant batch effect. This dendrogram also shows, that the separation between day 1.5 mesoderm, endoderm, and CPCs is larger than between the CPCs. We provide this dendrogram now as part of the revised supplementary figure 4 (panel b).

2.2 The gene expression analysis in Fig. 1c is helpful to understand the different clusters. All of the identified clusters should be included in this analysis, not just what seems a selective choice for each sample.

We thank the reviewer for this suggestion that helped to improve the readability of the data. We now include all clusters in the dotplot presented in the revised Supplementary Fig. 5c, which also aids in highlighting the “proliferative counterpart” nature of certain clusters (see also answer to question 2.4).

2.3 The D4.5 samples for this analysis were sorted. Feature plots of *NKX2-5* and *TBX5* of the UMPA in 1b should be included, either in main or supplementary figures to illustrate the success of this approach. It is not entirely clear why sorting was necessary, as the analysis of the resulting cell types of the -RA and +RA protocols, and thus the rationale for looking at the CPCs in each protocol were performed on non-sorted differentiations (all of Figure 1).

We agree with the reviewer that sorting of the CPC populations is not strictly necessary, which we also took advantage of by extending our protocol to other cell lines including patient specific hiPSCs. Initially we assumed that the flow cytometry-based cell sorting would aid in further enriching certain subpopulations (similar to the findings by Zhang et al., 2019) which, as the reviewer correctly points out, turned out not to be so pronounced in our differentiation conditions. However, since we are convinced that it does add information to the analysis of the single cell RNA sequencing data (e.g. the pre-JCF cluster 9 arising mainly from *TBX5*+/*NKX2.5*- cells) we decided to present it in its current form.

Following the reviewer’s suggestion, we now provide Feature plots for *NKX2.5* and *TBX5* in the sorted populations based on the UMAP plot from Figure 2f in the revised Supplementary figure 4a. *TBX5* is a transcription factor with rather low expression levels and suffers from severe “dropouts” using droplet-based sequencing technologies like the one provided by 10x Genomics, which was used to create this dataset. To explore this further, we imputed missing values using ALRA, which is an imputation method for single-cell RNA-seq data aiming at distinguishing biological zeros from technical zeros (Lindermann et al., 2022). After performing imputation, *TBX5* expression was more homogenously expressed in the respective subpopulations supporting the hypothesis that the sparse *TBX5* expression is largely due to technical “dropouts” (figure for the reviewer).

(a) Feature plots showing expression levels of *TBX5* and *NKX2.5* in the indicated samples at d1.5 and d4.5 in the UMAP plot shown in Fig. 2f. N = *NKX2.5* (eGFP+); T = *TBX5* (mCherry+). **(b)** Feature plots showing expression levels of *TBX5* and *NKX2.5* in the indicated samples at d1.5 and d4.5 in the UMAP plot shown in Fig. 2f. N = *NKX2.5* (eGFP+); T = *TBX5* (mCherry+). Missing values were imputed using ALRA. **(c)** Representative immunofluorescence images of obtained aggregates of indicated

samples one day after sorting. Cells were sorted based on TBX5 (mCherry) or NKX2.5 (eGFP) expression. T = TBX5, N = NKX2.5.

2.4 The manuscript states several times that distinct clusters have a ‘proliferative counterpart’, but it is not explained how this conclusion has been taken, and only cell cycle status of the clusters is shown to support this statement (Fig. S4a).

We apologize for the suboptimal representation of this piece of data. We now have added the dotplot which aids in highlighting the “proliferative counterpart” nature of certain clusters (Supplementary Fig. 5c).

3. Identification of the JCF population:

The study next delves deeper into a population that may represent the newly discovered JCF, and finds that ITGA8 may identify such a population. IGF8+ cells at day 4 have the potential to form both cardiomyocytes and epicardium, which may constitute a new approach to study the JCF.

3.1 WT1 should be included in the epicardial analysis, as it is a more epicardium-specific marker compared to the markers that are currently included (ZO1, CK18 and TCF21).

Following the reviewer’s suggestion, we have added immunofluorescent staining for WT1 to the panel of epicardial markers (revised Fig. 3j; 3l).

3.2 Ideally one should compare another CPC population from the same differentiation for its potential to form cardiomyocytes and epicardium, to show that the ability to do so is to some extent specific to the JCF, and not broadly due to RA-exposure of mesoderm (as epicardium protocols are all based on addition of RA as well). The available reporter cell line should make this a feasible task.

We thank the reviewer for this suggestion, which helped us to strengthen the data on the JCF-like population significantly. To address the point raised, we sorted hiPSCs that constitutively express eGFP through an insertion into the AAV locus into an ITGA8⁺ and ITGA8⁻ population at day 5 of differentiation and mixed them with non-labeled hiPSCs from the same differentiation timepoint approximating 5 eGFP⁺ cells with 20000 eGFP⁻ cells per well of a 12 well chamber slide. This seeding density allowed us to assess clones derived from a single eGFP⁺ cell for fate decision. In newly established differentiation conditions enabling the occurrence of epicardial cells as well cardiomyocytes simultaneously, the ITGA8⁺ cells showed an increased potential to give rise to epicardial cells, while maintaining the potential for cardiomyocyte differentiation. While the vast majority of eGFP⁺/ITGA8⁺ clones differentiated into one lineage, around 10% of them gave rise to both epicardial cells and cardiomyocytes (Fig. 3m, Supplementary Data 4). In part, this could also be due to technical aspects since the differentiation occurred in clusters where the surrounding cells could have influenced fate decision. The epicardial differentiation potential observed in the ITGA8⁻ fraction is most likely due to the imperfect overlap of *ITGA8* and *MAB21L2* expression (Figure 3b) resulting in JCF-like progenitors also in the ITGA8⁻ fraction. Thus, ITGA8 can be used to enrich/deplete JCF-like cells but not for a black and white sorting.

These results have been included in Figure 3k-m and are now discussed in the results part of the manuscript:

Starting on page 8, lines 300: *“To assess the fate potential of the ITGA8⁺ population at a clonal level, we took advantage of hiPSCs constitutively expressing eGFP and sorted ITGA8⁺ and*

ITGA8⁺ cells emerging at d5 of differentiation in presence of RA (1 μM, 4d). We plated these cells at very low density with equivalent, unsorted d5 CPCs generated from un-labeled hiPSCs (approx. 5 eGFP⁺ cells with 20000 eGFP⁻ cells/well of a 12 well chamber slide) and differentiated them for 10 days in culture conditions that allow for simultaneous generation of epicardial cells and CMs (Fig. 3k). As compared to the eGFP⁺/ITGA8⁻ fraction, eGFP⁺/ITGA8⁺ cells showed a higher propensity to give rise to epicardial cells, while maintaining the ability for CM differentiation (Fig. 3l, m; Supplementary Data 4). While the vast majority of eGFP⁺/ITGA8⁺ clones differentiated into one lineage, around 10% of them gave rise to both epicardial cells and CMs (Fig. 3m, Supplementary Fig. 6d, Supplementary Data 4). This data provides further proof that a bipotent JCF progenitor pool giving rise to CMs and epicardial cells also exists during human cardiogenesis and can be enriched using ITGA8 as a membrane marker.”

4. Characterization of differentiated cells in -RA and +RA protocols:

The sorting followed by differentiation represents a rigorous and informative approach to test the differentiation potential of the CPCs in the -RA and +RA protocols, and this approach aligns well with the single cell transcriptomic data earlier in the paper. Most of my comments address data analysis strategies and whether single cell RNAseq analysis is sufficient to draw the conclusions drawn here.

4.1 The rationale for including the D1 and D4.5 data in this analysis is unclear, as this comparison does not add any information on the day 30 cells themselves. Rather, day 30 cells should be analyzed separately, as this will likely provide a better understanding of the heterogeneity of these cells. Without this, the current data display makes it challenging to truly understand the cellular heterogeneity of the cells derived from the distinct CPCs/protocols. The same batch correction comment noted in point 1 should be applied here as well.

We thank the reviewer for raising this point. The presentation of the data in the current form was to allow for appreciation of the development of the cells from day 1.5 to day 30. However, we do agree with the reviewer, that this is unnecessarily complicated and might obscure smaller heterogeneity aspects in the day 30 dataset. Thus, we followed the reviewer’s suggestion and performed analysis of the day 30 dataset separately. While the overall segregation into clusters was largely maintained, it indeed improved resolving the remaining heterogeneity. We now observed separation of CPC-RA-derived CMs into an RA- OFT-like cluster (*MYL2* low) and an RA- vCM-like cluster (*MYL2* high). The *MYOZ2*-enriched CMs, which were part of a mixed cluster before, now mingle with other ventricular myocytes, both in LV-like clusters (0,2,3,4) and the RV-like cluster (1). In addition, a new cluster 14 formed (that before was belonging to cluster 26) which is characterized by enrichment of *CPNE5* and *IGFBP5*. This cluster could represent a transitional population of nodal cells (Goodyer et al., 2019; Wiesinger et al., 2022), which we observed in low abundance also in the electrophysiological analysis.

We have replaced the UMAP and related plots in revised Figure 4 with the new results and discuss them now in the results part of the manuscript:

Starting on page 9, line 326: *“The transcriptional profiles of ventricular-like CMs derived from the CPC-RA+ (clusters 0, 2, 3 and 4) and CPC-RA- (cluster 1) were overall quite similar, in line with data from human in vivo embryonic hearts showing that transcriptional differences between vCMs located in the LV vs. RV are subtle^{26,27} (Fig. 4d, e; Supplementary Data 6). Among the genes that show a preferential expression in either the LV or the RV in vivo²⁶, we observed higher levels for *TBX5* and *GJA1* in the LV-like clusters (0, 2, 3, 4) as compared to the RV-like cluster (1), while the expression level for *VCAN* and *GPRIN3* was higher in the RV-like cluster (1) as compared to the LV-like clusters (0, 2, 3, 4) (Supplementary Fig. 8a, b). Notably, the expression level of *HAND1* was highest in the OFT-*

CM cluster 6 (Supplementary Fig. 8a). This is in line with reports in the literature showing that *HAND1* is expressed not only in LV CMs, but also in OFT and AVC CMs^{1,17}. OFT-like CMs (cluster 6) were almost exclusively found as derivatives of CPC-RA- (Fig. 4e) and expressed markers of human conoventricular CMs, such as *BMP2* and *RSPO3*¹⁸, as well as SMC markers (e.g., *ACTA2*, *TAGLN*, *COL1A2*) and only low expression levels of *MYL2*, according with the transcriptional profile of OFT-CMs in mice²⁸ (Fig. 4d; Supplementary Fig. 8c). Moreover, they did not express *TBX5*, which is consistent with data from human fetal tissue showing absence of *TBX5* in OFT structures²⁹ (Supplementary Fig. 8c). “

Concerning batch effect, the same line of argumentation as above applies (see answer to question 2.1). We again assessed the presence of a potential batch effect further by calculating the Euclidean distance between each cluster and plotted the data as dendrogram. This reveals that the proximity of clusters is based on biological information (e.g. proliferation, celltype) rather than sample, arguing against the presence of a significant batch effect (Supplementary Fig. 7a).

4.2 Related to the point above, the data shown for candidates such as *MYL2* and *TBX5* (Fig S6) suggest that these markers are indeed expressed, but they are not expressed in all cells of a given cluster but rather are expressed heterogeneously throughout all clusters, suggesting again that the current clustering analysis contains a substantial amount of heterogeneity still. Including that of important chamber-specific markers. In addition to interrogating known and expected markers in the analysis, the top differentially expressed candidates should be listed, for example as a supplementary table, to better understand what most prominently distinguishes the different CM clusters.

As already stated in our answer to 2.3 we are convinced that the heterogenous expression of low expressed transcripts such as *TBX5* is largely due to the high “dropout” rate seen in droplet-based sequencing technologies (figure for the reviewer). Concerning *MYL2*, the heterogenous expression observed among the CMs derived from CPC-RA- is indeed based on heterogenous cell populations, which was not well resolved in the old embedding and has improved substantially in the new separate embedding of the day 30 data (following the reviewer’s suggestion related to question 4.1). The new embedding shows that *MYL2* expression is present in the majority of RA- vCMs (now cluster 1) but largely absent in RA- OFT CMs (now cluster 6), which is in line with the transcriptional profile of OFT-CMs in mice²⁹. As discussed in our answer to question 1.6 we are convinced that the co-expression of *TBX5* and *MYL2* in the same clusters is due to the fact that *TBX5* is also expressed in human vCMs and that *TBX5* expressing CMs in our dataset do not represent atrial-like CMs, but vCMs.

We have replaced the plots accordingly. In addition, following the reviewer’s suggestion, we now provide DEG lists for all clusters in Supplementary Data 6 (please, see our response to point 4.4 below for details).

(a) Feature plots showing expression levels of *TBX5* and *NKX2.5* in the indicated samples at d1.5 and d4.5 in the UMAP plot shown in Fig. 2f. N = *NKX2.5* (eGFP+); T = *TBX5* (mCherry+). **(b)** Feature plots showing expression levels of *TBX5* and *NKX2.5* in the indicated samples at d1.5 and d4.5 in the UMAP plot shown in Fig. 2f. N = *NKX2.5* (eGFP+); T = *TBX5* (mCherry+). Missing values were imputed using ALRA. **(c)** Representative immunofluorescence images of obtained aggregates of indicated

samples one day after sorting. Cells were sorted based on TBX5 (mCherry) or NKX2.5 (eGFP) expression. T = TBX5, N = NKX2.5.

4.3 While scRNAseq is a great first pass at the question of what CPC progeny is formed, the important conclusions in the paper should be strengthened by IF analysis of the many available distinct chamber-specific cardiomyocyte markers (MLC2v, MLC2a, Nr2f2) in both -RA and +RA derived populations.

Following the reviewer's suggestion, we have performed immunostaining for MLC2v and MLC2a as well as flow cytometry for the atrial marker NR2F2 to complement the data derived from the electrophysiologic studies (Fig. 1h,i). In addition, we have extended the qPCR panel and added *SHOX2* as well as *NR2F2* (Fig. 1g; Supplementary Fig. 1d). These new data confirmed the results from the electrophysiologic studies (please, see our response to point 4.4 below for details).

4.4 The paper in general ignores the fact that atrial cardiomyocytes can also be derived from FHF cells, albeit at a much lower propensity. Atrial markers, as well as left right ventricular markers should be included in the current analysis.

We agree with the reviewer, that FHF cells also give rise to atrial cardiomyocytes. As suggested by the reviewer, we have now included immunostaining for MLC2a and provide flow cytometry data as well as qPCR data for NR2F2 to further assess the amount of atrial CMs (Fig. 1h, i). Notably, the newly included RA treatment scheme (1 μ M for 10 days) indeed increases the amount of atrial CMs most likely reflecting FHF-derived atrial cells. However, as the reviewer already correctly stated, the potency of this population in giving rise to atrial cells was limited. We now discuss this in the results part of the manuscript.

Page 5, line 175: *"To confirm our findings on the transcriptomic level we performed immunofluorescence analysis for the ventricular specific myosin light chain isoform – MLC2v, and atrial isoform – MLC2a (Fig. 1h) as well as flow cytometry-based quantification of cells staining positive for the atrial specific marker NR2F2 (Fig. 1i). Without addition of RA most of the cells were exclusively positive for MLC2v and we observed only around 10% of cells staining positive for NR2F2 confirming that most cells acquired a ventricular-like fate (Figure 1h, 1i). This did not change upon addition of RA up to 1 μ M for 4 days (Figure 1h, 1i). However, longer exposure of a higher dose of RA (10 days, 1 μ M) resulted in an increase of MLC2a positive CMs (Fig. 1h) as well as an increase in NR2F2+ cells (Fig. 1i) suggesting partial atrial fate acquisition in line with previous reports"*

Additionally, we have now assessed the expression of genes preferentially expressed in LV and RV CMs from human embryonic hearts and have included data for *TBX5*, *GJA1*, *VCAN*, *GPRIN3* and *HAND1* into the new supplementary figure 8a and discuss this now in the results part of the manuscript (please see answer to question 1.3 for details).

5. Fate decision tree analysis:

Using URD trajectory analysis is an informative approach to understand fate decisions, particularly as the data for both CPC and differentiated cells are available.

5.1 To split the analysis between the -RA and +RA samples seems an unnecessarily biased way to do this however. Rather, the samples should be properly merged, batch corrected, subjected to URD analysis and then differences between -RA and +RA trajectories identified.

We agree with the reviewer, that the URD analysis suffers from limitations. We aimed at following the reviewer's suggestion in calculating a URD trajectory from the combined dataset, but failed due to the huge number of cells, for which URD is not designed. Random subsampling of the dataset to circumvent this issue resulted in variable results, so that we did not feel confident to present a URD tree from the combined analysis. Due to the above mentioned limitations and the fact, that we do not generate novel insights through the URD analysis we have decided to remove the analysis from the manuscript, to gain space for highlighting the new biological data generated as part of the revision process.

5.2 Similarly to previous analysis, the study could benefit from identifying the candidates that most strongly mark the branching points in an unbiased manner, in addition to illustrating expression and segregation of known/expected markers.

We removed the URD analysis from the manuscript (see comment to question above).

6. Contribution of CPCs during mouse heart development:

This part of the study represents an ambitious approach to test CPC potential in vivo, during early heart development. Some comments apply as follows:

The manuscript states that all CPCs have successfully integrated into the host myocardium. The data presented (Fig. 5c) do not support this conclusion, rather the CPC-derived cells appear to reside on the outside of the mouse heart, and the image quality does not allow to evaluate if the HNA+ cells do indeed express troponin. The manuscript should also state if the CPCs from both conditions were sorted prior to injection, to help understand how these results compare to the previous analyses.

We apologize for the suboptimal images in Figure 5, which do not allow to properly assess positions of the cells. We now provide additional images including section views showing integration of the injected CPCs as part of the supplementary material (Supplementary Fig. 10a). We now also provide single-channel images for the integrated HNA+ cells as part of the supplementary figure 10b and d that demonstrate that HNA+ cells have a very similar staining pattern for cTnT as compared to the surrounding host myocardium. The cells were not sorted prior to injection since (as stated above in point 2.3) we agree with the reviewer that sorting of the CPC populations is not strictly necessary and we took advantage of this in our CPC injection studies. We state this now more clearly in the manuscript Methods starting on page 22, lines 805.

“Unsorted CPCs at d4.5 of differentiation were dissociated using Accutase as described above, resuspended in the CDM-BSA media and injected into the heart region of dissected mouse embryos at the cardiac crescent stage. At this stage, structures within the mouse embryo are small and FHF and SHF regions appear in the close proximity enabling targeting the whole heart-forming region with a single injection.”

The distinction between a left versus a right ventricular location at this early stage seems challenging, as the chambers have not fully formed yet. It should be addressed what criteria were used to assign the cells a specific location.

We agree with the reviewer that assigning compartments in these early stages of heart development is challenging in particular in *ex vivo* cultured embryos. This is the reason, why we decided for relatively broad categories when allocating the position of cells, as we now state more clearly starting on page 11, line 388:

“Since separation of the heart compartments in these early stages of heart development is still incomplete, we defined rather broadly delineated cardiac areas (OFT/RV, RV/LV, LV, IFT) to evaluate the regional contribution of the injected CPCs, as shown in Supplementary Movie 1”

To enable the reader to better assess how we assigned locations, we now provide additional images and section views (Supplementary Fig. 10a) as well as a video of a 3D reconstruction (Supplementary Movie 1). In addition, we provide a more detailed explanation as part of the methods section, starting on page 23, line 854 as follow:

“OFT and IFT could be clearly distinguished by their morphology and its right/cranial or caudal position, respectively. OFT and RV at the analyzed stage of heart development are not as clearly separated therefore combined in the category OFT/RV. Human cells which were not clearly found in left or right ventricle were assigned to the category RV/LV.”

The analysis seems underpowered if the data is correctly interpreted: 5 embryos in -RA and 6 embryos in +RA?

Following the reviewer’s suggestion, we have performed new injection experiments and added 4 additional embryos, split between the conditions. The results of these new experiments are in line with the previous findings, highlighting the reproducibility of the previous data. Additionally, we provide now also new results on injection of CPCs from HLHS patient-specific hiPSC lines (see point 7 below).

We have included the data from the additional embryos into the revised Figure 5f.

7. HLHS disease modeling:

While an interesting question with respect to disease modeling in different differentiation conditions, or different CPC populations, the conclusions drawn are inappropriate given that the only data included are PCR data on bulk cultures from one control and one HLHS line each, without any additional characterization of the cells at any of the different stages.

We thank the reviewer for the interest in our disease model and agree on the limitations highlighted. To address these, we have now included a second control cell line as well as a second unrelated patient-specific hiPSC HLHS line, which show results in line with our previous findings. Details on the lines including de-novo mutations and hPSCreg line IDs are now provided as part of the method section. In addition, we now also provide stainings for MLC2v as well as cTnT. This data shows that HLHS patient-specific hiPSCs, despite the perturbed transcriptional profile of the CPCs, gave rise to differentiated CMs but at lower efficiency in particular in the CPC-RA- conditions as compared to the control (Fig. 6 c). In addition, these CMs showed much less, if any, expression of MLC2v, suggesting impaired attainment of ventricular identity (Fig. 6c). To investigate that further, we now also provide functional data on the HLHS-derived CPCs by performing the *ex vivo* injection experiments with CPC-RA+ and CPC-RA- derived from the HLHS-lines. These experiments revealed that the HLHS-derived CPCs still preferentially located to their respective heart compartments (CPC-RA+ to the LV, CPC-RA- to the RV/OFT). However, we observed some cells of both conditions in other compartments of the heart suggesting that the progenitor identity is only partially conserved, in line with our findings of altered expression of heart-field specific markers. In addition, the percentage of cTnT⁺ cells among the HLHS-derived CPC-RA+ was significantly reduced as compared to wildtype

CPC-RA+ (43% vs. 78%), suggesting a lower capacity of HLHS-derived CPC-RA+ to differentiate into LV vCMs. Notably, the HLHS-derived CPC-RA- showed a similar amount of cTnT⁺ cells in the injection experiments as compared to wildtype CPC-RA⁻ (29% vs. 34%) suggesting that the differentiation defect observed in the 2D system for this population can be, at least partially, compensated in the ex vivo setting.

Taken together, this new set of data suggests that HLHS patient-specific hiPSCs fail to acquire a FHF-like fate at the progenitor level, resulting in perturbed CMs without proper LV vCM identity (also supported by a reduced capacity to form cTnT⁺ cells when injected ex vivo). Notably, they also show a defect in aSHF progenitor differentiation resulting in CMs with significant dysregulation of vCM and OFT-CM markers. However, this does not result in a reduced number of cTnT⁺ cells in the ex vivo injection experiments for this population.

We have included this new data together with the previous data into a new main figure 6 and describe it in the results part of the manuscript:

Starting on page 11, line 417: *“Differentiating two of the previously reported HLHS hiPSC lines (see Methods for details on line ID and genotype) into FHF-like (CPC-RA+) and aSHF-like progenitors (CPC-RA-), we observed, in both conditions, a downregulation of the key CPC marker NKX2.5 (Fig. 6a), which is in agreement with our previous findings⁴¹. This was paralleled by a substantial reduction of TNNT2 levels in CPC-RA+, suggestive of impaired myocytic commitment, and a significant dysregulation of the aSHF-marker FGF10, together indicating that altered CPC-fate acquisition in HLHS seems to affect both, the FHF-like (CPC-RA+) as well as the aSHF-like (CPC-RA-) population (Fig. 6a). During further differentiation, HLHS patient-specific hiPSCs gave rise to CMs in both conditions as shown by immunofluorescence analysis of cTNT and qPCR for TNNT2, although at lower efficiency (in particular in the CPC-RA- conditions) as compared to the control (Fig. 6b, Supplementary Fig. 10c). In addition, these CMs showed much less, if any, expression of MLC2v, suggesting impaired attainment of ventricular identity (Fig. 6c). qPCR analysis confirmed a reduction of ventricular cardiomyocyte transcripts (MYH7, MYL2; Fig. 6d) and revealed increased MYL7/MYL2 and TNNI3/TNNI1 isoform ratio as indicator of cardiac immaturity (Fig. 6d), corroborating our previous findings on maturation defect in HLHS patient-specific CMs⁴¹. Notably, HLHS CMs arising from the CPC-RA+ also displayed lower levels of the LV-specific marker FHL2 (Fig. 6d). At the same time, they exhibited higher expression levels of genes related to OFT-CMs (RSPO3) rendering their transcriptional profile similar to the patient-specific CMs arising from the CPC-RA-, indicative of a failed commitment to a LV CM fate (Supplementary Fig. 10c). We also observed dysregulation of OFT-CM transcripts (RSPO3) in the CMs derived from the CPC-RA-, which could likely be a consequence of the altered FGF10 level in the progenitors, since this gene is crucially required in the aSHF for proper OFT formation in mice⁴²⁻⁴⁴. The fact that both progenitor pools (CPC-RA+ and CPC-RA-) and derived CMs appear affected is consistent with observations in vivo showing that CMs from the LV as well as the RV of HLHS patient share similar transcriptional defects^{41,45}.*

To further evaluate the fate commitment of HLHS-derived CPCs, we injected the cells into the heart region of developing mouse embryos at the cardiac crescent stage and subjected those to whole-embryo ex vivo culture for 24 or 48 hours. Immunofluorescence analysis indicated that HLHS-derived CPCs still contributed preferentially to their respective heart compartments (CPC-RA+ to the LV, CPC-RA- to the RV/OFT). However, their segregation to specific cardiac areas was less defined as compared to the control, with cells from both conditions being also present in other heart compartments (Fig. 6e-g; Supplementary Fig. 10d). In addition, we observed a significant reduction in the percentage of human cTnT⁺ cells after injection of the CPC-RA+ as compared to the control (43% vs. 78%, $p=0,0087$), suggesting impaired CM differentiation in line with the findings from the 2D culture (Fig. 6h; Supplementary Fig. 10e; Supplementary Data 8). Interestingly, this was not the case in the embryos injected with the HLHS-derived CPC-RA-, where we detected a similar proportion of

human cTnT⁺ cells as compared to the control (29% vs. 34%) (Fig. 6h; Supplementary Fig. 10e; Supplementary Data 8). This suggests that the differentiation defect seen in the 2D system for this population can be, at least partially, compensated in the ex vivo setting in line with observations in patients where RV cardiomyocytes, although altered on the transcriptional level, form a heart chamber^{41,45}.”

Minor comments:

1. The citations for RA signaling to generate atrial cardiomyocytes are incomplete. Please add Devalla et al., 2015 and the original work by Zhang et al., 2011

We apologize for omitting these references and have included them into the manuscript.

2. Figure 2B: Cluster 17 in the -RA CPC sample in the UMAP is labelled in such a way that it might be interpreted as two clusters, 7 and 1

We have corrected the labeling.

REVIEWER COMMENTS

Reviewer #1 (Remarks to the Author):

Zawada and coauthors have responded to many issues raised by reviewers, but some issues remain. There are concerns as to whether RA treatment is actually inducing all of the JCF or only a portion particularly given that there does not appear to be an appreciable induction of ventricular cardiomyocytes, which can also derive from the JCF. At the very least, the authors should revise the interpretation of their paper to more accurately reflect the results. Additionally, there are major concerns about the rigor and reproducibility of the data. In particular, the lack of replicates is a key issue that as pointed out by other reviewers raises concerns about whether there are potential batch effects that could be confounding the interpretation of the data. Furthermore, these data were only compared to Tyser et al., but Zhang et al data should be also used, particularly as these data and findings more comprehensively compared and integrated these mouse datasets. Such analysis could help improve the overall interpretation of the data and possibly even resolve issues raised by other reviewers about the identity of clusters discovered from single cell analysis. However, without adequately addressing this rigor and reproducibility issue, where it is generally standard to provide replicates for such -omic data, it is challenging to draw any meaningful scientific conclusion from current findings. Overall, major issues remain that still need to be better addressed in order for the manuscript to be a significant contribution to the cardiac field.

Reviewer #2 (Remarks to the Author):

The manuscript from Zawada et al. has been well revised. The authors have completed significant revisions based on my comments. I find the current version of the manuscript considerably improved. The authors addressed my main concern regarding the use of the marker to support their findings. They further confirmed that depending on dosage of RA atrial marker expression are increased (Fig 1g). The authors included another RA scheme to their protocol (1 μ M RA over 10 days). When they applied this scheme they observed a more pronounced increase in atrial markers on the expression level. They have revised their manuscript regarding these novel results. The authors completed their electrophysiological studies by adding immunostaining and flow cytometry for atrial marker NR2F2. They replied to my comment regarding previous findings showing that RA treatment increases atrial cells. They also revised the suboptimal presentation of some figures. To address my comments regarding the expression profile of RA receptors and co-receptors, they performed new analysis of scRNAseq data. Based on this analysis they concluded that the expression profile of RA receptors and co-receptors was comparable between cells at day 1.5 and the mid/late primitive streak cells in the mouse. I am satisfied with the answers of the authors concerning the effects of batch and the use of URD. The authors expanded their analysis of HLHS hiPSC-lines and included new data (Fig 6).

Therefore, the authors responded to all my comments.

Reviewer #3 (Remarks to the Author):

The authors have performed extensive revisions for the manuscript and have attempted to address every single comments raised by the 3 reviewers, which I find impressive. The large majority of the new data and analyses added is convincing and supports, or expands on the previous conclusions. Overall this is an interesting manuscript with rigorous experimentation and analysis and the strengths definitively outweigh the minor remaining concerns. For final comments please point by point below:

Points 1 and 2 (differentiation to new progenitor populations with low and early RA exposure): Some of the key experiments suggested by multiple reviewers are added in the revised manuscript and

support the authors previous conclusions:

- Confirmation of comparable cardiac differentiation efficiency via NKX2-5 gene expression (Sup. Fig. 1)

- IF analysis for MLC2a and MLC2v in differentiated (D30) CMs (Fig. 1h). The IF for MLC2v is particularly convincing. The newly added condition (RA for 10 days) is somewhat confusing, as the cells seem to not express MLC2v, but only 30% of the cells express NR2F2 – it is not clear what these cells are, but it also matters less to the current project.

- Extensive additional analysis of the scRNAseq data (confirmation of markers used for sorting, expanded cluster annotation, stratification of proliferation in the various clusters, etc). The comparison with the in vivo data from Tyser et al is particularly relevant as it demonstrates the similarity of the RA+ population with the most differentiated cell cluster at a similar stage in the embryo.

Note: The notion that none of the conditions give rise to a significant amount of atrial cells remains confusing, but the authors argue that this is explained by a protocol that appears to be rather different from what the majority of the field is using. In the context of how their findings might be interpreted and potentially adopted by others, it seems important to clearly add this information to the manuscript, ideally both in the results section and in more detail in the discussion.

Point 3 (JCF): I commend the authors for going to great lengths to assess the differentiation potential of ITG8⁻ and ITG8⁺ progenitors (Fig. 3). The results suggest that ITG8 does not specifically label the JCF cells in vitro. The enrichment of cells that can give rise to both CMs and epicardium is rather modest in the ITG8⁺ condition. Given the technically challenging approach and the transparent data display and interpretation provided by the authors this seems a valid analysis to be included in the manuscript never the less.

Point 4 (analysis of differentiated cells): All of my points were addressed in detail and the new data and new analysis strongly support the authors' previous conclusions on the fate of the differentiated CMs from the CPCs.

Point 5 (URDs) I agree with the author's decision to remove this analysis, as it doesn't provide any new information and the paper has more exciting data to be highlighted.

Point 6 (in vivo integration): I understand the rationale for this experiment, and appreciate how technically challenging it is. The authors have significantly improved the quality of the data and added additional replicates to strengthen their conclusions. Despite the success of the revision for this part, I believe it's the least meaningful part of the study, because the numbers remain low and the developmental stage is too early to distinctly identify spatial contribution.

Point 7 (HLHS): the authors added an extensive amount of new data to this aspect of the study, such as important additional lines and new analyses. Overall this is a strong improvement, some questions remain:

- The data on troponin T expression in the -RA protocol in control cells in Fig. 6a and Fig. 6b does not align (no message lots of protein?).

- The analysis is underpowered for the conclusions drawn (4 and 5 embryos respectively; Fig. 6e-h)

- Please specify which HLHS line is used in the in vivo analysis

Point-by-point response to the reviewers' comments

We want to thank all Reviewers for their thoughtful comments and suggestions in the second round of revision, which helped us to further improve the manuscript. We have addressed all remaining issues raised by refining our transcriptomic analyses as well as *via* editorial revision, as outlined in the point-by-point response below.

REVIEWER COMMENTS

Reviewer #1 (Remarks to the Author):

Zawada and coauthors have responded to many issues raised by reviewers, but some issues remain. There are concerns as to whether RA treatment is actually inducing all of the JCF or only a portion particularly given that there does not appear to be an appreciable induction of ventricular cardiomyocytes, which can also derive from the JCF. At the very least, the authors should revise the interpretation of their paper to more accurately reflect the results.

We thank the Reviewer for raising this issue, which we now discuss as part of the discussion section. The concept of the JCF is rather novel and there is still a number of open questions, e.g. its relation to the extraembryonic tissue, its exact developmental origin and contributions, as well as its heterogeneity. Following up on previous reports in the mouse that described the common progenitor of myocardial and epicardial cells during cardiac development (Tyser et al., 2021; Zhang et al., 2021) we show that a similar progenitor population can be generated *in vitro* from hPSCs. However, we agree with the reviewer that it is possible that we generated only a subset of the JCF, and that other JCF-like progenitors might exist that exhibit a different myocytic vs epicardial differentiation potential as compared to what we observed. To acknowledge this aspect, we have added the following paragraph to the discussion section (page 14, starting at line 518):

“Whether the JCF-like cells arising in our differentiation conditions represent the entire JCF population or only a subset remains elusive. Further studies on the origin of this population, its relation to the extraembryonic tissue, its exact contributions to the developing heart as well as its heterogeneity are needed to shed further light on this aspect.”

Additionally, there are major concerns about the rigor and reproducibility of the data. In particular, the lack of replicates is a key issue that as pointed out by other reviewers raises concerns about whether there are potential batch effects that could be confounding the interpretation of the data.

We agree with the Reviewer that reproducibility is an important aspect in biomedical sciences, in particular when dealing with complex *in vitro* differentiation protocols. This led us to rigorously assess our differentiation protocol using an extensive qPCR panel, immunofluorescence stainings as well as flow cytometry analysis at various stages during the differentiation process providing data from at least three independent biological replicates for each assay (whenever applicable).

Concerning the interpretation of our scRNA-seq data, we respectfully disagree with the Reviewer that the likelihood of the presence of a batch effect which would substantially change the interpretation of the data is very high. To further substantiate this assumption, we have re-analyzed a whole transcriptome dataset (bulk RNA-seq) performed on sorted progenitors at day 4.5 (approx. 1000 cells per sample; for method details, please see Peschke et al., 2022). This dataset consists of 3 technical and 2 biological replicates (independent differentiation experiments) and was generated prior to the scRNA-seq data as a quality control experiment.

As depicted in the Figure for the Reviewer below, the majority of the variation in this dataset stems from the differentiation process per se (PC1), with the remaining variance being largely explained by the different treatment regiments (RA vs. noRA; PC2) with some minor biological variation, which seem not to interfere with the main sources of variance. Notably, in line with our observations in the scRNA-seq dataset, the differences between the various sorted populations arising in the presence of RA (eGFP+/mCherry-; eGFP-mCherry+; eGFP+/mCherry+) appear rather small likely representing different stages of development on the same differentiation trajectory.

Taken together, we are convinced that despite the potential presence of confounders our protocol is sufficiently rigorous and reproducible to allow investigations of the data as outlined in our manuscript. In addition, we want to highlight here, that we use the scRNA-seq data mainly for generating hypothesis, which we confirm by wet lab experiments with multiple biological replicates throughout the manuscript.

Figure 1: Bulk RNA-seq of cell populations sorted for mCherry and/or eGFP at d4.5 of differentiation with or without RA and hESCs collected at d0 before start of differentiation (hESCs) (6 samples: 2 biological and 3 technical replicates). Colors mark cell populations and shapes mark biological replicates. N = 6, n = 2.

Furthermore, these data were only compared to Tyser et al., but Zhang et al data should be also used, particularly as these data and findings more comprehensively compared and integrated these mouse datasets. Such analysis could help improve the overall interpretation of the data and possibly even resolve issues raised by other reviewers about the identity of clusters discovered from single cell analysis.

We thank the Reviewer for this helpful suggestion. Following the advice, we have now included the dataset from Zhang et al. (the cardiomyocyte subclustering as presented in their Figure 2) into our comparison with Tyser et al. As shown in the revised Fig. 2i, the two mouse datasets matched nicely in this analysis providing confirmation of their respective cardiac progenitor identities and stressing the high quality of both datasets. In addition, this analysis further confirmed the identity of our cardiac progenitor populations and in particular highlights that our JCF-like clusters (14 and 9) are very similar not only to the me5 cluster of Tyser et al., but also to the progenitor clusters CP6 and CP7 which represent the progenitors closely related to the extraembryonic mesoderm as described in Zhang et al. We adapted the according results part of the manuscript on page 7, starting at line 268, as follows:

“The cluster identities were further evaluated by integrating the cells from the non-proliferative CPC clusters (clusters 0, 1, 3, 9, 12, 14, and 17) with the scRNA-seq datasets of Tyser et al.⁶

as well as Zhang et al.⁵ covering stages of mouse embryonic heart development that include JCF emergence⁶. This analysis confirmed the FHF- and aSHF-like characteristic of the above-mentioned clusters and showed that clusters 14 and 9 are closely related with the mouse JCF cluster me5 described by Tyser et al. Notably, these cells were also very similar to the cardiac progenitor clusters CP6 and CP7 described by Zhang et al., which are related to the late extraembryonic mesoderm in mice and have a comparable transcriptional signature with the JCF. This suggests that cluster 9 as well as cluster 14 both represent JCF-like cells (Fig. 2i)."

However, without adequately addressing this rigor and reproducibility issue, where it is generally standard to provide replicates for such -omic data, it is challenging to draw any meaningful scientific conclusion from current findings. Overall, major issues remain that still need to be better addressed in order for the manuscript to be a significant contribution to the cardiac field.

We hope that the additional analysis outlined above persuaded the Reviewer about the relevance of our work. We are convinced that, despite the lack of replicates, our scRNA-Seq analyses combined with experimental validation have led to valuable insights on the emergence, transcriptional profile, and fate potential of human cardiovascular progenitors.

As a note, we kindly disagree with the Reviewer on the statement that multiple replicates of scRNA-seq data have yet become a "general standard" for manuscripts being currently published since many recent publications in journals with high quality standards still do not provide replicates and have nonetheless presented valuable scientific conclusions (e.g., Yang et al, Cell Stem Cell, 2022; Mikryukow et al., Cell Stem Cell, 2020; Luff et al., Nat Cell Biol, 2022; Hurley et al., Cell Stem Cell, 2020; Gunne-Braden, Cell Stem Cell., 2020). However, we do agree that replicates of scRNA-seq data are helpful in addressing questions related to batch effect and other confounders and are now, also due to declining prices, more and more common and should be included in newly generated datasets.

Reviewer #2 (Remarks to the Author):

The manuscript from Zawada et al. has been well revised. The authors have completed significant revisions based on my comments. I find the current version of the manuscript considerably improved. The authors addressed my main concern regarding the use of the marker to support their findings. They further confirmed that depending on dosage of RA atrial marker expression are increased (Fig 1g). The authors included another RA scheme to their protocol (1uM RA over 10 days). When they applied this scheme they observed a more pronounced increase in atrial markers on the expression level. They have revised their manuscript regarding these novel results. The authors completed their electrophysiological studies by adding immunostaining and flow cytometry for atrial marker NR2F2. They replied to my comment regarding previous findings showing that RA treatment increases atrial cells. They also revised the suboptimal presentation of some figures. To address my comments regarding the expression profile of RA receptors and co-receptors, they performed new analysis of scRNAseq data. Based on this analysis they concluded that the expression profile of RA receptors and co-receptors was comparable between cells at day 1.5 and the mid/late primitive streak cells in the mouse. I am satisfied with the answers of the authors concerning the effects of batch and the use of URD. The authors expanded their analysis of HLHS hiPSC-lines and included new data (Fig 6).

Therefore, the authors responded to all my comments.

We thank the Reviewer for appreciating the additional data and thank them again for helpful suggestions and comments that led us to improve our manuscript substantially.

Reviewer #3 (Remarks to the Author):

The authors have performed extensive revisions for the manuscript and have attempted to address every single comments raised by the 3 reviewers, which I find impressive. The large majority of the new data and analyses added is convincing and supports, or expands on the previous conclusions. Overall this is an interesting manuscript with rigorous experimentation and analysis and the strengths definitively outweigh the minor remaining concerns. For final comments please point by point below:

We thank the Reviewer for appreciating our efforts and for helpful suggestions and comments that improved our manuscript substantially.

Points 1 and 2 (differentiation to new progenitor populations with low and early RA exposure): Some of the key experiments suggested by multiple reviewers are added in the revised manuscript and support the authors previous conclusions: -Confirmation of comparable cardiac differentiation efficiency via NKX2-5 gene expression (Sup. Fig. 1)-IF analysis for MLC2a and MLC2v in differentiated (D30) CMs (Fig. 1h). The IF for MLC2v is particularly convincing. The newly added condition (RA for 10 days) is somewhat confusing, as the cells seem to not express MLC2v, but only 30% of the cells express NR2F2 – it is not clear what these cells are, but it also matters less to the current project. -Extensive additional analysis of the scRNAseq data (confirmation of markers used for sorting, expanded cluster annotation, stratification of proliferation in the various clusters, etc). The comparison with the in vivo data from Tyser et al is particularly relevant as it demonstrates the similarity of the RA+ population with the most differentiated cell cluster at a similar stage in the embryo.

Note: The notion that none of the conditions give rise to a significant amount of atrial cells remains confusing, but the authors argue that this is explained by a protocol that appears to be rather different from what the majority of the field is using. In the context of how their findings might be interpreted and potentially adopted by others, it seems important to clearly add this information to the manuscript, ideally both in the results section and in more detail in the discussion.

We thank the Reviewer for these comments and agree on the importance of highlighting the distinct features of our protocol in relation to published literature. We have already discussed this in the discussion section of the revised manuscript (page 13, starting at line 489), but we have now also added a clarifying statement at the beginning of the results section (page 3, starting at line 131):

“To investigate the influence of RA dosage and timing on the appearance and characteristics of early human cardiovascular progenitors in the cardiogenic mesoderm, we utilized a growth-factor based protocol for the directed differentiation of hPSCs towards CMs, which is distinct from other in vitro CM differentiation protocols utilizing RA to generate atrial cardiomyocytes (Fig. 1a).”

Point 3 (JCF): I commend the authors for going to great lengths to assess the differentiation potential of ITG8- and ITG8+ progenitors (Fig. 3). The results suggest that ITG8 does not specifically label the JCF cells in vitro. The enrichment of cells that can give rise to both CMs and epicardium is rather modest in the ITG8+ condition. Given the technically challenging approach and the transparent data display and interpretation provided by the authors this seems a valid analysis to be included in the manuscript never the less.

We thank the Reviewer for their careful assessment of these new data.

Point 4 (analysis of differentiated cells): All of my points were addressed in detail and the new data and new analysis strongly support the authors' previous conclusions on the fate of the differentiated CMs from the CPCs.

We thank the Reviewer for their appreciation of our efforts.

Point 5 (URDs) I agree with the author's decision to remove this analysis, as it doesn't provide any new information and the paper has more exciting data to be highlighted.

We are glad that the Reviewer agrees with our decision.

Point 6 (in vivo integration): I understand the rationale for this experiment, and appreciate how technically challenging it is. The authors have significantly improved the quality of the data and added additional replicates to strengthen their conclusions. Despite the success of the revision for this part, I believe it's the least meaningful part of the study, because the numbers remain low and the developmental stage is too early to distinctly identify spatial contribution.

We thank the Reviewer for this comment.

Point 7 (HLHS): the authors added an extensive amount of new data to this aspect of the study, such as important additional lines and new analyses. Overall this is a strong improvement, some questions remain: The data on troponin T expression in the -RA protocol in control cells in Fig. 6a and Fig. 6b does not align (no message lots of protein?). The analysis is underpowered for the conclusions drawn (4 and 5 embryos respectively; Fig. 6e-h). Please specify which HLHS line is used in the in vivo analysis

We thank the Reviewer for this comment that helped us to improve the suboptimal presentation of the data, which led to confusion. Fig. 6a represents the analysis of mRNA expression on d5 while Fig 6b shows representative images captured at d45. The fact that this data stems from two very different time points explains the "mismatch". The qPCR data of TNNT2 expression in d30 healthy and diseased cardiomyocytes, which corresponds more closely to the data presented in Figure 6b, is indicated in Supplementary Fig 10c. We have improved the labeling in the figure to make this clearer.

Following the Reviewer's suggestion, we now specified the contribution of the different HLHS lines in all relevant data in the main figure as well as in the Supplementary Datasets. Moreover, we agree that the number of embryos in the HLHS study is on the lower end and discuss this limitation now in the Results part on page 13, starting at line 463 of the revised manuscript as follows:

"As far as the limited number of embryos in each group allows, this suggests that the differentiation defect seen in the 2D system for this population can be, at least partially, compensated in the ex vivo setting in line with observations in patients where RV cardiomyocytes, although altered on the transcriptional level, form a heart chamber^{41,45}."